# Ca²⁺ oscillation in vascular smooth muscle cells control myogenic spontaneous vasomotion and counteract post-ischemic no-reflow

Jinze Li [1,2,3,4,5] ✉, Yiyi Zhang[1,2,3,4], Dongdong Zhang[1,2,3], Wentao Wang[1,2,3], Huiqi Xie[1,2,3], Jiayu Ruan [1,2,3], Yuxiao Jin [1,2,3], Tingbo Li[1,2,3], Xuzhao Li[1,2,3], Bingrui Zhao[1,2,3], Xiaoxuan Zhang[1,2,3], Jiayi Lin[1,2], Hongjun Shi[1,2] & Jie-Min Jia [1,2,3,5] ✉

Ischemic stroke produces the highest adult disability. Despite successful recanalization, no-reflow, or the futile restoration of the cerebral perfusion after ischemia, is a major cause of brain lesion expansion. However, the vascular mechanism underlying this hypoperfusion is largely unknown, and no approach is available to actively promote optimal reperfusion to treat no-reflow. Here, by combining two-photon laser scanning microscopy (2PLSM) and a mouse middle cerebral arteriolar occlusion (MCAO) model, we find myogenic vasomotion deficits correlated with post-ischemic cerebral circulation interruptions and no-reflow. Transient occlusion-induced transient loss of mitochondrial membrane potential (ΔΨm) permanently impairs mitochondria-endoplasmic reticulum (ER) contacts and abolish Ca²⁺ oscillation in smooth muscle cells (SMCs), the driving force of myogenic spontaneous vasomotion. Furthermore, tethering mitochondria and ER by specific overexpression of ME-Linker in SMCs restores cytosolic Ca²⁺ homeostasis, remotivates myogenic spontaneous vasomotion, achieves optimal reperfusion, and ameliorates neurological injury. Collectively, the maintaining of arteriolar myogenic vasomotion and mitochondria-ER contacts in SMCs, are of critical importance in preventing post-ischemic no-reflow.

Stroke is a devastating vessel-originating brain disease that kills millions of people each year. Ischemic stroke, the dominant type, occludes vessels, resulting in neuronal death by depriving cells of oxygen and nutrition. Re-establishing optimal and thorough blood perfusion, which ensures brain recovery and is a strong predictor of clinical outcome[1–3], has remained a challenge due to the fairly futile recanalization rate among patients[4,5].

In the clinic, the incomplete restoration of the cerebral perfusion despite full recanalization of the previously occluded territory is termed the no-reflow phenomenon, which is reported both in ischemic brain and heart diseases and regarded as one of the direct causes of lesion growth[6–8]. While current therapeutics succeed in removing intraluminal clots, they provide no means to proactively promote optimal perfusion in the affected

regions[9,10], where vascular walls that are initially healthy are often secondarily damaged by the initial occlusion, with consequent long-lasting hypoperfusion. At present, few achievements have been made toward treating no-reflow due to the incompletely understood vascular pathological changes after ischemia.

Notably, vascular segments are impaired along with the ischemic stroke, and their function is a major determinant of the depth of ischemic injuries[11]. While it has been well studied that ischemic stroke affects multiple aspects of the cerebral vasculature[12,13], the direct causal link underlying vascular property and post-ischemic cerebral blood flow regulation remains unknown. Arterial spontaneous vasomotion manifests a vital aspect of the myogenic characteristics of the vascular smooth muscle cells (SMCs),

[1]Key Laboratory of Growth Regulation and Translational Research of Zhejiang Province, School of Life Sciences, Westlake University, Hangzhou, China. [2]Westlake Laboratory of Life Sciences and Biomedicine, Hangzhou, China. [3]Laboratory of Neurovascular Biology, Institute of Basic Medical Sciences, Westlake Institute for Advanced Study, Hangzhou, China. [4]These authors contributed equally: Jinze Li, Yiyi Zhang. [5]These authors jointly supervised this work: Jinze Li, Jie-Min Jia. ✉e-mail: lijinze@westlake.edu.cn; jiajiemin@westlake.edu.cn

facilitating periodic sinusoidal contraction and relaxation of vascular walls. Vasomotion evokes corresponding variations in blood flow velocity, namely flowmotion[14], which is nowadays thought to promote functional microcirculation[15,16], protect tissue oxygenation[17–20], and drive paravascular clearance of solutes[21,22] in the central nervous system. Theoretically, it has been discussed that oscillatory haemodynamic therapies may improve the perfusion and oxygenation of impaired tissue[23], however, the following questions have not been answered: (1) Does ischemic stroke impair the myogenic spontaneous vasomotion of the cerebral arteriolar network? (2) What is the temporal relationship between these deficits and the occurrence of no-reflow? (3) Can the maintenance of vasomotion effectively counteract the no-reflow phenomenon, facilitate optimal reperfusion, and enhance post-stroke brain recovery?

The ultraslow rhythmicity of vasomotion centered at 0.1 Hz[24,25] is thought to be attributable to SMC-mediated myogenic control. The intrinsic capability of SMCs to contract and relax periodically and spontaneously is called SMC contractility, which eventually relies on $Ca^{2+}$-dependent rhythmic activation of cross-bridge cycling[14]. As the central hub for rhythmic vasomotion, physiological $Ca^{2+}$ oscillation in SMCs is underpinned and regulated by a plethora of $Ca^{2+}$ channels, pumps, transporters, and binding proteins in complex ways[26–28]. Intriguingly, in the subcellular level, the interactions between the mitochondria and endoplasmic reticulum (ER) have been frequently observed to participate in $Ca^{2+}$ transportation[29–31]. Moreover, the $Ca^{2+}$ shuttling between the mitochondria and ER has been implicated in a potential pacemaker role for generating $Ca^{2+}$ oscillation[32]. Besides, synthetic mitochondria-ER linker has been proven to enhance mitochondria-ER association and calcium coupling[33]. More recently, overexpression of the Mitochondria-ER linker in astrocytes successfully promotes blood vessel remodeling in a brain injury model by restoring $Ca^{2+}$ oscillations in astrocyte endfeet[34]. Taken together, we reasoned that mitochondria-ER contacts might be the potential keystone for maintaining intrinsic SMC contractility, functional vasomotion, and subsequent cerebral circulation. Particularly, decoding the vascular pathogenesis behind post-ischemic injury is crucially important for post-recanalization intervention, as strategies to counteract no-reflow do not exist.

Here, by combining in vivo cerebral blood flow detection and in vivo two-photon laser scanning microscopy, we obtained systematic and repeated measurements of mouse brain perfusion, vascular wall dynamics, blood cells velocity and flux both in the arteriolar network of the middle cerebral artery and its feeding brain parenchyma capillaries. Strikingly, we found that dampened myogenic spontaneous vasomotion along the arteriolar wall is the major vascular pathological phenotype during the recanalization period in the transient middle cerebral artery occlusion mouse model. Meanwhile, no-reflow was reproduced in company with cerebral circulation interruptions. We further found that ischemic stroke could induce transient loss of mitochondrial membrane potential and, subsequently, a long-lasting disruption in the subcellular mitochondria-ER contacts and $Ca^{2+}$ oscillation in SMCs. Furthermore, specific overexpression of ME-Linker in SMCs in vivo restored cytosolic $Ca^{2+}$ oscillation and arteriolar vasomotion. Due to these improvements, we finally found that arteriolar vasomotion replenishment achieved optimal capillary reperfusion and alleviated the no-reflow-associated neurological injuries. Our study is highlighted by the urgent need for elucidating the difficulty in post-ischemic no-reflow treatment.

## Results
### Myogenic spontaneous vasomotion mainly governs cerebral arterioles but not venules in anesthetized mice
To investigate myogenic spontaneous vasomotion along the whole vasculature, we used two-photon laser scanning microscopy (2PLSM) to capture cerebral vascular wall motility in pentobarbital-anesthetized mice. The cerebral vasculature of *SMCreER:Ai47* mice was visualized through the intravenous injection of the red fluorescent dye rhodamine B-dextran, where the arteriolar and venular SMCs (sparsely) were reported by three tandemly linked GFP molecules under the control of SMA promotor-driven CreER activity[35,36] (Fig. S1a, S1b). Interestingly, we found that the motility

between the two sides of the arteriolar wall was heterogeneous, as reflected in the rhythmic differences of the time-lapse radius changes traced between bilateral arteriolar walls of the same arteriole (Fig. S1c, S1d). We found there was no correlation between bilateral arteriolar walls of the same arteriole (correlation coefficient, R = 0.001 ± 0.046) (Fig. S1e). The variability of the correlation coefficient values is likely to be the result of the heterogeneous contractility of the two sides of the arteriolar wall, due to the heterogeneous morphological patterns of the arteriolar SMCs[37]. Therefore, we analyzed the radius, rather than the diameter, by reslicing the time-lapse images and calculating the vascular radius change curves (Fig. S1f, S1g).

In anesthetized mice, time-lapse imaging of the cerebral vessels had suggested that myogenic spontaneous vasomotion only governs the arteriole and penetrating arteriole (PA), where the vascular segments were featured by covering with the concentric ring-like arteriolar SMCs. On the contrary, in the stellate-shaped venular SMCs covered venules, no vasomotion was observed (Movie 1). Fourier transform analyses of the time courses of the radius change data revealed an obvious peak around 0.1 Hz in arterioles but not in venules (Fig. S1h), consistent with previous reports in awake mice[22]. The full-frequency (0–0.3 Hz) domain power in arterioles was significantly higher compared to venules, suggesting arterioles fluctuated greater than venules (Fig. S1i). Furthermore, to gain quantitative insights into multivariate oscillatory features of myogenic spontaneous vasomotion, we developed the 'vasomotion index', including frequency of rhythmic cycles, standard deviation (SD) of the peak intervals, and amplitude of changes in the vascular radius. The reliability of our vasomotion index was confirmed by the successful detection of the differences between arterioles and venules, which has been documented that venular walls are less active than arteriolar walls in the awake mouse brain[22]. As expected, venular walls fluctuated with small amplitudes rarely (0.0249 ± 0.00579 Hz, 0.547 ± 0.0922%) (Fig. S1j). In addition, the peak interval SD of venule (20.37 ± 5.033 s) was much higher compared to arteriole (6.663 ± 0.7265 s) and PA (9.076 ± 1.062 s), indicating venules behaved much more irregularly than arterioles (Fig. S1k). Besides, we found that there was no difference in frequency between arterioles (0.152 ± 0.0126 Hz) and penetrating arterioles (0.126 ± 0.0151 Hz), whereas the latter had a larger oscillatory amplitude than the former (4.88 ± 0.414% vs. 2.78 ± 0.178%) (Fig. S1l). While there is no significant different in arteriolar vasomotion index between male and female mice, indicating sex-related factors had very limited impact on the characteristics of arteriolar myogenic vasomotion physiologically (Fig. S2a–c). These data demonstrated that the vasomotion index that we developed can quantitively assess myogenic spontaneous vasomotion features with rigorous and unbiased sensitivity. Our findings suggested that the myogenic vasomotion of arteriolar SMCs should be, if not all, the prime driving force of the spontaneous force in anesthetized mice. In this way, we move our focus on the cerebral arterioles which manifest robust myogenic spontaneous vasomotion, rather than the quiet cerebral venules, in the following study.

### Stroke evokes long-lasting damage in myogenic spontaneous vasomotion, correlating with the no-reflow time window
To determine whether there is any pathological change of arteriolar myogenic spontaneous vasomotion along with ischemic stroke, we implemented a transient (2 h) monofilament-mediated middle cerebral artery occlusion (MCAO) and a subsequent 22 h reperfusion to mimic ischemic injury, and performed in vivo imaging of the identical middle cerebral artery (MCA) in before and post-ischemia conditions under 2PLSM (Fig. 1a) (Movie 2). Consistent with our previous experiments, physiologically, arterioles revealed sinusoidally fluctuating myogenic spontaneous vasomotions (Figs. 1b, c). Intriguingly and surprisingly, ischemic stroke diminished or even sometimes deprived these motive features, as we can observe a significant attenuated trend in the consecutive kymographs and time-series curves of the arteriolar radius changes (Fig. 1b, c). Notably, Fourier transform analyses revealed that myogenic spontaneous vasomotion after stroke exhibited significantly lower power in the frequency dimension than that recorded before stroke, with the frequency peak around 0.1 Hz disappearing (Figs. 1d, e).

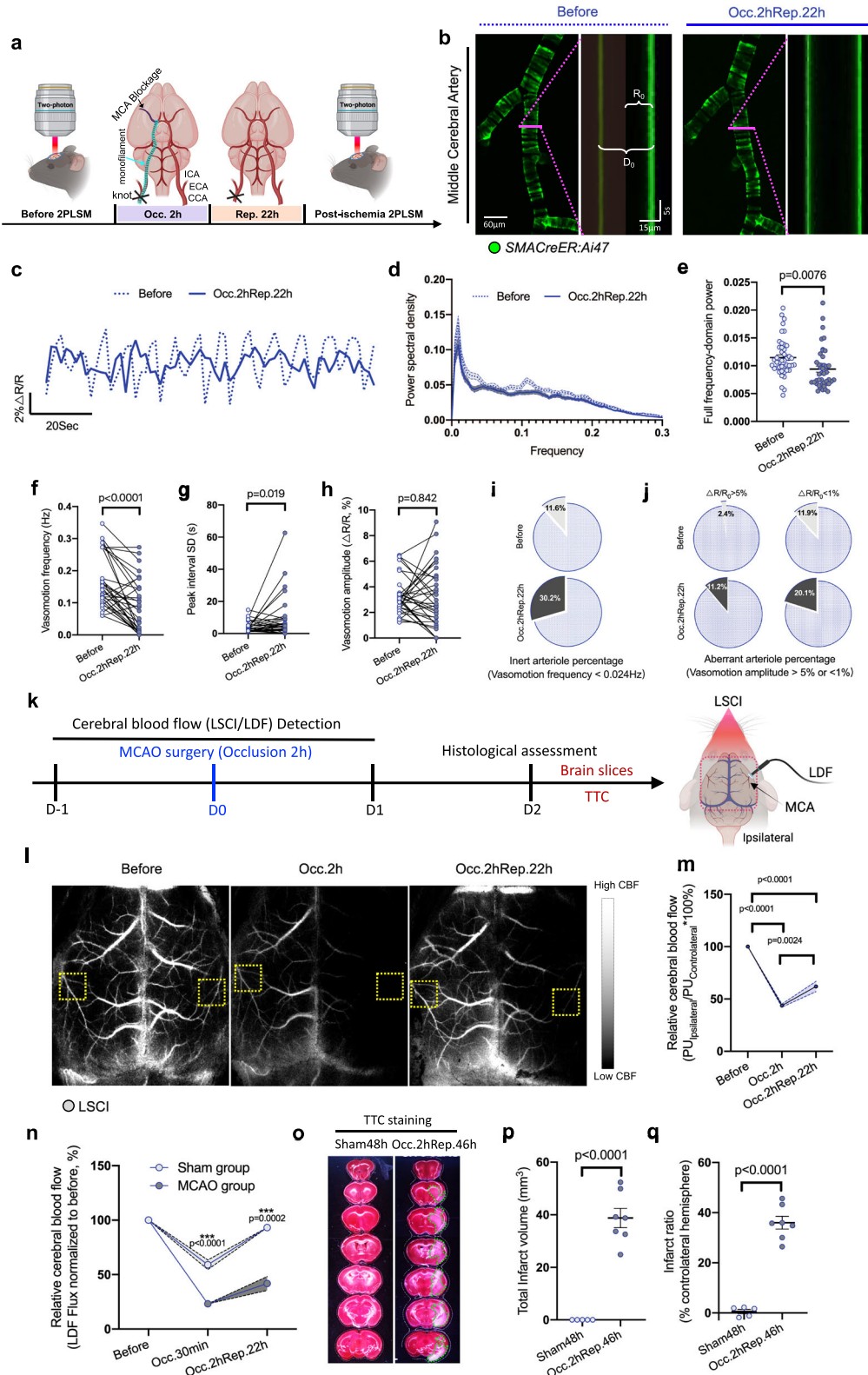

Furthermore, we performed strict matched-pair comparisons for the vasomotion index analyses. After examining about 33 arterioles from 7 mice, we found the vasomotion frequency had decreased significantly from the before $0.168 \pm 0.0143$ Hz to the post-ischemic $0.0889 \pm 0.0142$ Hz (Fig. 1f), and the peak interval SD had increased from the before $3.993 \pm 0.561$ s to the post-ischemic $9.816 \pm 2.419$ s (Fig. 1g), confirming the existence of the post-ischemic injury in the arteriolar myogenic spontaneous

vasomotion. While the averaged changes in vasomotion amplitude were comparable, in post-ischemic group, most oscillatory amplitude peaks were polarized (Fig. 1h). We used the venular vasomotion frequency (0.0249 Hz) as a cutoff threshold to define 'inert arteriole' in temporal dimension. We found that, before stroke, although 88.7% of arterioles were active, a small portion of arterioles (11.3%) were constitutionally inert. Notably, this percentage increased by threefold post-ischemia, suggesting a irregular

**Fig. 1 | Stroke evokes long-lasting damage in myogenic spontaneous vasomotion, correlating with the no-reflow time window. a** Experimental design of the myogenic spontaneous vasomotion detection under pre-ischemic and post-ischemic (Occ.2hRep.22 h) conditions. **b** Representative maximum intensity projection (MIP) images and kymographs of MCA before and after (Occ.2hRep.22 h) ischemic stroke in *SMACreER:Ai47* mouse under 2PLSM. $D_0$ represents the basal arteriolar diameter, and $R_0$ for arteriolar radius. The magenta solid line represents the resliced position. **c** Representative time-lapse radius changes trace of arteriole before (dash line) and after (solid line) (Occ.2hRep.22 h) ischemic stroke. **d** Fourier transform analysis of the rhythmic fluctuations in the arterioles before and after (Occ.2hRep.22 h) ischemic stroke ($N = 4$ mice). **e** Statistical analysis of accumulated power (AUC, area under the curve) of vasomotion within the frequency range of 0–0.3 Hz before and after (Occ.2hRep.22 h) ischemic stroke ($N = 4$ mice). Paired vasomotion index analysis of the MCA before and after (Occ.2hRep.22 h) ischemic stroke ($N = 7$ mice, $n = 33$ vessels), including frequency of rhythmic cycles (**f**), standard deviation (SD) of the peak intervals (**g**), and amplitude of changes in the vascular radius (**h**). **i** Pie chart representing the percentage of 'inert arterioles' before and after (Occ.2hRep.22 h) ischemic stroke in a paired measurement ($N = 7$ mice,

$n = 46$ vessels), which oscillates with a frequency less than venular vasomotion (0.026 Hz). **j** Pie chart representing the percentage of 'aberrant arterioles' before and after (Occ.2hRep.22 h) ischemic stroke ($N = 7$ mice, $n = 46$ vessels). **k** Scheme of timepoints for MCAO surgery, CBF detection using LDF and LSCI, and histological assessment using TTC staining, in the short-term (2 days) of ischemic-induced neuronal injury evaluation assay. **l** LSCI images of the mouse whole brain indicate the time course changes in the cerebral blood flow (CBF) before, during (Occ.2 h), and after (Occ.2hRep.22 h) ischemic stroke. **m** Statistical analysis of the relative CBF changes (LSCI method) between the contralateral and ipsilateral at different time points during ischemic stroke ($N = 5$ mice). All measurements were normalized to the basal CBF before ischemic stroke. **n** Statistical analysis of the relative CBF changes (LDF method) at different time points before and after MCAO or sham surgery ($N = 4$ mice for each group). **o** TTC staining for mouse brain after 2 days of MCAO or sham surgery ($N = 5$ or 7). **p–q** Statistical analysis of the total infarct volume and the infarct ratio (% contralateral hemisphere) using TTC staining. Data are expressed as the mean ± SEM. Statistic for (**e**, **p**, **q**), data were analyzed using unpaired *t* tests. Statistics for (**f**, **g**, **h**), paired *t* tests were used. Otherwise, data were analyzed using one-way ANOVA, followed by Tukey's post hoc analysis.

vasomotion status (Fig. 1i). We also defined 'aberrant arteriole' in spatial dimension with the cutoff threshold of vasomotion amplitude >5% or <1%. We found more aberrant events in post-ischemic arterioles in both the largest category (11.2% vs. 2.4% in amplitude >5%) and the smallest category (20.1% vs. 11.9% in amplitude <1%) than those detected from the same arterioles before ischemic stroke (Fig. 1j), indicating an extreme distributed pattern of arteriolar myogenic vasomotion after ischemic challenges.

Next, we monitored the cerebral circulation using both laser speckle contrast imaging (LSCI) and laser Doppler flowmetry (LDF) system, to explore how cerebral blood flow changes following transient ischemia and vasomotion recession in our MCAO model (Fig. 1k). As expected, in MCAO mice, LSCI data had revealed remarkable ipsilateral reductions (43.70 ± 1.454%) in cerebral perfusion at the end of 2 h occlusion, as for the time point of 2 h occlusion and 22 h reperfusion, the relative CBF had dropped to 61.89 ± 4.874% compared to before conditions (Fig. 1l, m). Moreover, by using LDF, we stably detected no-reflow on the ipsilateral hemisphere at the time point of 2 h occlusion and 22 h reperfusion as the relative CBF had dropped to 41.79 ± 6.351%, while no significant ipsilateral CBF deccending had been detected by LDF in sham-surgery groups (93.18 ± 2.544%) (Fig. 1n). The neurological deficits were accessed by TTC staining after 2 days of surgery, the infarct volume (38.76 ± 3.673 mm³) and the infarct ratio (36.00 ± 2.555%) in MCAO group mice were significant higher than sham group mice, indicating a successful MCAO surgery (Fig. 1o, p, q).

To exclude the influence of the permanent ligation of the unilateral common carotid artery (CCA) toward myogenic spontaneous vasomotion after MCAO surgery, we imaged the MCA motility under 2PLSM before and post-sham surgery (Fig. S3a, S3b). Vasomotion index analyses showed that in the parameters including vasomotion frequency, peak interval SD and vasomotion amplitude, all values were comparable before and post-sham surgery, in arteriole, PA and venule (Fig. S3c, S3d, S3e). Last but not least, we checked the global cerebral blood flow (CBF) changes using LSCI system. We found the ipsilateral relative CBF decreased shortly (65.26% to 70.05% in 2 h) after CCA ligation, and recovered to 80.55 ± 3.160% after 24 h of sham surgery (Fig. S3f, S3g). Taken together, the post-ischemic no-reflow time point are properly corresponded to the time point of cerebral spontaneous vasomotion decline, these results demonstrated a possible association between arteriolar myogenic spontaneous vasomotion impairments and the no-reflow phenomenon.

## Post-ischemic injury disrupts cerebral circulation independent on the anisotropic arteriolar diameter changes

After finding the arteriolar myogenic spontaneous vasomotion was impaired during no-reflow, we wonder how cerebral blood flow in

individual vessels was affected during no-reflow. Blood flow in arterioles was shown by fluorescent DiO-labeled blood cells, and the capillary flow was shown by water-soluble rhodamine B-dextran, through intravenous injection in vivo (Fig. 2a). Repeated line scanning of the same individual vessels before and after stroke (Occ.2hRep.22 h) revealed reduced blood flow speed and flux at both the arteriolar and capillary levels, in line with the cerebral global CBF reduction in LSCI/LDF results. Reversed blood flow in arterioles occasionally occurred during the initial 10 min and more often during 1 to 2 h of occlusion (Fig. 2b, c). We found that the blood cell flow speed in arterioles in post-ischemia period (13.43 ± 0.75 mm/s) was reduced to 47.57% of that before ischemic stroke (6.39 ± 0.55 mm/s) (Fig. 2d). In addition to arterioles, reductions in capillary flow level profoundly contribute to no-reflow. We examined 39 capillaries in 7 mice with repeated imaging before and after (Occ.2hRep.22 h) stroke (Fig. 2e), quantitative analyses showed similar prolonged reductions in blood cell velocity (60.03%) and flow (68.37%) (Fig. 2f, g).

Given that capillary blocks have been frequently reported in ischemic stroke[37,38], we wonder how capillary obstruction participated in our model. Interestingly, we found there are few of capillaries work dynamically, as 4.33% of the capillaries were stalled tempeorarilly under our inclusion criterion of stalled capillary counting (10.84 s /10 frames no flowing during 218.81 s/200 frames, see methods) in naive mice. Consistent with previous reports[39,40], our data exhibited an increasing trend of capillary stalls during occlusion, in contrast, the functional capillary number at 22 h reperfusion period recovered to a level comparable to that before stroke (Fig. S4a, S4b). There is 4.92% of increasement of the capillary stall rate at the time point Occ.2hRep.22 h compare to before ischemia, implying microvascular obstruction do exist but limited in the MCAO model of transient ischemic stroke. These results strongly implied that particularly at the time point of 2 h occlusion and 22 h reperfusion, no-reflow was not associated with the number of nonfunctional capillaries but with the number of low-functional capillaries.

Next, since how arteriolar diameter changes before and after stroke remains controversial which may be due to different experimental settings[37,41–43], we explored whether arteriolar diameter changes are associated with no-reflow. We specifically targeted MCA branching arterioles in the pia with 20–60 μm diameters. Our results revealed anisotropic changes in diameter in response to ischemic insults, including three circumstances: bidirectional changes and no changes (Fig. 2h). For statistics, we found there is no significant difference between absolute arteriolar diameter before and after (Occ.2hRep.22 h) ischemic stroke (Fig. 2i), and the change rate of the diameter is 100.2 ± 1.1% (Fig. 2j), which suggested no changes. Furthermore, we also checked brain vasculature density in infarct core area during ischemia by using *Cdh5CreER:Ai47* mice, in which endothelial cells were labeled with EGFP, we confirmed that there was no obvious vascular density

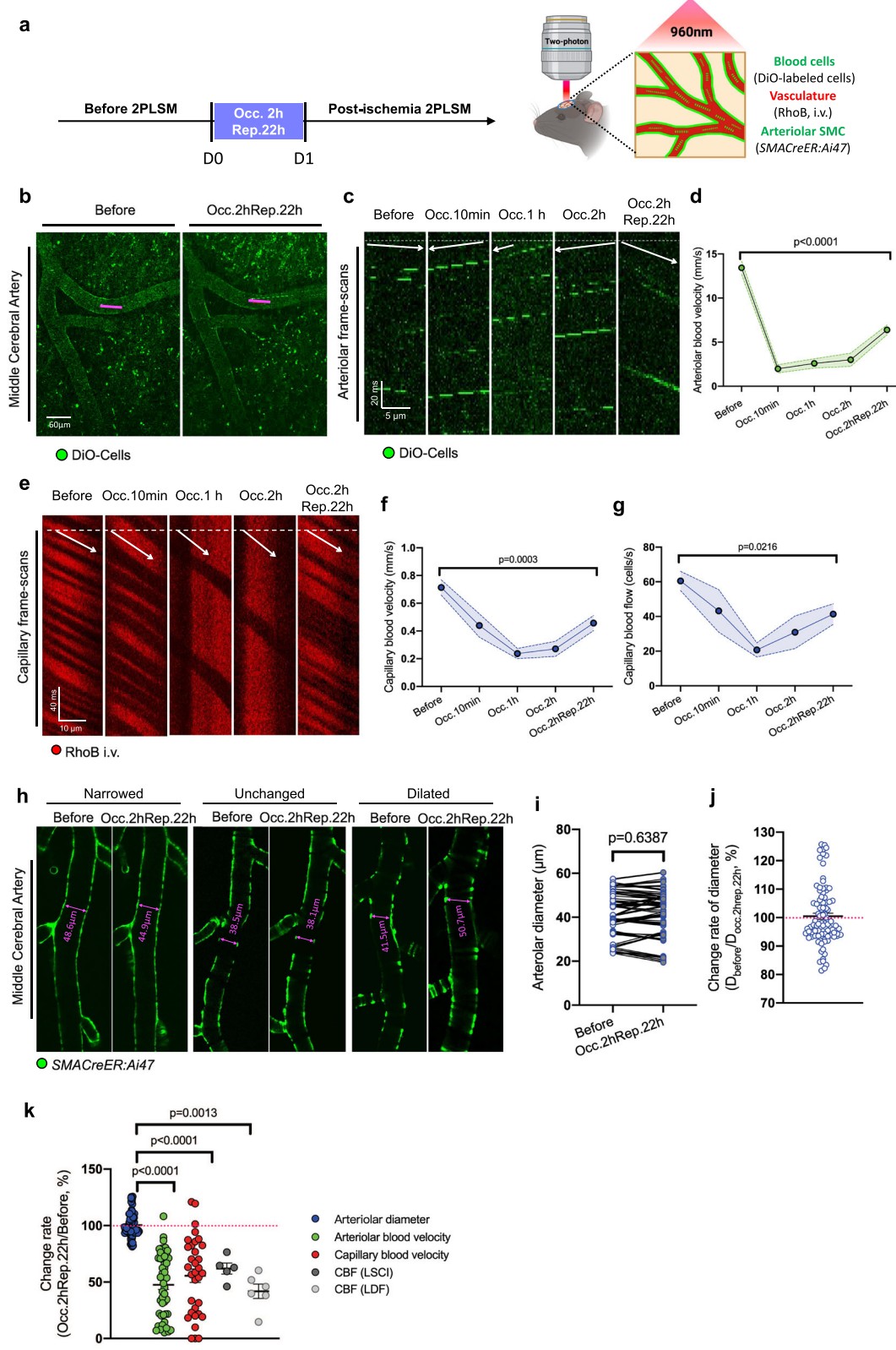

loss at 2 h occlusion and 22 h reperfusion when no-reflow was fully established (Fig. S5a, S5b). Our results are consistent with the published data that endothelial death was not observed after ischemia at this time point[40]. The brain surface arteriolar density was checked by using *SMACreER:Ai47* mice, still no obvious vascular density loss was found (Fig. S5c, S5d).

Pooling together all the parameters presented thus far (Fig. 2k), our data demonstrated that depressed cerebral arteriolar and capillary blood flow are correlated with post-ischemic CBF reduction, but not arteriolar diameter changes. In other words, no-reflow is unrelated with changes in resting arteriolar diameters. Regarding to the close tie up between cerebral

**Fig. 2 | Post-ischemic injury disrupts cerebral circulation independent on the anisotropic arteriolar diameter changes. a** Scheme of timepoints for MCAO surgery, 2PLSM and experimental design. **b** Representative frame-scan images of DiO-labeled blood cell trajectories in MCA before and after (Occ.2hRep.22 h) ischemic stroke. The magenta solid line indicates the line-scan area in (**c**). **c** Representative line-scan images of DiO-labeled blood cell trajectories in MCA before and after (Occ.2hRep.22 h) ischemic stroke. The white arrow indicates the flow direction, which was reversed during occlusion and recovered after reperfusion compared with before ischemia. **d** Statistical analysis of the arteriolar blood flow velocity (mm/s) in MCA before and after ischemic (Occ.2hRep.22 h) stroke ($N = 4$, $n = 48$ vessels). The blood flow velocity was analyzed by calculating the slope of the DiO-labeled blood cell trajectory in the kymographs in (**c**). **e** Representative line-scan images of RhoB-labeled capillary blood flow before and after (Occ.2hRep.22 h) ischemic stroke. The shaded streaks indicate the RBC motion trajectory, and the red streaks indicate the tracer-filled capillary lumen. The white arrow indicates the flow direction. **f, g** Capillary blood velocity and blood flow changes curve during ischemic stroke, data were analyzed by calculating the streaks slope and the cell number in the kymographs ($N = 7$ mice, $n = 39$ vessels). **h** Representative images of MCA changes before and after (Occ.2hRep.22 h) ischemic stroke. Examples of constricted, unchanged, and dilated arterioles are displayed. **i, j** Statistical analysis of paired arteriolar diameter (**h**) and diameter change rate (**i**) before and after (Occ.2hRep.22 h) ischemic stroke ($N = 6$ mice, $n = 90$ vessels). **k** Change rate analyses for arteriolar diameter, arteriolar blood velocity, capillary blood velocity, and CBF (including LSCI and LDF) changes before and after (Occ.2hRep.22 h) ischemic stroke. All data were normalized to the basal value before ischemic stroke. Data are expressed as the mean ± SEM. Statistic for (**h**), paired $t$ tests were used. Otherwise, data were analyzed using one-way ANOVA, followed by Tukey's post hoc analysis.

circulation interruption and vasomotion impairment in the time window of no-reflow, thus, we are moving to figure out the pathogenesis of arteriolar myogenic injury after ischemia.

## SMC Ca²⁺ dynamics are prolongedly inhibited by transient ischemia

To decode the mystery of myogenic spontaneous vasomotion recession post-ischemia, we hypothesized that cytosolic Ca²⁺ oscillation may play an important role, as it has been extensively shown that IP3R-dependent cyclic Ca²⁺ release from the endoplasmic reticulum serves as a pacemaker for SMC spontaneous contraction activity[44–46]. Therefore, we used *SMACreER:Ai96* mice that specifically express the genetically encoded calcium indicator GCaMP6s in SMCs to examine how SMC Ca²⁺ dynamics respond to ischemic stroke (Fig. 3a). Notably, the calcium oscillation can be observed in the SMCs of the MCA, but attenuated significantly after 2 h occlusion and 22 h reperfusion of ischemic stroke (Figs. 3b, c) (Movie 3). Besides, stroke remarkably diminished basal cytoplasmic Ca²⁺ levels by 63.1% relatively, through a paired statistical analysis in arteriolar SMCs (Fig. 3d). Fourier transform analyses revealed that Ca²⁺ oscillation after stroke exhibited significantly lower power in the frequency dimension than that recorded before stroke, with the frequency peak around 0.1 Hz disappearing (Figs. 3e, f). Furthermore, we performed strict matched-pair comparisons for the calcium index analyses. After examining about 41 SMCs from 5 mice, we found the Ca²⁺ oscillation frequency had decreased significantly from the before $0.1021 \pm 0.0052$ Hz to the post-ischemic $0.04559 \pm 0.0045$ Hz (Fig. 3g), the peak interval SD had increased from the before $5.442 \pm 0.402$ s to the post-ischemic $10.60 \pm 1.109$ s (Fig. 3h), and the Ca²⁺ oscillation amplitude had decreased significantly from the before $23.64 \pm 2.219\%$ to the post-ischemic $14.10 \pm 1.752\%$ (Fig. 3i).

In order to validate the methodology of Ca²⁺ signals monitoring in vivo, we first checked the funtional effectiveness of the calcium sensor in post-ischemic conditions. The concentrated KCl solution is widely used to mimic cortical spreading depolarization through producing breakdown of ion gradients[47]. In this model, spreading depolarization shift is normally followed by a drastic vasoconstriction[47]. To verify whether the calcium sensor GCaMP6s could indicate calcium levels in SMCs after ischemic stroke, we have performed additional calcium imaging experiments in the post-ischemic arterioles under KCl-triggered cortical spreading depolarization conditions(Fig. 3j). Interestingly, we found that 300 mM KCl treatment through the administration hole could evoke a remarkable growth of Ca²⁺ signals in arteriolar SMCs, where the calcium oscillation was once suppressed by ischemic insults (Fig. 3k, l). In this way, the supression of calcium oscillation in post-ischemic SMCs are excluded from the possibility of the GCaMP6s calcium sensor dysfunction. Further more, We also excluded the possibility of SMC death or migration after stroke, as we had checked the SMC integrity in *SMACreER:Ai47* mice, where the retaining of the EGFP signal could manifest the membrane integrity and the live state of SMCs. Statistical analysis of the SMC density along the arterioles had shown no significant difference between before and after stroke (Fig. S6a, S6b). Consequently, our findings demonstrated that the SMC Ca²⁺ oscillation

should be deeply involved in myogenic spontaneous vasomotion maintenance.

## Mitochondrial membrane potential (ΔΨm) transient loss could induce mitochondria-ER dissociation and Ca²⁺ oscillation recession in SMCs

It has been reported that arterial vasomotion is depdndent on a cytosolic oscillator involving the periodic release of internal calcium stores via IP3R from ER, coupled to rhythmic oscillations in membrane deplorization and eventually relied on the termporal oscillatory control over calcium-dependent of cross-bridge cycling[14]. Indeed, as shown in Fig. S8, 2-APB (IP3R inhibitor) administration could block the spontaneous calcium oscillation in SMCs significantly, suggesting IP3R-induced calcium release should be the main source of the calcium oscillations intracellularlly. Furthermore, Ca²⁺ shuttling between the ER and mitochondria may have a pacemaker role in the generation of Ca²⁺ oscillations[32]. In this way, to decode Ca²⁺ oscillation recession under ischemia, we particularly examined mitochondrial function in SMCs in vivo following ischemic insults. Before live imaging under 2PLSM, the ΔΨm probe TMRM, in combination with mitochondrial probe mitotracker Green as the reference, were applied through the cranial mouse window (Fig. 4a). To our surprise, we found mitochondrial membrane potential (ΔΨm) loss gradually after ischemia onset but not for the reference dye (Fig. 4b, c). Moreover, ex vivo TMRM staining of the whole brain acutely dissected from post-ischemic mice (Fig. S7a) showed that ΔΨm in ipsilateral SMCs remained at low levels despite successful reperfusion for 1 hour, but recovered to the normal level at 22 h post reperfusion (Fig. S7b–S7e). A summary graph clearly showed a transient ΔΨm loss trajectory before and along stroke progression (Fig. S7f). Therefore, we concluded that ΔΨm was sufficient but unnecessary for Ca²⁺ homeostasis maintenance.

As the interaction between mitochondria and ER are potential critical for Ca²⁺ transportation and Ca²⁺ oscillation[29,31], we next examined whether ΔΨm deprivation was sufficient to deteriorate the mitochondria-ER contact ultrastructure using correlative light and electron microscopy (CLEM) technique (Fig. 4d). In cultured primary SMCs (identified by tdT positive, see method), quantitative analyses of the fraction of mitochondrial perimeter covered by ER revealed that 10 min treatment with CCCP, a protonophore and uncoupler, could not only prominently reduce the mitochondria-ER contact fraction from $42.19 \pm 4.145\%$ to $10.98 \pm 1.975\%$ (Fig. 4e), but also dramatically changed the tubular morphology into spherical mitochondria without affecting mitochondrion areas (Fig. 4f). In addition, the average distance between mitochondria and the ER expanded by 1.72-fold (Fig. 4g), and the average mitochondria-ER contact length reduced by 2.56-fold (Fig. 4h). These data indicated that mitochondria-ER contact can be affected by the loss of ΔΨm.

To manipulate the Ca²⁺ oscillations in SMCs, we sought to use an established molecular tool called the mitochondria-ER linker (ME-Linker)[33], through tethering the two organelles artificially. In cultured primary SMCs (tdT⁺), AAV-*ME-Linker* virus and AAV-control virus administrated 2 days before live imaging (Figs. 4i, j), and the cytosol calcium was

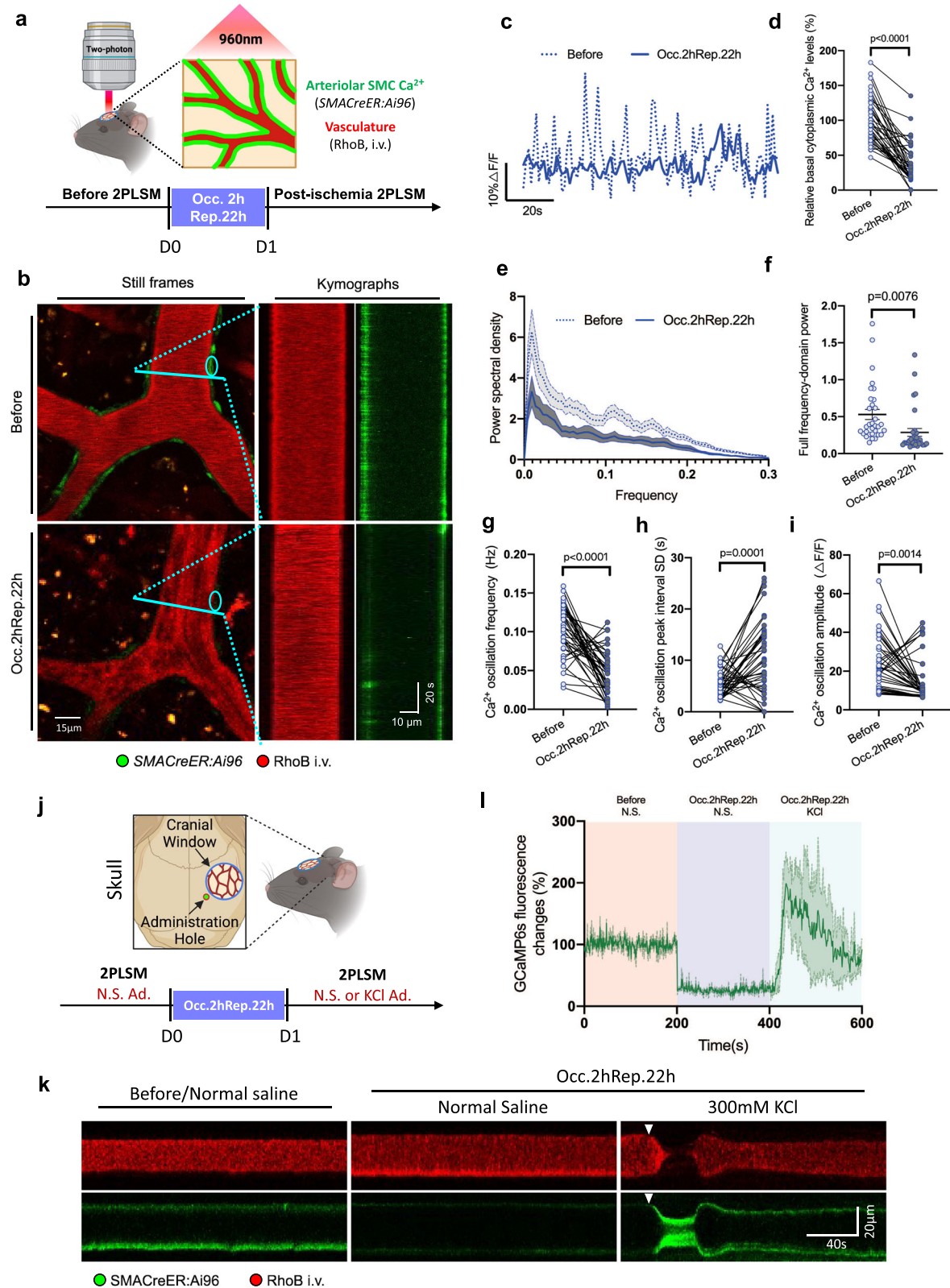

visualized through the $Ca^{2+}$ indicator YTnC2[48,49] during imaging (Fig. 4k) (Movie 4). Notably, in vehicle groups, ME-Linker not only enhanced the amplitude but also increased the frequency of $Ca^{2+}$ oscillation, demonstrating that artificial tethering of mitochondria and ER sufficiently promoted $Ca^{2+}$ oscillation under nonpathological conditions (Figs. 4l, m). Moreover, ME-Linker replenished $Ca^{2+}$ oscillation regarding amplitude and frequency in cells after CCCP treatment. In contrast, as expected, those transduced with the control virus lost $Ca^{2+}$ oscillation shortly after the CCCP treatment (Figs. 4k, l). ME-Linker overexpression in SMC also increased the percentage of high $Ca^{2+}$ oscillation frequency (>0.1 Hz) and the high $Ca^{2+}$ oscillation amplitude (>10%) either in vehicle or CCCP treatment conditions (Fig. 4n, o).

**Fig. 3 | SMC Ca²⁺ dynamics that underlie myogenic spontaneous vasomotion are prolongedly inhibited by transient ischemia. a** Scheme of timepoints for MCAO surgery, 2PLSM and experimental design. **b** Representative still-frame images and kymographs of the same MCA before and after ischemic (Occ.2hRep.22 h) stroke. Calcium oscillation in SMCs was reported using *SMACreER:Ai96* mice. The turquoise solid line represents the resliced position. **c** Representative time-lapse calcium oscillation trace of SMC before (dash line) and after (solid line) (Occ.2hRep.22 h) ischemic stroke. Target SMC was labeled in b with turquoise circle. **d** Basal cytoplasmic Ca²⁺ analyses in SMCs before and after ischemic (Occ.2hRep.22 h) stroke. All data were normalized to the basal value before ischemic stroke ($N = 5$ mice, $n = 40$ cells). **e** Fourier transform analysis of the rhythmic Ca²⁺ oscillation in the arterioles before and after (Occ.2hRep.22 h) ischemic stroke ($N = 3$ mice). **f** Statistical analysis of accumulated power (AUC, area under the curve) of Ca²⁺ oscillation within the frequency range of 0–0.3 Hz before and after (Occ.2hRep.22 h) ischemic stroke ($N = 3$ mice). Calcium index analysis of the Ca²⁺ oscillation in SMCs

before and after (Occ.2hRep.22 h) ischemic stroke ($N = 5$ mice, $n = 41$ cells), including frequency (**g**), SD of the peak intervals (**h**), and amplitude (**i**). **j** Scheme of the KCl (300 mM) administration strategy on the brain surface during post-ischemic period and experimental timeline of 2PLSM. **k** Kymographs of MCA before and after (Occ.2hRep.22 h) ischemic stroke under 2PLSM. Especially, high dose of KCl (300 mM) was administrated at the timepoint of Occ.2hRep.22 h, and a drastic KCl-indused spreading deporlization and vasoconstriction can be observed in these vasomotion/calcium oscillation depressed arterioles (under normal sline treatment conditions). The red signals are the intravenous injected RhoB dye and represents arteriolar lumen. The green signals in the vascular wall represents Ca²⁺ signals indicated by GCaMP6s. **l** Time-lapse fluorescent GcaMP6s signal changes trace in the MCA along with ischemic stroke, and high dose of KCl administration ($N = 3$ mice). Data are expressed as the mean ± SEM. All data were analyzed using unpaired *t* tests.

## Conditional overexpression of ME-Linker in SMCs restores mitochondria-ER contacts and Ca²⁺ homeostasis in response to ischemic stroke

Since the ME-Linker successfully promoted Ca²⁺ oscillation in vitro, we moved on to examine the ME-Linker function in vivo. We established a new mouse line, RCL (*ROSA26/CAG promoter/LoxP-STOP-LoxP*)-*ME-Linker* mice (Fig. 5a), and crossed it with the *SMACreER* line to produce *SMA-CreER:ME-Linker* mice to achieve SMC specific manipulation (Fig. S9). Regarding to the expression of ME-linker in peripheral vascular system in this double positive mouse, we first checked the heart function using echocardiology in isoflurane anesthetized mice. We found there is no structural or functional differences along comprehensive heart examination processes between *SMACreER:ME-Linker* mice and littermates (Fig. S10a–s). In addition, in awake mice, we also checked systolic blood pressure (Fig. S10t) and rectal temperature (Fig. S10u) in *SMACreER:ME-liner* mice before and after tamoxifen induction but found no differences. Furthermore, we found there is no significant difference between transgentic and control littermates in resting LSCI perfusion unit (Fig. S11a, 11b). These results suggest that ME-Linker does not interrupt both cardiovascular function systematically and the resting absolute cortical CBF values. Importantly, under physiological condition, we found ME-Linker overexpression in SMC in vivo could improve basal cytoplasmic calcium levels (Fig. S11c, S11d) and myogenic vasomotion through promoting vasomotion amplitude (Fig. S11e). Interestingly, there is no significant differences in vasomotion frequency and peak interval SD between *SMACreER:ME-Linker* mice and littermates (Fig. S11f, S11g), suggest that the myogenic vasomotion is relatively conserved in the frequency domain. ME-Linker overexpression in SMC is not sufficient to change the resting-state main MCA diameter (around 6 mm distal from MCA root under 2PLSM) (Fig. S11h, S11i), indicating there is no abnormal arteriolar constriction in *SMACreER:ME-Linker* mice and the calcium oscillation promotion by ME-Linker are limited within a homeostatic resting-state.

To assess weather ME-Linker could sustain mitochondria-ER contacts after ischemic challenges, we detected the ultrastructure of the SMCs from ipsilateral and contralateral MCA using transmission electron microscopy (TEM) (Fig. 5b). In control littermates, our data revealed that transient ischemia prolongedly reduced the fraction of mitochondrial perimeter covered by ER from 20.26 ± 1.481% to 10.79 ± 1.126% (Fig. 5c) and mitochondrial area from 0.2194 ± 0.02432 μm² to 0.1166 ± 0.01526 μm² (Fig. 5d). Importantly, the ME-Linker fully resisted (111.07%) the reduction in the fraction of mitochondrial perimeter covered by ER by stroke (Fig. 5c). In addition, the reduction in mitochondrial size was slightly rescued (75.09%) in *SMACreER:ME-Linker* mice (Fig. 5d).

We next directly examined Ca²⁺ features before and after stroke in *SMACreER:ME-Linker* mice, where GCaMP6s was co-expressed following the P2A cassette (Fig. 5a, e). In contrast to the robust reduction (−51.51 ± 6.040%) in *SMACreER:Ai96* mice (Fig. 3a, b), the relative basal Ca²⁺ level in SMC was intensely enhanced (40.17 ± 3.23%) in *SMA-CreER:ME-Linker* mice after ischemic stroke (Figs. 5f, g) (Movie 5). In

addition, the Ca²⁺ oscillation in SMCs was rescued after overexpression of the ME-Linker, as shown in the data that the change rate of calcium index was significant restored after 2 h occlusion and 22 h reperfusion comparing to *SMACreER:Ai96* mice (Fig. 5h, i, j). These results are consistent with the pervious in vitro studies, suggest that the mitochondria-ER tethering in restoring Ca²⁺ oscillation works sufficiently in vivo.

## Forced mitochondria-ER tethering rescues post-ischemic impairments in arteriolar myogenic spontaneous vasomotion and capillary perfusion

We next examined the effects of the ME-Linker in maintaining arteriolar wall dynamic features (Fig. 6a). Indeed, in *SMACreER:ME-Linker* mice, through 2PLSM live imaging, we observed obvious myogenic spontaneous vasomotion restoration after 2 h occlusion and 22 h reperfusion (Fig. 6b, c) (Movie 6). In Fourier transform analyses, in *SMACreER:ME-Linker* mice, we found that the frequency peak around 0.1 Hz was retained after ischemic challenges, and the power in the frequency dimension was comparable before and after ischemic stroke (Fig. 6d, e). Besides, in vasomotion index analyses, we found the 'inert arteriole' percentage had decreased to 20.5% (30.2% in wild-type mice) after ischemic stroke (Fig. 6f), and all vasomotion indexes were counteracted in *SMA-CreER:ME-Linker* mice compared with control littermates by showing fewer reductions in the change rate of frequency (Fig. 6g) and peak interval SD (Fig. 6h), and exhibiting increased amplitude change rate (Fig. 6i). In other words, the post-ischemic myogenic spontaneous vasomotion impairments, both the temporal (frequency) and the spatial (amplitude) aspects, are partially rescued through tethering mitochondria and ER in SMCs.

Myogenic spontaneous vasomotion consists of more than the temporal and spatial aspects, including a third aspect, synchronization activity between SMCs along the whole arteriolar segment[14]. It has been well-documented synchronization activity occurred on a scale of millimeter (a macro view) and observed in the arterioles of isolated peripheral tissue ex vivo[50–52]. However, this parameter has not been reported systematically yet in vivo, especially in live cerebral arterioles. Due to the field-of-view limitation and the natural curvature degree of brain vessels, we investigate synchronization activity on pial arterioles at hundred micrometers scale. We developed a 'cooperation index' to characterize the micro view of the synchronization activity between neighboring SMCs. We profiled the degree of similarity (Pearson correlation coefficient analysis) of the time-course arteriolar radius changes between any two sites crossing 70 μm-long arteriole (2.5 μm interval between each adjacent checkpoint, see Methods), every single site was compared with its sequential neighboring sites from proximal to distal. Notably, we found that the cooperation index value was attenuated in post-ischemic condition in control littermates, if a high cooperation level was defined by R value ≥ 0.3, we found that in control littermates, SMCs collectively cooperated with an extent of 17.5-27.5 μm but significantly reduced to 2.5–7.5 μm after ischemic stroke (Fig. 6j, k). What's more, the cooperation index was sufficiently enhanced in normal

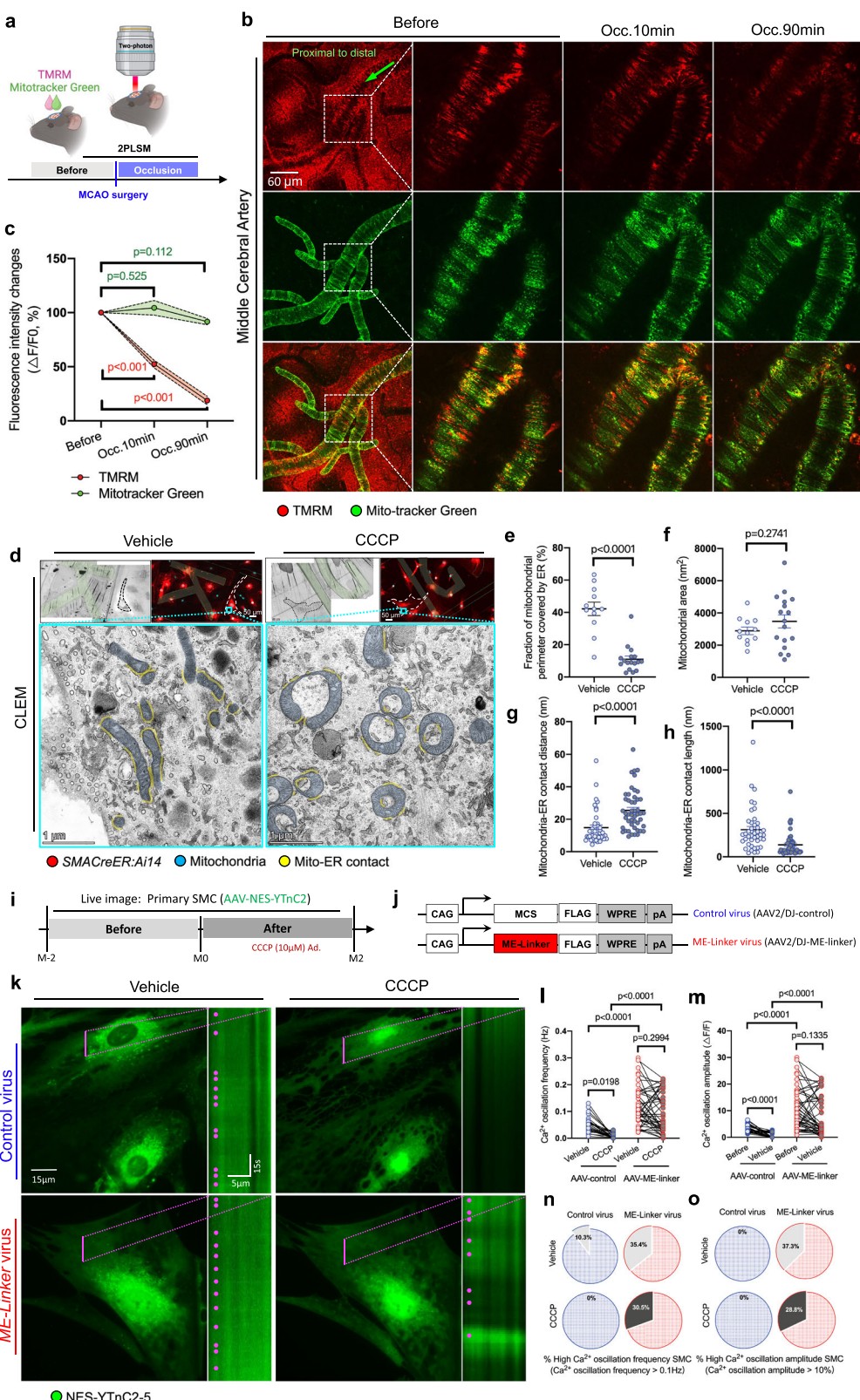

*SMACreER:ME-linker* mice compared to that in control littermates, and manifested a robustly higher values than that in control littermates under post-ischemic conditions. In *SMACreER:ME-Linker* mice, the cooperated arteriolar segments spanned around 27.5-30 μm, remarkably, the cooperated arteriolar segments post-ischemia was increased to around 50 μm (Figs. 6j and 6k). We do not know the intercellular mechanism of ME-Linker to manipulate cooperation index yet, if there is any genetic level modification between adjacent ME-Linker overexpressed SMCs. Nevertheless, we can decipher these facts through the individual SMC perspective, which should be the enhancement of the $Ca^{2+}$ oscillation and vasomotion amplitude in ME-Linker overexpressed SMCs, ultimately strengthening the contractility of the myogenic vasomotion.

**Fig. 4 | Mitochondrial membrane potential transient (ΔΨm) loss could induce instant Mitochondria-ER dissociation, which was associated with Ca²⁺ oscillation maintenance in SMCs. a** Experimental design of the in vivo SMC ΔΨm detection under 2PLSM. Both the probe of TMRM and Mitotracker Green were loaded through the cranial window before imaging. **b** Representative images of MCA labeling with TMRM (red) and Mitotracker Green (green) through the carinal window before and after ischemic stroke under 2PLSM in vivo. Green arrowheads represent blood flow direction. **c** Statistical analysis of the relative TMRM and Mitotracker Green levels in SMCs of the MCA in the time points before, 10 min and 90 min after occlusion (*N* = 3 mice, *n* = 6 vessels). **d** (Top) Bright-field and fluorescent images of the CLEM assay in the vehicle and CCCP treatment group. *SMACreER:Ai14* double-positive cells (red) were used as the indicator of SMCs in primary cultures. (Bottom) Representative TME images of subcellular ultrastructure in SMCs of the vehicle and CCCP treatment group, mitochondria (blue) and the mitochondrial-ER contact areas (yellow) were labeled with the pseudo-color. Statistical analysis of the Mito-ME contact and mitochondrial morphology parameters in the comparison of the vehicle and CCCP treatment groups (*N* = 12 or 17 views), including fraction of mitochondrial perimeter covered by ER (**e**), mitochondrial area (**f**), Mitochondria-ER contact distance (**g**) and Mitochondria-ER contact length (**h**). **i** Experimental design of the in vitro live imaging and drug treatment. **j** Scheme showing the AAV constructs utilized to express ME-Linker in primary SMCs. Control AAV constructs was the mock-vector with an intact multiple cloning site (MCS). **k** Still-frame images and kymographs of primary-culture SMCs before and after CCCP treatment in the control virus and *ME-Linker* virus groups. The calcium signal was indicated by NES-YTnC2. The magenta dots in kymographs indicate the calcium oscillation peaks in the ROI area (magenta line). Statistical analysis of the calcium oscillation frequency (**l**) and calcium oscillation amplitude (**m**) in control virus and *ME-Linker* virus groups before and after CCCP treatment (*N* = 3 assays, *n* = 30 or 51 cells). Percentage of the high calcium oscillation frequency (>0.1 Hz) (**n**) and high calcium transient amplitude (>10%) (**o**) in the control virus and *ME-Linker* virus groups before and after CCCP treatment (*N* = 3 assays, *n* = 30 or 51 cells). Data are expressed as the mean ± SEM. Statistic for (**e**, **f**, **g**, **h**), data were analyzed using unpaired *t* tests. Statistic for (**c**, **l**, **m**), data were analyzed using one-way ANOVA, followed by Tukey's post hoc analysis.

Following myogenic spontaneous vasomotion promotion, the capillary blood flow had also been improved after ischemic stroke in *SMA-CreER:ME-Linker* mice. Under 2PLSM, we explored the capillary blood flow patterns after the injection of the red fluorescent dye rhodamine B-dextran intravenously in control littermates and *SMACreER:ME-Linker* mice. As shown in the frame-scans and line-scans (Figs. 6l, m), the capillary blood velocity and flow were significantly restored, especially during reperfusion period either 20 min or 22 h, in *SMACreER:ME-Linker* mice (Figs. 6n, o). The capillary stall rate was comparable between two genotypic mice (Fig. 6p), suggested that the biological function of ME-Linker was independent of the formation and elimination of capillary obstructions during ischemic stroke. To strengthen this conclusion, we also performed rt-PA (10 mg/kg, Actilyse, Boehringer Ingelheim) and neutrophil depletion administration (Ly6G antibody injection, see method) along ischemic stroke in our experimental setup (Fig. S13a). Interestingly, through hematology analysis in peripheral blood, we do not find significant differences in neutrophil proporation and absolute number between *SMACreER:ME-Linker* mice and littermates physiologically (Fig. S13b–S13d), in contrast, ischemic stroke could promote neutrophil proliferation (5.21-fold in control mice) in 22 h extremely (Fig. S13e–S13g). Moreover, we found that half neutrophil depletion do not help in restoring no-reflow in both control mice and *SMACreER:ME-Linker* mice in MCAO ischemic model (Fig. S13h, S13i), espically at the timepoint of 22 h after occlusion.

## Arteriolar myogenic spontaneous vasomotion improvement are sufficient to replenish global cerebral circulation and attenuates brain atrophy after ischemic stroke

To understand the neuroprotective effect in restoration spontaneous myogenic vasomotion, we explored whether *SMACreER:ME-Linker* mice could alleviate global cerebral no-reflow and its associated neurological injuries. Firstly, we checked the neuronal survival threshold to ischemic injury, the oxygen and glucose deprivation (OGD), in the acute brain slices of both *SMACreER:ME-Linker* mice and littermates (Fig. S12a). In this assay, the membrane impermeant dye propidium iodide (PI) was used to exclude from viable cells. We found that not only in the all brain cells (hoechst33342 positve), but also in the neurons (Nissl positive), 1 hour OGD insult could induce comparable cell death increment in *SMA-CreER:ME-Linker* mice and littermates (Fig. S12b–S12e). The neuron percentage in cerebral cortex was also not changes between two genotypes (Fig. S12f, S12g). Based on this fact the anti-ischemia proterity of neurons was unchanged, we then tracked the CBF changes by LDF and short-term ischemic injury by TTC staining in live mice under ischemic stroke (Fig. 7a). Strikingly, we found the no-refow was fairly rescued (Fig. 7b), followed by the significant decrease of the total infarct volume and infarct ratio by TTC at the time point of two days after occlusion, in *SMACreER:ME-Linker* mice comparing to control mice (Figs. 7c–e).

Furthermore, by using LSCI, repeated measurements of CBF at multiple time points after ischemia (Fig. 7f) revealed that the hypoperfusion in control littermates could last for two weeks, however, the *SMACreER:ME-Linker* mice could reach thorough reperfusion more quickly and robustly (Figs. 7g, h). What's more, we observed significantly less brain atrophy in *SMACreER:ME-Linker* mice than in control littermates (Fig. 7i), as the rate of cortical hemisphere atrophy had decreased from 15.39 ± 1.63% to 7.66 ± 2.92% (Fig. 7j). Fluoro-Jade C (FJC) staining at the level of brain sections further revealed a significant reduction in neuronal death in *SMACreER:ME-Linker* mice (Fig. 7k), the neurodegenerative area had decreased from 19.00 ± 1.03% to 8.11 ± 0.54% with statistically significant differences (Fig. 7l). To increase the robustness of the behavioral and histological assessment, meanwhile, we checked the neurological severity scores (NSS) along ischemic stroke over time, the the expression of neuronal marker Map2 and NeuN and the fluorescent Nissl morphology in the serial post-ischemic brain slices (Fig. S14a). We found the behavior imparments, including both motor and sensory inactivity, can be alleviated in *SMA-CreER:ME-Linker* mice (Fig. S14b, S14c). In the long-term (two weeks) ischemic injury examination of the ipsilateral atrophied brains (Fig. S14d, S14e), consistent with the FJC staning resluts, we found that the neuronal injury (including Map2/NeuN loss and Nissl breakage) volume was declined significantly in *SMACreER:ME-Linker* mice (Fig. S14f–S14k), indicates a positive role in post-ischemic neuroprotection.

For the overall evaluation of the living quality, we recorded the body weight changes and the survival rate along the time course of before and after stroke for up to 12 days of reperfusion in control littermates and *SMACreER:ME-Linker* mice. We found that the body weight was lower in *SMACreER:ME-Linker* mice at the early phase (5th day) of reperfusion but recovered faster once the 5th day was passed, comparing to control littermates (Fig. 7m). There was no significant difference in the percent survival, comparing between control littermates and *SMACreER:ME-Linker* mice (Fig. 7n).

Above all, we conclude that intrinsic SMC rhythmic contractility is critical for maintaining adequate capillary blood flow and preventing no-reflow after ischemic stroke. The neuronal protection function of ME-linker overexpression in SMCs suggest that the strategy of myogenic spontaneous vasomotion repairment could achieve the no-reflow injury relief, through alleviating hypoperfusion along post-ischemia.

## Discussion

This study systematically and quantitatively characterized progressive changes in cerebral perfusion and arteriolar wall motility pathology under stroke, and identified vascular paralysis with diminished vasomotion as the key factor that led to post-ischemic no-reflow (Fig. 8). Our findings demonstrated that transient SMC ΔΨm loss during ischemia leads to a permanent impairment in the mitochondria-ER contact

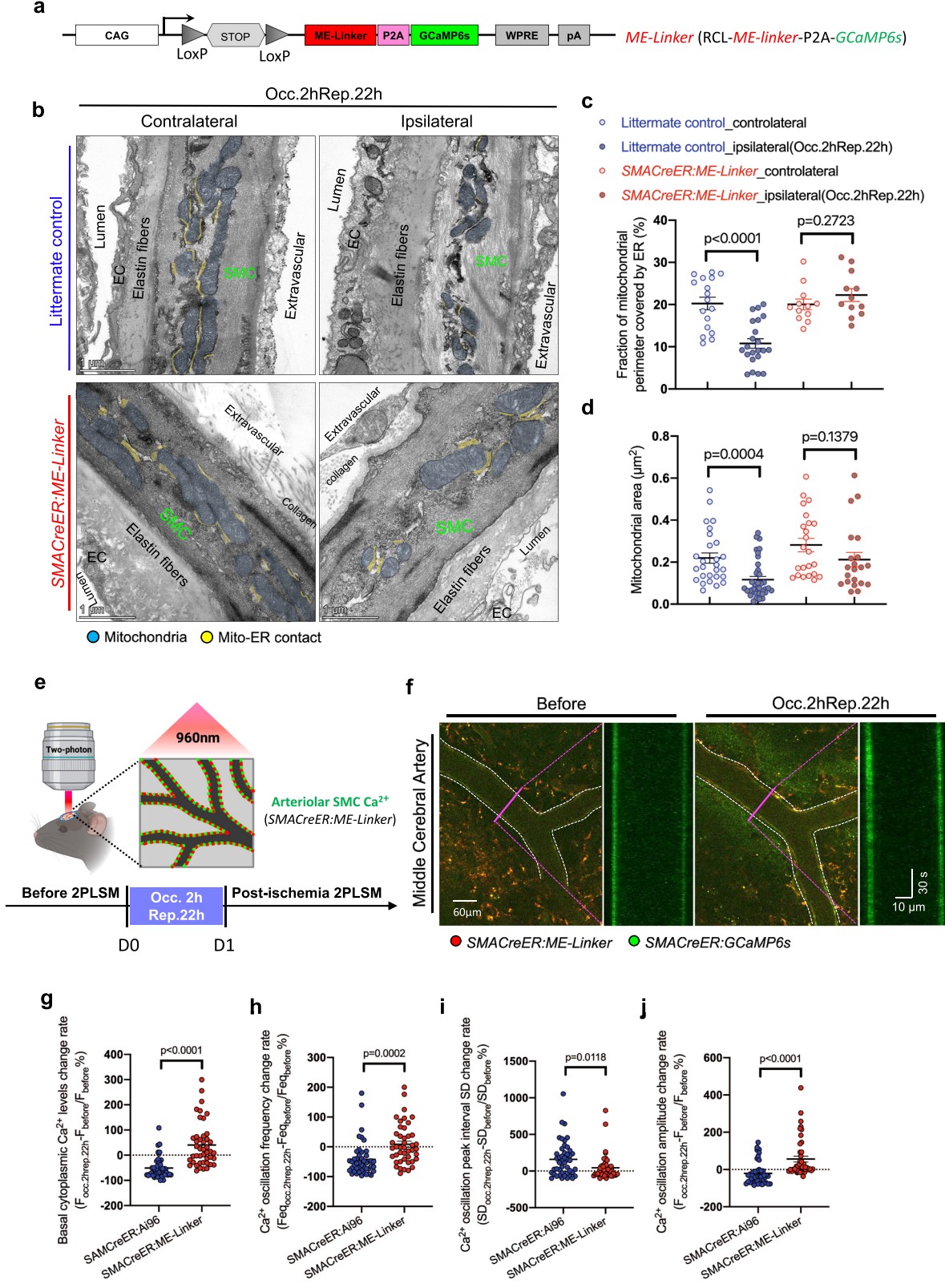

structures, which irreversibly deteriorates the intracellular $Ca^{2+}$ oscillation and eventually injures SMC contractility prolongedly. Importantly, our data indicate that increased vasomotion patency of upstream arterioles by strengthening the mitochondria-ER contacts in SMCs prevents no-reflow, achieving thorough reperfusion, and prevents the ischemic brain from atrophy.

While the vascular spontaneous oscillation occurring at the frequencies below the rate of respiration had been observed experimentally as early as in 1853 in bat wings[53], the direct physiological relevance of vasomotion remains vague. Indeed, as it is shown in our data from anesthetized mice, the oscillatory amplitude of the cerebral arteriolar radius is $3.263 \pm 0.251\%$ (Fig. 1h), which is relative slight but robust, and could only be captured

**Fig. 5 | Conditional overexpression of ME-Linker in SMCs restores Mitochondria-ER contacts and Ca²⁺ homeostasis in response to ischemic stroke.** **a** Configuration of ME-Linker tandemly linked GcaMP6s-expressing ROSA26 reporter line. **b** Representative TME images of SMC subcellular ultrastructure of the contralateral and ipsilateral (Occ.2hRep.22 h) MCA in control littermate and *SMACreER:ME-Linker* mouse. SMCs were charactered by the robust myofilaments in cytoplasm, and mitochondria (blue) and the mitochondrial-ER contact areas (yellow) were labeled with the pseudo-color. **c** Statistical analysis of the fraction of mitochondrial perimeter covered by ER in the comparison of the contralateral and ipsilateral (Occ.2hRep.22 h) SMCs control littermate and *SMACreER:ME-Linker* mouse (N = 12–22 views). **d** Statistical analysis of the fraction of mitochondrial area in the comparison of the contralateral and ipsilateral (Occ.2hRep.22 h) SMCs control littermate and *SMACreER:ME-Linker* mouse (N = 21–36 views). **e** Scheme of

timepoints for MCAO surgery, 2PLSM and experimental design. **f** Representative still-frame images and kymographs of MCA before and after (Occ.2hRep.22 h) ischemic stroke in *SMACreER:ME-Linker* mouse under 2PLSM. The green signals in the vascular wall represents Ca²⁺ signals indicated by GCaMP6s. The magenta solid line represents the resliced position. **g** Statistic of the basal Ca²⁺ levels change rate in the SMC cytosol in the comparison of before and after (Occ.2hRep.22 h) ischemic stroke in *SMACreER:Ai96* mice (blue, N = 6, n = 47 cells) and *SMACreER:ME-Linker* mice (red) (N = 5, n = 44 cells). **h–j** Statistical analysis of the calcium index change rate before and after (Occ.2hRep.22 h) ischemic stroke in in *SMACreER:Ai96* mice (blue, N = 6 mice, n = 47 cells) and *SMACreER:ME-Linker* mice (red, N = 5 mice, n = 44 cells), including Ca²⁺ oscillation frequency (**h**), Ca²⁺ oscillation peak interval SD (**i**), and Ca²⁺ oscillation amplitude (**j**). Data are expressed as the mean ± SEM. Statistic were analyzed using unpaired *t* tests.

under high-resolution microscopy in vivo. However, there are increasing evidences to show that vasomotion and its corresponding flowmotion could promote functional microcirculation[15,16] and protect tissue oxygenation[17–20] in the central nervous system. Based on these previous foundations, we advanced the research area by demonstrating that the vasomotion restoration could promote cerebral circulation after recanalization, which links the vasomotion dysfunction to the pathophysiological relevance in ischemic stroke. Our results are consistent with recently published theoretical biology papers, where they explained the efficient mechanism of vasomotion for the spatial regulation of microcirculation[54], and the turbulence suppression function of pulsatile flows[55].

No-reflow following futile recanalization leads to over half of surgery-treated patients not recovering to an independent life[56,57]. After the pioneer report of no-reflow phenomenon in albino rabbit brain by Dr. Adalbert Ames in 1967[58], the implicated mechanisms in no-reflow are mainly restricted in microvascular dysfunctions, however, how cerebral arteries may actively affect cerebral blood flow during post-ischemic no-reflow had not been studied before. Accordingly, our study had provided a direct evidence that no-reflow is not associated with changes in the arteriolar diameter before and after stroke, but with the dynamic vasomotion of the arteriolar wall. The cellular mechanism underlying cerebral blood flow regulation has been debated for nearly a decade regarding whether capillary pericytes[39,40] or arteriolar SMCs[37] are the major contributors. In addition to the likelihood that SMCs were misidentified as pericytes in these two reports[39,40], all these debates are based on correlation studies without manipulating the contractility of SMCs or pericytes. This study filled this gap by specifically enhancing SMC oscillatory contraction through artificial strengthening of the Ca²⁺ oscillation while leaving pericytes unmanipulated, which still led to the repairment of the post-ischemic cerebral blood flow. These data strongly support a major contribution from SMCs, otherwise, they would not achieve thorough cerebral circulation recovery.

Nevertheless, we mean not to exclude the important role of capillary network dysfunction in the formation of post-ischemic no-reflow. We hold the opinion that multiple factors orchestrated together, but with different temporal and spatial strengths, in the formation of post-ischemic no-reflow in clinic. Most researchers would agree that the complete picture of ischemic pathological changes cannot be exhibited in one particular kind of pathological process and disease model. We conclude here that the vascular protection strategy in the ischemic stroke treatment can be as important as neuron protection, and the maintaining of arteriolar myogenic vasomotion are of critical importance in preventing post-ischemic no-reflow.

In this project, we were focusing on the myogenic aspect of the spontaneous vasomotion during ischemic stroke, as the non-neuronal factors that influence blood flow can be as large as or larger than the neuronal-evoked components[59]. All mice were anesthetized with phenobarbital sodium intraperitoneally in this study, in this way, the neuronal factors which may affect vasomotion were blocked or attenuated simultaneously. Our experimental system was designed to study the non-neuronal spontaneous vasomotion, however, we should not neglect neuronal regulating affects toward vasomotion during ischemic stroke. To examine the vasomotion during ischemic

stroke in awake animals may improve our understanding in this area and should be studied in the future.

This research opens a new avenue toward developing SMC-specific Ca²⁺ oscillators for preclinical testing. Further investigation to seek approaches, such as Ca²⁺ oscillators, that can pump up the periodical contraction is merited. In summary, our study demonstrates that under ischemic conditions, upstream myogenic SMCs control optimal blood perfusion at the downstream capillary network. Mitochondria-ER contact regulators might serve as promising druggable targets to treat no-reflow. Finally, since myogenic spontaneous vasomotion occurs broadly in many organs, the concept cultivated by this study may be generalized to post-ischemic injuries in peripheral organs.

## Methods
### Mice
All animal protocols were approved by the Institutional Animal Care and Use Committee (IACUC) at Westlake University. We have complied with all relevant ethical regulations for animal use. The following mouse strains were used: wild type (C57BL/6 J), *SMACreER*, *Cdh5CreER*, *Ai14* (JAX:007914), *Ai47*, *Ai96* (JAX:028866), and *RCL-ME-Linker* (this paper). All mice were bred and maintained in a specific-pathogen-free animal room on a 12-hour light-dark cycle and provided food and water *ad libitum* at Westlake University Laboratory Animal Resource Center (LARC). 3 to 6 months old adult mice were used in this study. Both male and female mice were used in this study. For breeding the conditional overexpression mice in SMC, an efficient temporally-controlled transgenic mouse line, the *SMCreER* mice were used[35]. For the tamoxifen-inducible Cre (CreER)-dependent conditional overexpression systems, we performed tamoxifen (MCE, Cat# HY-13757A/CS-2870) administration (15 mg/ml dissolved in corn oil, 3 mg daily for three consecutive days through intraperitoneal injection) and induction on adult mice.

### Generation of RCL-ME-Linker transgenic mice
The RCL (*ROSA26/CAG promoter/LoxP-STOP-LoxP*)-*ME-Linker* mice, commissioned by our lab, were generated by Biocytogen Inc. (Beijing, China). The targeting vector contained a CAG promoter and a floxed-terminator followed by *ME-Linker*, with *GCaMP6s* coexpressed (*ME-Linker-P2A-GCaMP6s*). After construction of the targeting vector, the phenotype was first validated in 293 T cells and primary SMC cultures in vitro. Next, the targeting vector, in association with the Cas9/sgRNA plasmid targeting the exon 1 and exon 2 sequence gap of the safe harbor ROSA26 locus, was microinjected into zygotes to obtain founder mice. Then, the positive founders were screened by PCR product sequencing. The positive founder mice were crossed with WT mice to obtain F1 mice that contained heterozygotes of the expected mice. Abnormal multiple-insertion mice were excluded through southern blot verification assays. *RCL-ME-Linker* mice were crossed with *SMACreER* mice to breed tamoxifen-dependent SMC-specific ME-Linker-overexpressing mice. For breeding the double positive mice, littermates who carry non-double positive genes were used as control mice in this study.

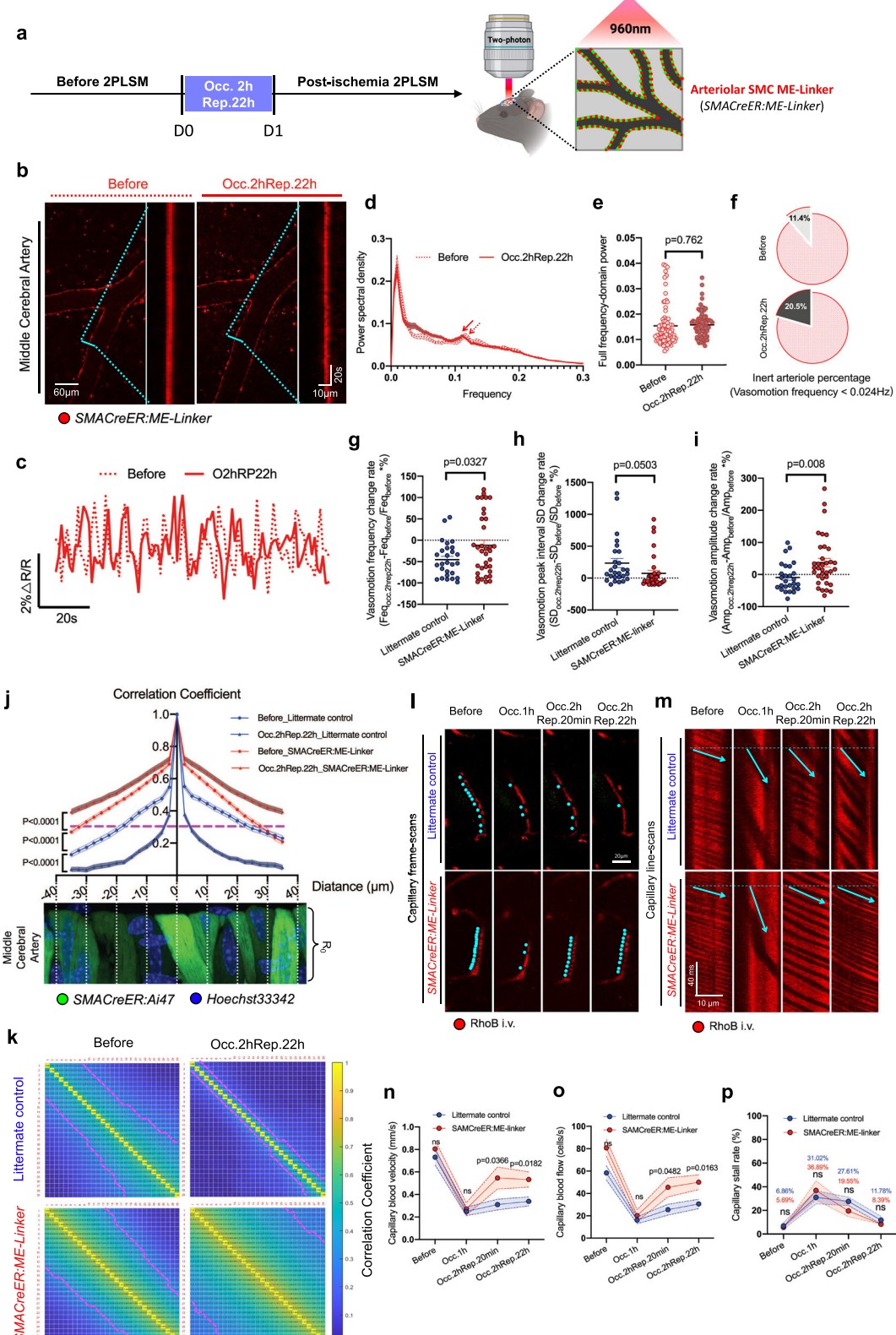

## Echocardiograpgy

Evaluation of heart functions of *SMACreER:ME-Linker* mice and littermates was conducted by transthoracic echocardiography (Vevo 3100, Visual Sonics) as previsouy reported[60]. Briefly, mouse was depilated and anesthetized using isoflurane, then transformed to the heating pad equiped with ECG electroes. The heart rate was maintained at around 450 beats per minute by adjusting isoflurane concentration (0.8 ~ 1.5%). Ultarsound gel was applied to the mouse precordium before ultrasound probe adjusting to the ultrasounic region. The left ventricle and outflow tract was captured in B-mode. The measurements of left ventricular posterior wall (LVPW), left ventricular interior diameter (LVID) and left ventricular anterior wall (LVAW) were determined in M-mode. The blood velocity in mitral valve

**Fig. 6 | Forced Mitochondria-ER tethering rescues post-ischemic impairments in arteriolar myogenic spontaneous vasomotion and capillary perfusion. a** Scheme of timepoints for MCAO surgery, 2PLSM and experimental design. **b** Representative still-frame images and kymographs of arterioles before and after (Occ.2hRep.22 h) ischemic stroke in *SMACreER:ME-Linker* mice. **c** Representative arteriolar radius time series curves before (dash line) and after (solid line) (Occ.2hRep.22 h) ischemic stroke in *SMACreER:ME-Linker* mice. **d** Fourier transform analysis of the rhythmic fluctuations in the arterioles before and after (Occ.2hRep.22 h) ischemic stroke in *SMACreER:ME-Linker* mice (N = 4 mice). The red arrow indicates the frequency peak around 0.1 Hz. **e** Statistical analysis of accumulated power (AUC, area under the curve) of vasomotion within the frequency range of 0–0.3 Hz before and after (Occ.2hRep.22 h) ischemic stroke in *SMACreER:ME-Linker* mice (N = 4 mice). **f** Pie chart representing the percentage of 'inert arterioles' before and after (Occ.2hRep.22 h) occlusion in a paired measurement in *SMACreER:ME-Linker* mice (N = 4 mice), which oscillates with a frequency less than venular vasomotion (0.026 Hz). Statistical analysis of the vasomotion index change rate before and after (Occ.2hRep.22 h) ischemic stroke in control littermates (blue) (N = 7 mice, n = 27 vessels) and *SMACreER:ME-Linker* mice (red) (N = 5 mice, n = 35 vessels), including vasomotion frequency (**g**), vasomotion peak interval SD (**h**), and vasomotion amplitude (**i**). **j** Cooperation index analysis represents the relationship of the Pearson correlation coefficient of the vascular radius changes along arterioles (2.5 μm interval between each adjacent checkpoint) in before and after (Occ.2hRep.22 h) ischemic stroke conditions (6–8 arterioles or 104–177 arteriolar segments from 4 mice). The representative image of the middle cerebral artery (from *SMACreER:Ai47* mouse) below the curve helps to depict the spatial structure of arteriolar SMCs. The magenta dash line indicates correlation coefficient is 0.3. **k**, Representative matrixplot of cooperation index of the same arterioles before and after (Occ.2hRep.22 h) ischemic stroke, in littermate control and *SMACreER:ME-Linker* mouse. The magenta dash line indicates correlation coefficient is 0.3. **l** Representative frame-scan images of RhoB-labeled capillary blood flow before and after (Occ.2hRep.22 h) ischemic stroke in control littermates and *SMAcreER:ME-Linker* mice. The turquoise dots represent the spatial position of the RBC. **m** Kymographs of RhoB-labeled capillary blood flow before and after (Occ.2hRep.22 h) ischemic stroke in control littermates and *SMAcreER:ME-Linker* mice. The shaded streaks indicate the RBC motion trajectory, and the red streaks indicate the tracer-filled capillary lumen. The turquoise arrow indicates the flow direction. The capillary blood velocity (**n**) and blood flow (**o**) analysis before and after (Occ.2hRep.22 h) ischemic stroke, based on the slope and the cell number counting of the kymograph data in control littermates and *SMAcreER:ME-Linker* mice (N = 4 mice, n = 26 or 28 vessels). **p** Capillary stall rate analysis during MCAO induced ischemic stroke in control littermates and *SMAcreER:ME-Linker* mice. Statistics were calculated as the percentage of stalled capillaries (no blood cell flowing longer than 10.84 seconds/10 fames) among all capillaries under one imaged view (0.259 mm²) (N = 4 mice, n = 12 views). Data are expressed as the mean ± SEM. For comparisons of data between two groups, unpaired *t* tests were used.

(MV), aortic valve (AV), pulmmonary valve (PV) and aortic arch (Ao Arch) were determined in PW Doppler mode. The MV velocity was determined in PW tissue Doppler mode. All the data with blinded genotypes were analyzed in Vevo Lab software (v5.5.1, FUJIFILM Sonosite Inc.).

## Blood pressure and body temperature measurement
To compare the physiological signals between different genotypes, all the data were collected in awake mice. The blood pressure of mouse were detected and analyzed using a noninvasive blood pressure measurement system (Kent, CODA). The tail-cuff blood pressure system is equipped with a volume pressure recording (VPR) sensor, by cooperating with the physiological signal analytical system (Biopac, MP160), systolic pressure of mouse were extracted. All the measurements were performed in a quiet room, in addition, all mice were left for enough time for them to return to calm before testing. The body temperature of mouse was recorded by rectal thermometer (WANCE, TH-212).

## Anesthetization
All mice were anesthetized with 1% pentobarbital sodium intraperitoneally (10 ml/kg for mice with a body weight of approximately 25–35 g) during surgery and live imaging assays in this study. Isoflurane was not occupied in this project (except echocardiography) due to its property to dilate arterioles and induce cerebral hyperemia[61,62], which may interfere the current study.

## Cranial window surgery
After anesthetized with pentobarbital sodium, mice were maintained at a temperature of 37 °C under a heating pad throughout the cranial window surgery. A butterfly metal adaptor was first glued onto the skull, which was used for head fixation under 2PLSM. The cranial surgery was performed by drilling a 3-mm round window on the anterolateral parietal bone overlying the middle cerebral artery (MCA) territory. Afterward, the cranial window was sealed with a 3-mm round coverslip (Warner Instruments, CS-3R, Cat# 64-0720) with the instance adhesive (deli 502 super glue, Cat#No.7146). A small hole on the skull was drilled near the coverslip, when there is any drug need to be delivered. Two-photon imaging was performed on the second day after cranial window surgery. Otherwise, to avoid the influence of acute inflammation on vasomotion after surgery, in the cases of ischemic arteriolar vasomotion detection, the mice with a two-week chronic cranial window were checked and showed no difference between the mice with an acute cranial window.

## Middle cerebral artery occlusion (MCAO) surgery
Focal cerebral ischemia was induced by the middle cerebral artery occlusion (MCAO) method as described previously[63]. Briefly, mice were laid on their backs and carefully placed on a 37 °C heating pad. A surgical incision in the neck region was made to expose the right common carotid artery (CCA). After ligating the distal side and the proximal side of the CCA, a small incision was subsequently made between the two ligatures. Then, a silicon rubber-coated monofilament with a rounded tip (Diccol, Cat# 7023910PK5Re) was inserted intraluminally. The monofilament was introduced along the internal carotid artery until the origin of the MCA was occluded. The monofilament was left for 2 h to prompt transient focal cerebral ischemia. Afterward, reperfusion was performed by withdrawing the monofilament for another 22 h. Sham surgery was achieved by inserting the monofilament but not to the MCA, then withdrawing immediately, leaving the CCA ligation permanently.

## Laser Doppler flowmetry (LDF)
Flowmetry measurements were performed with the moorVMS-LDF monitor (Moor Instruments) equiped with a 785 nm laser. A single-fiber probe was located by holding on a custom tube-maded adaptor over the thinned skull and the MCA branches. All mice were anesthetized with 1% pentobarbital sodium during CBF detection. The laser Doppler tissue blood flow (flux) was collected along with the ischemic stroke, which is related to the product of average speed and concentration of moving blood cells in the tissue sample volume. All data were alalyzed with the moorVMS-PC software (Moor Instruments). The relative CBF was calculated by normalizing the LDF flux to the baseline conditions.

## Laser speckle contrast imaging (LSCI)
Blood flow in the mouse brain was recorded in real-time using an RFLSI III device (RWD Life Sciences, Shenzhen, China) throughout the whole process of ischemic stroke. All mice were anesthetized with 1% pentobarbital sodium during CBF detection. For the first-time detection, the mouse scalp was cut off to expose the skull bone, and saline was instilled on the skull surface to maintain its moisture. The detection laser (785 nm) power was set to 110 mW to obtain clear speckle contrast images in the focal plane of the cerebral vasculature. Afterward, stitch the skin briefly, the skull-exposed area was left for follow-up detection and smeared with Vaseline ointment daily to avoid tissue dehydration. The ROIs selected for CBF analysis were rectangles involving the MCA-covered cortical areas. Relative CBF detection and analysis were performed with built-in LSCI software (RWD Life Sciences, V01.00.05.18305).

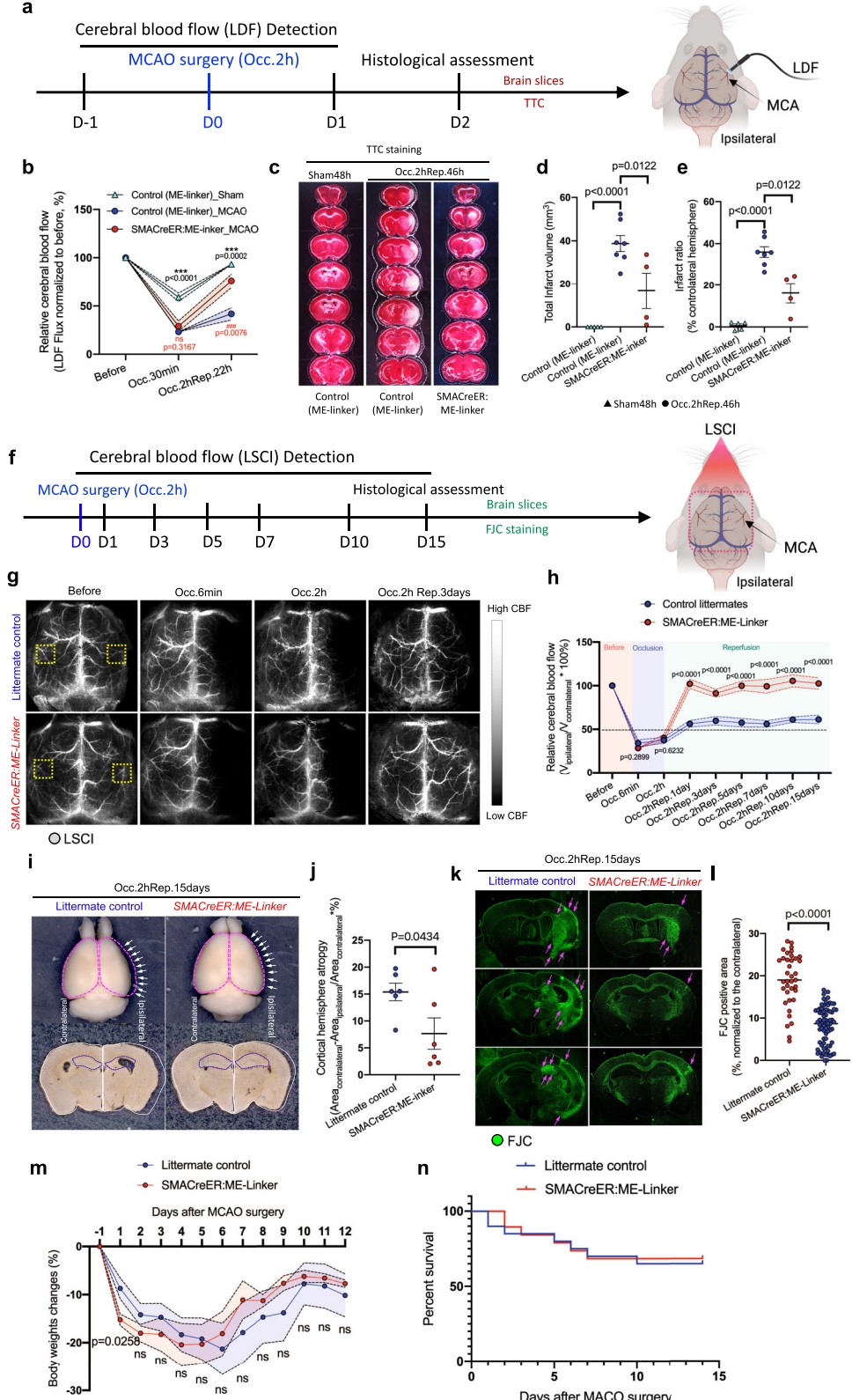

### Two-photon laser scanning microscopy (2PLSM) and in vivo time-lapse imaging

Mice were live-imaged using a two-photon laser scanning microscope (Olympus, FLUOVIEW, FVMPE-RS) equipped with a cooled high-sensitivity GaAsP PMT detector and an ultrafast IR pulsed laser system (Spectra-Physics, InSight X3, continuously variable wavelength range 680 nm–1300 nm). Pictures were acquired in a 512 pixel × 512 pixel square with a 0.994-μm pixel size under a 25x water-immersion objective (Olympus, XLPLN25XWMP2, NA = 1.05).

The galvanometer scanner was used to take time-lapse images with a scan speed of 2 or 4 pixels per microsecond; in this way, the frequency of the frame scan images could be achieved as fast as 0.625–0.926 Hz. All mice

**Fig. 7 | Arteriolar myogenic spontaneous vasomotion improvement are sufficient to replenish global cerebral circulation and attenuates brain atrophy after ischemic stroke. a** Scheme of timepoints for MCAO surgery, 2PLSM and experimental design. **b** Statistical analysis of the relative CBF changes (LDF method) at different time points before and after MCAO or sham surgery in control (*ME-Linker*) and *SMACreER:ME-Linker* mice ($N = 4$ mice for each group). All measurements were normalized to the basal CBF before surgery. Data are expressed as the mean ± SEM. **c** TTC staining for control (*ME-Linker*) and *SMACreER:ME-Linker* mice after 2 days of MCAO or sham surgery ($N = 4$–7 mice). **d, e** Statistical analysis of the total infarct volume and the infarct ration (% contralateral hemisphere) using TTC staining. **f** Scheme of timepoints for MCAO surgery, 2PLSM and experimental design. **g** LSCI images of the mouse whole brain indicate the time course changes in the CBF between control littermates and *SMACreER:ME-Linker* mice after ischemic stroke. **h** Statistical analysis of the relative CBF changes at different time points after ischemic stroke ($N = 10$ or 8 mice). All measurements were normalized to the basal CBF before ischemic stroke. **i** Representative images of the mouse brains and brain slices after (Occ.2 hRep15days) ischemic stroke in

control littermates and *SMACreER:ME-Linker* mice. White arrows indicate the ipsilateral atrophy tissue and magenta dash lines outline hemispheres. **j** Statistical analysis of the cortical hemisphere atrophy rate after (Occ.2 hRep15days) ischemic stroke in control littermates and *SMACreER:ME-Linker* mice ($N = 6$ mice). **k** Representative images of FJC staining results after (Occ.2hRep15days) ischemic stroke in control littermates and *SMACreER:ME-Linker* mice. The magenta arrows indicate the FJC positive in control littermate brain slices. **l** Statistical analysis of the percentage of the FJC-positive infarct area of the whole brain after (Occ.2hRep15-days) ischemic stroke in control littermates and *SMACreER:ME-Linker* mice ($N = 6$ mice, $n = 37$ or 72 slices). **m** Body weight changes along the time course of reperfusion after 2 h occlusion in control littermates and *SMACreER:ME-Linker* mice ($N = 8$ or 10 mice). **n** The survival rate of the control littermates and *SMACreER:ME-Linker* mice ($N = 19$ or 20 mice) along the time course of reperfusion after 2h occlusion in the MCAO ischemic model. Data are expressed as the mean ± SEM. Statistics for (**d, e**), one-way ANOVA, followed by Tukey's post hoc analysis were used. For comparisons of data between two groups, unpaired *t* tests were used.

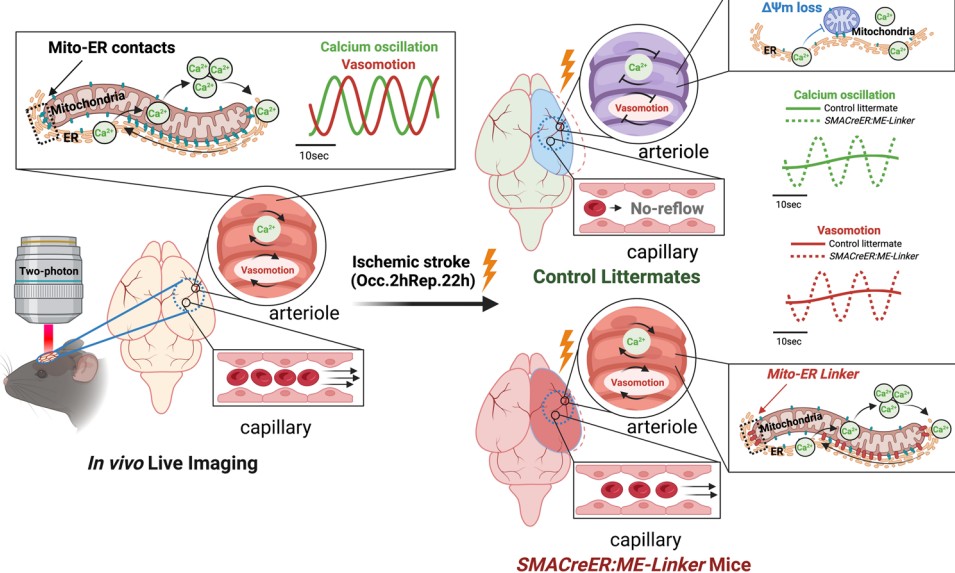

**Fig. 8 | Myogenic spontaneous vasomotion in no-reflow counteraction.** (Left) Physiologically, SMCs manifests spontaneous cytosolic Ca²⁺ oscillation and couples with sinusoidally fluctuating myogenic spontaneous vasomotion in cerebral arterioles. (Right top) After ischemic stroke, transient ΔΨm loss in SMCs impairs mitochondriaito-ER contact structures, which irreversibly deteriorates the intracellular Ca²⁺ pacemaker, and eventually destorys arteriolar myogenic spontaneous vasomotion and cerebral circulation. (Right bottom) ME-Linker overexpression in SMCs could remotivate myogenic spontaneous vasomotion by tethering mitochondria and ER, prevents no-reflow and alleviates post-ischemic neuronal injuries.

were anesthetized with 1% pentobarbital sodium during live imaging. For vasomotion detection in vivo, *SMACreER:Ai47* mice were used to label the vascular walls. For SMC cytosolic calcium detection in vivo, *SMACreER:Ai96* mice (GCaMP6s reported) and *SMACreER:ME-Linker* mice (*GCaMP6s* was tandemly linked with *ME-Linker*) were used. The vascular lumen was labeled by the fluorescent dye rhodamine B-dextran (Wt~70kD) (Sigma, Cat#R9379, 10 mg/ml) through tail intravenous injection. A 960 nm wavelength excitation was used navigate SMC (EGFP), calcium oscillation (GCaMP6s) and vasculature (RhoB). In MCAO models, the same region was detected before and after (Occ.2hRep.22 h) ischemic stroke with the frame scan mode under 2PLSM. For all the mice, time-lapse vascular changes in the same area under any intervention were fully recorded during the entire assay. All mice were maintained at a temperature of 37 °C under a heating pad throughout the imaging process.

## SMC Ca²⁺ and vascular diameter/radius measurement
All time-lapse pictures were analyzed in Fiji (version 2.3.0) and MATLAB (version R2021a; MathWorks) using custom-written scripts. SMC calcium, vascular diameter and radius were determined as described above from frame scan images collected at frequencies of 0.625–0.926 Hz. Measurements in which the mouse had substantially moved were excluded from further analysis. In the calcium-signal-containing (*SMACreER:Ai96* and *SMACreER:ME-Linker*) images, regions of interest

(ROIs) were drawn on several SMCs according to their outline (on average 3–5 per FOV), and the change in calcium signal under time stacks was reflected by the changing of the average green fluorescence brightness of each SMC. In the vessel-containing images, ROIs were drawn on several vessel segments (on average 3–5 segments per FOV), where pixel brightness along a line orthogonal to the length of the blood vessel was extracted. The diameter of the vessel was determined as the full width at half maximum of the line profile, and the vessel mask of every time stack was created by brightness variation along horizontal pixels. The peak and valley intensity thresholds were determined visually for each image to ensure optical detection of the surface vessels. The parameters of the full width at half maximum were calculated by linear interpolation. The diameter change was obtained by sorting the diameter obtained by each time stack. For radius calculation, the midline of the resliced blood vessel observed visually was used as a fixed boundary, and the half-height position obtained by linear interpolation on the either left or right side was used as the other boundary. The radius change was obtained by sorting the radius of each time stack. The codes can be found at part 1 "1_Diameter_detection" at https://github.com/JialabEleven/Vasomotion_index.

## Vasomotion index calculation
For myogenic spontaneous vasomotion characterization, all data used for extracting the vasomotion index must follow a normal distribution. The

vasomotion index is defined as the statistics on the total number of vaso-motion events that exceeded the double standard deviation (SD) of the baseline[64]. Baseline values were determined as the sixth smallest number in a twenty sliding window producing a variable $F_0$ time series, referred to as the "baseline smooth". The changing $F_0$ represents the baseline. The vasomotion index includes the calcium index, diameter index, and radius index. These indexes all contained parameters of frequency, amplitude, and SD of peak intervals. When the amplitude of vasomotion exceeded the double SD of the baseline, it is defined as a single event. Among them, frequency means the events obtained according to the above judgment indicators divided by the total time stacks. The SD of the peak interval (the time interval between two consecutive events) shows the stability of the time interval between two consecutive events. For the calcium index, the amplitude represents the changes in GCaMP6s fluorescence intensity, displayed as a change in cal-cium ($\Delta F$) over baseline ($F_0$). For the diameter or radius index, the ampli-tude represents the change in width values obtained from the two boundaries ($\Delta D$ or $\Delta R$) over baseline ($D_0$ or $R_0$). The inclusion criteria for the calcium events were similar as in the events selection in vasomotion. The codes can be found in part 2 "2_Vasomotion_index" at https://github.com/JialabEleven/Vasomotion_index.

### Fourier transform analysis of vasomotion

Fourier transform plots were generated in MATLAB and presented the power spectral density on the y axis and frequency on the x axis according to shooting parameters. By applying fast Fourier transform (FFT) to the time traces of various vasomotion index changes in vessels, the frequency char-acteristics of diameter, radius and calcium fluorescence intensity change were analyzed separately, and the filtered power of the ultra-low frequency spectrum (0–0.3 Hz) was shown. The sum of power of ultra-low frequency spectrum (0–0.3 Hz) in each vasomotion index was calculated and statis-tically summarized, and the differences of ultra-low frequency power spectral density of vasomotion index (diameter, radius and calcium) between pathological and normal physiological conditions were compared. The codes can be found at part 3 "3_Frequency_calculation" at https://github.com/JialabEleven/Vasomotion_index.

### Cooperation index analysis of arteriolar segments

The Pearson coefficient of correlation and matrixplot of SMC-controlled vasomotion were generated in MATLAB. Along the arteriole, the core position is selected randomly, and the radius change is calculated in this place with time stacks as core vasomotion. By choosing intervals of 2.5 μm, the vasomotion is calculated at each position ±35 μm from the core position. Compare the correlation coefficient between each position vasomotion and the core position vasomotion, the corresponding curve can be obtained. To analyze the cooperation of SMCs under different pathological conditions in different genotype mice, multiple correlation coefficient curves per vessel of various mice were summarized and averaged. In addition, the correlation coefficient matrixplot of different genotype mice is shown as correlation coefficient comparison of each vasomotion statistic position. The codes can be found at part 4 "4_Cooperation_SMC" at https://github.com/JialabEleven/Vasomotion_index.

### In vivo imaging and analysis of arteriolar blood flow

To explore the velocity of the arteriolar blood flow, we used fluorescent DiO-labeled blood cells as the reference substance, and the kymographs of blood cells were collected by 2PLSM as the original data. To prepare DiO-labeled blood cells, we first drew 200 μl of blood from the orbital sinus using a glass capillary tube into 800 μl of HBSS containing 10 mM EDTA disodium. Afterward, the whole blood was washed with HBSS twice by centrifugation for 15 min at 200 rcf (relative centrifuge force). Staining was performed by adding 10 μl of VybrantTM DiO (Thermo Fisher, Cat#V22886, 1 mM) into 490 μl of resuspended whole blood in HBSS and mixing gently in a shaker at 37 °C for 20 min. Next, the DiO-labeled blood cells were spun down, washed once, and resuspended in 500 μl of sterile HBSS. The DiO-labeled blood cells were freshly prepared; routinely, a 100-μl volume of DiO-labeled blood cells

was injected via the caudal vein after the mice were anesthetized. For the baseline arteriolar blood flow imaging, the straight-line ROIs were selected randomly within the lumen area of the MCA, and the imaging speed was set to 2 μm/px. For imaging at time points during occlusion and reperfusion, all the former ROIs were reused to ensure that an identical region was detected during the whole process of ischemic stroke. We analyzed the arteriolar RBC velocity manually by calculating the slope of the DiO-labeled RBC streaks of the line-scan kymographs. The distance traveled by the DiO-labeled RBCs was calculated by counting the pixel number in the X dimension and converted to the physical length by multiplying by the imaging scaling of 0.994 μm/px. The time the DiO-labeled blood cells traveled was calculated in the Y dimension by counting the row number of the streaks and multiplying by the line-sampling rate. The division of the distance and the time values derives the arteriolar RBC velocity. The data were calculated containing directional information, and the basal blood flow direction under the before occlusion condition was marked as a positive value by default.

### In vivo imaging and analysis of capillary blood flow

Capillaries are thin vessels through which only single blood cells can pass. The blood flow in capillaries was detected with occupying red blood cells (RBCs) used as the reference substance, and line-scan images of the RhoB-filled capillary blood flow were collected by 2PLSM. In this way, the shaded streaks indicated the RBC motion trajectory, and the red streaks indicated the tracer-filled capillary lumen. Straight-line ROIs parallel to the lumen were selected within the capillary wall, and the imaging speed was set to 2 μm/px. All the ROIs were reused for the different time points of detection. The analysis of the capillary blood velocity was performed as described in the arteriolar blood velocity analysis using the kymographic data. Fur-thermore, the capillary blood flux was calculated by counting the number of shaded streaks (representing the number of RBCs) and dividing by the total time along line scanning. The capillary stall rate was analyzed with frame-scan time-lapse data, and the proportion of the flowing capillary number within the total capillary number was calculated manually under identical regions before and after (Occ.2hRep.22 h) ischemic stroke. The inclusion criterion of the stalled capillary was that there was no blood cell flowing at least within 10.84 s (10 frames) during the imaging duration of 215.81 s (200 frames).

### Cerebral arteriolar SMC mitochondrial membrane potential (ΔΨm) measurement

For in vivo ΔΨm detection, the anesthetized mice were first subjected to cranial surgery, and the probe was loaded by immersion of the TMRM (Thermo Fisher, Cat#T668, 1 μM dissolved in PBS) and mitotracker Green (Thermo Fisher, Cat#M7514, 1 μM dissolved in PBS)-soaked hemostatic sponge for 20 min through the cranial window. The pial mater was ripped carefully and slightly with acupuncture needle avoid injuring any vessels, to increase the probe labeling efficiency. Concurrently, MCAO surgery was performed, but the last step of inserting monofilament was postponed until the appropriate timing. The baseline ΔΨm levels were then live imaged using 2PLSM in time-lapse mode before occlusion. Afterward, the mono-filament was inserted to induce ischemia as quickly as possible, and this process needed to be finished in 8 min to shorten the time delay between basal and ischemic photographs.

For ex vivo ΔΨm detection, wild-type mice were anesthetized with 1% pentobarbital sodium and then subjected to 2 h MCAO surgery. At the time point of reperfusion 1 h and reperfusion 22 h, mice were perfused with fresh PBS. Then, the brain was extracted and transferred into 1 μM TMRM PBS solution for 10 min of labeling (37 °C). The mouse brains were next washed once with fresh PBS and imaged under 2PLSM in the 1-μm step-size Z-stack mode. The ipsilateral TMRM signal was collected before the contralateral signal to exclude the influence of fluorescence decay.

### TEM sample preparation and TEM microscopy

For MCA ultrastructure detection by transmission electron microscopy (TEM), the mice after MCAO surgery at 22 h of reperfusion were

anesthetized with 1% pentobarbital sodium and then thoroughly perfused with fresh PBS and fresh fixative (2% paraformaldehyde and 2.5% glutaraldehyde in 0.1 M cacodylate, pH = 7.2). Then, the brain was dissected and transferred into fixative for 24 h at 4 °C. The whole MCA vessel was carefully removed from the surface of the mouse brain under a stereomicroscope. For the TEM sample preparation, vessels were first washed with 0.1 M cacodylate buffer and postfixed in 1% osmium tetroxide (in 0.1 M cacodylate) on ice for 1 h before rinsing with cacodylate and ddH2O and incubating in 1% uranyl acetate (in ddH2O) for 1 h at RT. Next, vessels were washed three times with ddH2O and then were dehydrated through an ascending series of ethanol (30%, 50%, 70%, 95%, 100%) each for 10 min at RT. Then, the vessels were rinsed with 100% acetone twice before incubation in an acetone and EPON resin (2:1 and 1:2) mixture and soaked in pure EPON resin overnight at RT. On the second day, the EPON resin was refreshed three times for 3 h prior to EPON resin polymerization for 48 h at 60 °C. Later, 70-nm ultra-thin sections were cut, collected and counterstained with lead citrate and uranyl acetate to further enhance contrast before imaging. TEM examination and image acquisition were conducted on a Talos L120C transmission electron microscope (Thermo Scientific, USA) equipped with a Ceta 16MP CMOS camera and Velox software (Thermo Scientific, USA).

## In vitro primary SMC culture

Primary cultured SMCs were prepared from the pia mater of P0 ~ P2 *SMACreER:Ai14* mice. Specifically, the neonatal pups were first rinsed with 75% ethanol and decapitated carefully to remove the brain from the skull. Then, the brain was transferred into ice-cold HBSS (with 1X penicillin-streptomycin), and the meninges were collected thoroughly by stereomicroscopy. Afterward, the brain tissue was trypsinized and filtered with a 40-µm cell strainer. Cells were then centrifuged and plated into flasks containing Dulbecco's modified Eagle's medium (DMEM) (Sigma-Aldrich, Cat# D6429) supplemented with 10% (v/v) fetal bovine serum (FBS) (Thermo Fisher, Cat#10099141). Thereafter, when the in vitro cultured SMCs reached 80% confluency, tamoxifen (working concentration 5 µM, $4000 \times 20$ mM stock dissolved in DMSO) was administered into the culture medium of cultured SMCs for 6 h to induce Cre recombinase translocating into the nucleus. Two days later, SMCs were ultimately reported to have fluorescent tdTomato (tdT$^+$) expression in the cytosol. To obtain pure SMCs, cultured cells were trypsinized into a single-cell suspension, and tdT$^+$cells were sorted with a flow cytometer sorter (Sony, MA900) with a 70-µm chip. The purification of the sorted SMCs was approximately 100% according to the reported fluorescent marker tdTomato protein.

## Live cell imaging

To evaluate cytosol calcium oscillation, the GECI (genetically encoded Ca$^{2+}$ indicator) tool of YTnC2 was used to label the cytosol calcium[48,49]. AAV2/DJ-CAG-*ME-Linker* virus was used to manipulate Mito-ER contacts, and AAV2/DJ-CAG-control virus was used in the control groups. AAV2/DJ transduction in primary SMCs was performed by adding 1 µl of the virus with a titer of 1^12 viral genomes (VG)/ml into a 35-mm glass-bottom confocal dish (NEST, Cat# 801001) with 1 ml of culture medium (multiplicity of infection (MOI) = 5*10^3 VG/cell). Various constructs were well expressed in the SMCs after 48 h of transduction. In IP3R blocking assay, the fluorescent Fluo-4 dye (ThermoFisher, Cat# F14201) at 1 µM was loaded to paimary SMC (tdT$^+$) for 1 h at 37 °C. The IP3R antagonist 2-APB (MCE, Cat# HY-W009724) at 10 µM was administrated following baseline imaging. Live cells were imaged with a DeltaVision Ultra automated widefield microscope (GE Healthcare) equipped with a custom-designed fluorescence illuminator and an opaque environmental chamber (37 °C, humidity, 21% O$_2$, and 5% CO$_2$ atmosphere). A PlanApoN 60x/1.42NA objective (Olympus) and a sensitive sCMOS camera were used for high-resolution (pixel size 0.1077 µm) and ultra-fast imaging. The interval of the time series frame-imaging is 1 s. The calcium index analysis in vitro was performed by running customized code on MATLAB.

## Correlative light and electron microscopy (CLEM)

Ai14-positive (tdT$^+$) SMCs cultured in a 35-mm 500 µm-grid imprinted dish (ibidi, Cat#81166) were first fixed with fresh fixative (2% paraformaldehyde and 2.5% glutaraldehyde in 0.1 M cacodylate, pH = 7.2) for 10 min at RT. After fixation, the fixed solution was aspirated, and the sample was washed twice with 0.1 M cacodylate buffer (pH = 7.2–7.4). Then, a large field of 10 * 20 panels (1024 × 1024 pixels with a pixel size of 0.1077 µm for each panel) was imaged with a DeltaVision Ultra automated widefield microscope (GE Healthcare) using a UPlanSApo 20x/0.75NA lens (Olympus). After all the light microscopy (LM) pictures were collected, cell samples were fixed secondarily with fresh fixative at 4 °C for 1 more hour. Cell EM sample preparation was performed as described in the vessel EM sample preparation section. After staining, the EPON resin-infiltrated SMCs were detached from the dish briefly in liquid nitrogen. The regions of interest were identified by correlating the grid, letter and number markers on the surface of the block before trimming manually using a sharp razor blade to form a small trapezoid blockface. Using a diamond knife, a 200-nm semithin section was cut and stained with toluidine blue for cell position locating. In addition, 70-nm ultra-thin sections were cut and collected on single slot grids and counterstained prior to EM imaging.

TEM was conducted on a Talos L120C transmission electron microscope (Thermo Scientific, USA). Targeted Ai14-positive (tdT$^+$) SMCs were located by correlating the location on the grid marker according to their pattern in the light microscopy with the grid marker position and cell pattern on low magnification TEM micrographs. Afterward, a series of high magnification pictures of the cells of interest were acquired. For the overlay analysis of the light microscopy and TEM images, all SMCs were identified by their morphology and location on the grid.

## Neutrophil depletion and hematology analysis

Neutrophils were depleted as described previously[38]. A single dose at 4 mg/kg bodyweight of the anti-Ly6G antibody (BD PharMingen, Cat# 551459) and isotope control antibody rat IgG2a (ThermoFisher, Cat# 02-9688) were injected intraperitoneally in mice. Fresh blood samples were collected from mouse orbit at the timepoints before and after (44 h) antibody injection. Whole blood samples were stored in the anticoagulation tube and analyzed on the hematology analyzers (IDEXX Laboratories, ProCyte Dx).

## Oxygen and glusose deprivation (OGD) on acute brain slices

The 100-µm thickness coronal sections of the fresh mouse brain were prepared by vibrating microtome (Leica VT1200) in saturated O$_2$/Glucose$^+$ artificial cerebrospinal fluid (aCSF) within half a hour. The OGD was induced by replacing the complete aCSF medium with the no glucose aCSF and culturing the brain slices in the hypoxia chamber (100% N$_2$ atmosphere) for 1 hour at 37 °C. Afterwards, brain slices were fixed and stained with PI (ThermoFisher, Cat# P3566, 1:3000), NeuroTrace 640/660 deep-red fluorescent dye (ThermoFisher, Cat# N21483, 1:200) and Hoeshst33342 (Sangon, Cat# E607328, 1:5000) for 1 h at room temperature. The brain slices were imaged with DeltaVision Ultra automated widefield microscope (GE Healthcare).

## Cortical hemisphere atrophy measurement

The hemisphere trophy ratio was calculated to reflect cell death-induced brain shrinkage, which is defined as the proportion of ipsilateral hemisphere area versus the contralateral hemisphere area. The dorsal view of the mouse brain and the brain slices was imaged with a Zeiss Axio Zoom. V16 microscope under bright field illumination.

## Infarction volume detection by TTC staining

Infarction volume was measured by TTC staining 46 h after 2 h occlusion. The brain tissue was sliced into seven coronal sections of 1 mm thickness and stained with 1% TTC (BBI, Cat# A610558) at 37 °C for 10 min. Brain slice were imaged with a Nikon SMZ18 microscope under bright field illumination. The infarct area was calculated as the difference between TTC positive contralateral and ipsilateral hemisphere area. The infarct volume

was calculated as the infarct area multiply by the slice thickness (1 mm), and the summation of each slice infarct volume constituted the total infarct volume. The infarct ratio was calculated as the infarct area divided by the TTC positive contralateral hemisphere area.

### Neurological histological assesement

Anesthetized mice were cardiac perfused with 4% paraformaldehyde (PFA), and the brain tissues were post-fixed in 4% PFA overnight. The 50-μm thickness coronal sections of the whole brain were prepared by vibrating microtome (Leica VT1200). To calculate neuronal injury volume, series brain slices (12 slices for one brain) with a interval of 500 μm were selected for staining. For Map2 and NeuN immunofluorescence staining, brain slices were permeabilized with 0.5% Triton X-100 and blocked in 3% BSA PBS solutions. Primary antibody of Map2 (Abclonal, Cat# A22205, 1:100) or NeuN (Abclonal, Cat #A19086, 1:100) were incubated with samples at 4 ℃ overnight, followed by the incubation of the fluorescent goat anti-rabbit Alexa Fluor-488 senond antibody (ThermoFisher, Cat# A1008, 1:2000) for 1 h at room temperature. For Nissl staining, the NeuroTrace 640/660 deep-red fluorescent dye (ThermoFisher, Cat# N21483, 1:200) was used in brain slices. Images of the brain slices were obtained by the widefield fluorescence microscope (Olympus BX53).

### FJC staining

The neurodegeneration ratio across hemispheres was detected with the Fluoro-Jade C (FJC, Bioensis, Cat#TR-100-FJ) method as previously described[65]. Briefly, 20-μm sections of the brain samples were first mounted on adhesion microscope slides and dried in an oven at 60 ℃ overnight. Next, samples on the slides were incubated in 80% alkaline ethanol (5 min), 70% ethanol (2 min), distilled water (2 min), potassium permanganate solutions (10 min) and FJC solutions (10 min) in an orderly manner in a Coplin jar at room temperature. Afterward, the slides were imaged under a widefield fluorescence microscope (Olympus BX53) equipped with a 2X PlanApo N objective (NA = 0.08). The infarct area was calculated as the proportion of the ipsilateral FJC-positive area under the contralateral hemisphere area.

### Neurological severity score (NSS) of mouse behavior test

We adaped the behavior tests for modified neurolohical severity score as perviously reported[66]. The NSS was recorded before and after ischemic stroke over time. Points were awarded either for the inability to perform, or for abnormal task performance, or for the lack of a tested reflex[66].

### Statistics and reproducibility

The numerical data concealed in the raw digital images were extracted and run on the software of Fiji (version 2.3.0/1.53 f) or MATLAB (version R2021a). All statistical analyses and graphical illustrations were performed using GraphPad Prism 8 software (version 8.3.1, California, USA). Unpaired two-tailed t-test and paired two-tailed t-test were used accordingly for the comparisons between the two groups. For multiple groups, data were analyzed using one-way ANOVA, followed by Tukey's post hoc analysis for multiple comparisons. For the time-dependent trend analysis, data were analyzed using two-way ANOVA. A confidence level of 95% was used when evaluating the results, and $P < 0.05$ was considered significant. Investigators were blinded to sample identity during quantitative analysis. All data are expressed as the mean ± standard error of the mean (SEM), and the SEM is displayed as error bars in the column graphs. Sample sizes were determined by the minimum need to offer sufficient statistical power to comply with the 3 R (reduction, replacement, refinement) principle, and optimized according to previous studies. The number of replicates are reported in the figure legends correspondingly. The biological replicates are defined as distinct samples representing an identical genotype or treatment.

### Reporting summary

Further information on research design is available in the Nature Portfolio Reporting Summary linked to this article.

## Data availability

The numerical source data for graphs in the manuscript can be found in supplementary data 1 file. All data generated in this study are included in this article and supplementary file.

## Code availability

Image processing, analysis, and visualization were performed in FIJI and MATLAB (MathWorks). The codes are divided into four parts: diameter and radius calculation of vessels, vasomotion index analysis, Fourier transform plot of vasomotion, and cooperation index analysis of different SMCs in the same blood vessel. All codes are available at https://github.com/JialabEleven/Vasomotion_index.

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

## Acknowledgements

We would like to thank Dr. Kiryl D. Piatkevich for providing the calcium indicator AAV-DJ-YTnC2. We thank Dr. M. Jiang and Mr. J. Chen for providing the mitochondrial probe mitotracker Green. We thank Dr. Y. Mu and Mr. Y. Qian from Center for Excellence in Brain Science and Intelligence Technology, Chinese Academy of Sciences for technical support on the code writing of vasomotion quantification. We thank Dr. Y. Ma from Shenzhen Institute of Advanced Technology (SIAT) for sharing the rt-PA. We thank Mr. G. Fang and Dr. Y. Sun for their help in TEM sample preparation, section, and imaging. We thank all the staffs in the Microscopy Core Facility (MCF) at Westlake University for their technical assistance. We thank all the animal keepers in the Laboratory Animal Resource Center (LARC) at Westlake University. Illustrations in this paper were created with BioRender.com. J.-M.J. acknowledges the support from Westlake University startup funding, the Westlake Education Foundation, the "Pioneer" and "Leading Goose" R&D Program of Zhejiang (grant 2024SSYS0031), Zhejiang Province Natural Science Foundation (Project #2022XHSJJ004), the National Natural Science Foundation of China (Projects #32170961 and #31970969), HRHI programs 203309002 and 201109013 of Westlake Laboratory of Life Sciences and Biomedicine. J.L. acknowledges the support from National Natural Science Foundation of China Youth Program (Project #82001267).

## Author contributions

J.L. and J.-M.J. conceived the project. J.L. and J.-M.J. designed the experiments. J.L., Y.Z., and W.W. performed experiments. D.Z. helped in the TEM sample preparation and imaging. H.X. and J.R. participated in the LSCI assay. Y.J. participated in the FJC staining assay. T.L. and X.L. participated in the brain slice preparation and imaging. B.Z. conducted in the animal breeding and genotyping. X.Z. helped in performing neutrophil depletion assays. H.S. and J.L. (Jiayi Lin) conceived and performed echocardiographic assays. Y.Z. wrote code for the current study. J.L., Y.Z. and W.W. analyzed all data. J.L. and J.-M.J. wrote the manuscript. J.L. and J.-M.J. acquired the financial support for the project leading to this publication. All authors read and approved the final manuscript.

## Competing interests

The authors declare no competing interests.
