## [Peer review file · Communications Biology]

Reviewers' comments:

Reviewer #1 (Remarks to the Author):

The authors investigated the role of vasomotion in the reduced microvascular flow observed after transient focal ischemia in a mouse model of temporary middle cerebral artery occlusion. Using a combination of technique to image the cerebral vasculature in vitro and in vivo in genetically engineered mice, they observed a suppression of vasomotion after ischemia-reperfusion. The effect was associated with a reduction in the Ca²⁺ oscillation in smooth muscle cells (SMC) driving vasomotion. Mitochondria-ER contact points were reduced after ischemia and reestablishing such contacts, by overexpression of the mitochondria ER-linker in SMC, rescued the Ca²⁺ oscillations and vasomotion, improved capillary perfusion and reduced the acute brain injury and subsequent post-ischemic brain atrophy. The data implicates mitochondria-ER contacts in the mechanisms of post-ischemic no-reflow.

This is an extensive study that sought to shed light on the mechanisms of the no-reflow phenomenon that accompanies ischemia-reperfusion in brain. While the topic is of interest in the era of thrombectomy and recanalization in stroke patients, the data raise a number of questions that remain to be addressed.

1. The no-reflow phenomenon has been extensively studied since the pioneering work of Adalbert Ames in the 1960s and a wide variety of mechanisms have been implicated, including vasoparalysis, intravascular occlusion by platelets and leukocytes, capillary stalling by neutrophil, and pericyte constriction, among others. The present study does not explore if suppressed vasomotion counteracts also these well-described aspects of the phenomenon.
2. It is unclear how reestablishing vasomotion, which travels in opposite direction to the flow in arterioles, is able to restore tissue perfusion. Data on red blood cell flow in isolated vessels does not provide insight into the perfusion of the entire ischemic territory.
3. Part of the problem is that cerebral blood flow was not measured, but only cerebral blood volume, which reflects mostly the venous compartment. Measurement of cerebral blood flow in the ischemic territory would be needed to assess the impact of vasomotion in the re-perfusion of the post-ischemic territory.
4. In this regard, the studies in the transgenic mice expressing the ME-linker are problematic in that quantitative cerebral blood flow data before and after ischemia is not provided. Therefore, it cannot be ascertained if the reduced brain injury is the results of a less severe ischemic challenge.
5. Similarly, it remains to be determined if the ME-linker overexpressing mice have the same threshold to ischemic injury as their wild-type counterpart. Studies in organotypic brain slices exposed to oxygen-glucose deprivation may provide insight into this issue.
6. The ischemic damage was assessed by fluoro-jade stain, an indirect marker of neuronal injury. A more robust assessment with histology should be performed at different times after ischemia to determine the volume of injury and if the protection is maintained over time.
7. The brain sections showing hemispheric atrophy do not show evidence of focal brain damage (fig.

7C). Therefore, the injury leading to subsequent atrophy cannot be appreciated.

8. The language needs editing e.g. line 145, 249, 250, 274, 278, etc.

Reviewer #2 (Remarks to the Author):

This study examines the rhythmicity of spontaneous vasomotion in the cerebral vasculature in vivo, and how it is affected during ischemic stroke. The authors correlated the spontaneous vasomotion activity with calcium signaling and stable ER-mitochondria contacts in vascular smooth muscle cells. The data suggest that both calcium oscillations and ER-mitochondria contact sites are altered after ischemic stroke. Data also suggest that re-establishing/strengthening the ER-mitochondria contacts prevents the loss of calcium oscillations, restores flow to the capillary region faster and more effectively, and ameliorate brain atrophy and cell death compared to control conditions. The reviewer has several comments for the authors' consideration.

1) A key concern is about physiological relevance. First, the spontaneous oscillations in control vascular smooth muscle are relatively small. Thus, it needs to be clarified how much they influence vascular function. Moreover, while the spontaneous oscillations go away after ischemic stroke and that correlates with no-reflow, the loss of spontaneous oscillations may result from the insult rather than a cause/effect.

2) The nature of the calcium signals needs to be clarified. First, the authors give the impression that they could be IP3R-mediated signals but also be mitochondria. The authors should establish the identity of these calcium oscillations. Also, what is a calcium clocker?

3) Data seem to suggest that maintaining ER-mitochondria contact sites in artery/arteriole vascular smooth muscle cells prevents/ameliorates any capillary blood flow dysfunction. This result contrasts with recent studies indicating that capillary blood flow restoration during pathology is mainly mediated by recovery of capillary function triggering retrograde signaling that dilates upstream arterioles and arteries. The question is, then, how does restoring altered mechanisms in vascular smooth muscle correct capillary dysfunction?

4) There are several concerns with the fluorescent imaging data and interpretation that require further validation. For example, is the expression of the GCaMP6 calcium sensor similar in control and ischemic mice? The calcium imaging data in Figure 4J is not convincing. There are no apparent changes in calcium oscillations from the images provided. Also, the loss of basal signal in CCCP suggests changes in calcium sensor conditions that may confound data interpretation. Figure 5E-F: basal calcium measurements are challenging to assess in isolated systems much less in an intact native system. How did the authors perform the measurements to reach the conclusion that basal cytoplasm is higher in the ER:ME-linker mice? Is SR calcium content different in ER:ME-linker mice compared to control mice? Is mitochondria calcium different in ER:ME-linker mice compared to control mice? Experiments using the TMRM compound to examine mitochondria membrane potential changes need to be more convincing. The loss of fluorescence could result from different

loading compared to control mice. Could the authors use another marker that does not change with ischemic stroke to convincingly establish that the loading of any indicator/sensor is not affected by the ischemic stroke?

5) Do the ER:ME-linker mice show higher cerebral artery myogenic tone, which is expected if there is more “cooperation” between smooth muscle cells? Examples showing smooth muscle cell synchronization will go a long way to support the authors’ claims.

Minor comments:

1) Page 7 – line 145: consider changing the words “fair and capable” to “rigorous and unbiased”.

2) Page 9 – Lines 177-178: What is “chaotic vasomotion”? Change wording.

3) Page 14 – Lines 303-304: The conclusion in this sentence is not supported by the data presented. Authors should change the wording.

4) Page 21 – Lines 460-462: Statements in this section are overstated. The authors never tested calcium antagonists and how they may alter vasomotion. This should be eliminated.

5) It is unclear whether the statistical comparisons performed in Fig 2J are appropriate. How was the data normalized?

Reviewer #3 (Remarks to the Author):

The present study investigated how cerebral ischemia reperfusion injury affects physiological cerebral vasomotion, as well as mechanisms underlying observed post-reperfusion deficits. The authors observed that Ca²⁺ oscillation in SMC underlie physiological arteriolar vasomotion in mice, likely due to communication between SR and mitochondria, all these processes are disrupted by ischemia / reperfusion. Interestingly, expression of a linker peptide increases areas of proximity between SR and mitochondria, partially restoring Ca²⁺ oscillations and vasomotion. Overall, this is a well-designed and executed study, and the findings are relevant to the field and add to the knowledge of cellular alterations caused by ischemia-reperfusion that affect cerebral microvascular control. Some minor issues were identified, described in detail below:

1. Unclear why the authors chose to use CCCP in cell culture experiments to investigate mitochondria-SR linkages rather than performing an oxygen-glucose deprivation model, which fits better with the MCAO data in the remainder of the manuscript. Given that the authors generated a mouse strain expressing the ME-linker in SMC, the cell culture experiments become unnecessary, as they do not add much value to the final conclusion of the study. This reviewer’s recommendation is to either remove them from the manuscript or repeat the experiments in an OGD paradigm;

2. The mitochondria-SR link in control of SMC Ca²⁺ oscillations is an interesting hypothesis, and the data presented in the manuscript are compelling. It would be interesting to investigate the mechanisms underlying this communication, and where the Ca²⁺ is coming from and what organelle is buffering Ca²⁺ release. Are the Ca²⁺ oscillations coming from opening of IP3R or RyR2 in the SR, then buffered by mitochondria? Or is the opposite happening, where mitochondria are responsible for Ca²⁺ release and SERCA buffers it? This should be explored further, if not experimentally at least

in the Discussion;

3. Measurement of basal Ca²⁺ levels in SMC using genetically-encoded indicators are unreliable, as differences in basal fluorescence can be explained by distinct pattern in GCaMP6 expression post-ischemia / genetic manipulation rather than real alterations in cytosolic Ca²⁺. These measurements should be removed from the manuscript, or corroborated using a different technique (for example, ratiometric dyes such as Fura2-AM);

4. In Supplemental Figure S8 the authors included diastolic blood pressure and heart rate, were measured using tail-cuff pletismography. Both measurements are unreliable using this technique, thus they should not be included in the manuscript, even though the data show that there is no difference between groups;

5. In the methodology it is stated that the study included both male and female mice, but there is no mention if the data were segregated and analyzed for potential sex differences in responses. These should be performed and included in the manuscript. Ideally, each datapoint in graphs should be segregated by biological sex;

6. There are some minor grammar inconsistencies and spelling errors throughout the manuscript. Suggest a review of English grammar and spelling.

COMMSBIO-23-1417A, manuscript entitled "Ca²⁺ Oscillation in Vascular Smooth Muscle Cells Control Myogenic Spontaneous Vasomotion and Counteract Post-ischemic No-reflow"

We are thankful for the careful review, helpful suggestions and constructive comments from the referees. We have expanded discussion and performed additional experiments to address all concerns in full, which have improved and strengthened this manuscript. The major changes to the revised manuscript are outlined as below.

1. To increase readability, we have added the scheme of timeline and experimental design in each of the main figures.
2. We have revised and improved the methodology about cerebral blood flow (CBF) detection in this revision. On the one hand, we have optimized the LSCI data analysis process to increase the accuracy in arteriolar CBF examination. On the other hand, have occupied laser Doppler flowmetry (LDF) system in monitoring CBF changes in this revision, along with various kinds of histological assessments in evaluating post-ischemic neurological injury over time.
3. We have expanded the discussion over physiological and pathological relevance of myogenic spontaneous vasomotion, the origins of the calcium oscillation in SMCs, and the conceptual advance of this study. In addition, we have performed in vivo neutrophils depletion, in vitro live calcium imaging, and ex vivo acute brain slice assays in strengthening those discussions.
4. We have improved the methodology of SMC mitochondrial membrane potential detection during ischemic stroke in live mice. Besides, we had designed and performed assays to clarify the effectiveness of the calcium sensors in post-ischemic condition in vivo. Moreover, we have clarified the rationality and calculation process about the basal cytoplasmic calcium levels changes rate in this revision.
5. We have added three co-authors (anonymous during DBPR) in this revision. Miss ?, who helped in performing neutrophil depletion assays. Dr. ? and Mr. ?, who conceived and performed echocardiographic assays in this revision.
6. We have checked and revised all the textual errors throughout the manuscript. Additional textual changes are performed as suggested.

Below are the point-by-point replies to criticisms of 3 referees, with the comments in bold (black) and our responses in regular (blue) text. The revised text in the manuscript were indicated by line number.

Reviewers' comments:

Reviewer #1 (Remarks to the Author):

The authors investigated the role of vasomotion in the reduced microvascular flow observed after transient focal ischemia in a mouse model of temporary middle cerebral artery occlusion. Using a combination of technique to image the cerebral vasculature in vitro and in vivo in genetically engineered mice, they observed a suppression of vasomotion after ischemia-reperfusion. The effect was associated with a reduction in the Ca²⁺ oscillation in smooth muscle cells (SMC) driving vasomotion. Mitochondria-ER contact points were reduced after ischemia and reestablishing such contacts, by overexpression of the mitochondria ER-linker in SMC, rescued the Ca²⁺ oscillations and vasomotion, improved capillary perfusion and reduced the acute brain injury and subsequent post-ischemic brain atrophy. The data implicates mitochondria-ER contacts in the mechanisms of post-ischemic no-reflow.

Response:

Thanks for your careful review of our manuscript.

This is an extensive study that sought to shed light on the mechanisms of the no-reflow phenomenon that accompanies ischemia-reperfusion in brain. While the topic is of interest in the era of thrombectomy and recanalization in stroke patients, the data raise a number of questions that remain to be addressed.

1. The no-reflow phenomenon has been extensively studied since the pioneering work of Adalbert Ames in the 1960s and a wide variety of mechanisms have been implicated, including vasoparalysis, intravascular occlusion by platelets and leukocytes, capillary stalling by neutrophil, and pericyte constriction, among others. The present study does not explore if suppressed vasomotion counteracts also these well-described aspects of the phenomenon.

Response:

The authors absolutely agree that there are a wide variety of mechanisms had been reported and implicated in interpreting no-reflow phenomenon, as the reviewer had mentioned, after the pioneer study in albino rabbit brain by Dr. Adalbert Ames in 1967¹. The mechanisms in no-reflow involve microvascular paralysis and obstruction, however, how cerebral arteries may actively affect cerebral blood flow during post-ischemic no-reflow had not been studied before. As arteriolar smooth muscle cell contractility controls regional blood flow in the normal and ischemic brain², and ischemia has long been reported could decrease myogenic reactivity in middle cerebral artery (MCA)³, we hypothesized arteriolar spontaneous myogenic vasomotion may play essential role in the post-ischemic cerebral circulation regulation. Accordingly, we found here that the dampened myogenic spontaneous vasomotion along the arteriolar wall contributed greatly to the post-ischemic no-reflow in live mice, which had highlighted the novelty of

the current study. To explore microvascular obstruction factor in our experimental setup, we reviewed the capillary live imaging data as shown in Figure S4a and S4b. Based on the data, we found there are few of capillaries work dynamically, as 4.33% of the capillaries were stalled temporarily under our inclusion criterion of stalled capillary counting (10.84 s /10 frames no flowing during 218.81 s/200 frames, see methods) in naive mice (Line 244-247). Further, we found that the capillary stall rate in the control mice was 21.24% at the time point Occ.2hRep.20min, and decreased to 9.25% at the time point Occ.2hRep.22h, suggesting time extension after recanalization could promote capillary blood flow recovery. There is 4.92% of increase of the capillary stall rate at the time point Occ.2hRep.22h compare to before ischemia, implying microvascular obstruction do exist but limited in the MCAO model of transient ischemic stroke (Line 250-253). This result is consistent with a recent preprint (see Figure 3a and 3b)⁴.

We noticed that it has been reported that the post-ischemia capillary stall rate are ~35% in infarct core and ~15% in penumbra at the time point of around 1h after recanalization in a thrombin stroke model⁵, where the authors highlighted capillary neutrophils stalling are a major cause of no-reflow in ischemic stroke. The inconsistency may attribute to the difference of several essential experimental setups, including the type of the transient ischemic stroke model, the detection time point of post-recanalization, and the inclusion criterion of stalled capillary. Last but not least, as shown in Figure 6p, the capillary stall rate was comparable between SMACreER:ME-Linker mice and control littermates, as we had discussed in the manuscript, suggesting that the biological function of ME-Linker was independent of the formation and elimination of capillary obstructions during ischemic stroke (Line 460-462).

Meanwhile, inspired by the above important comment, we had checked whether capillary obstruction is involved in the effect of arteriolar myogenic vasomotion repairment in counteracting post-ischemic no-reflow in this revision (Figure S12). We adapted the method of rt-PA (10mg/kg, Actilyse, Boehringer Ingelheim) administration and neutrophil depletion from the published paper by Mohamad El Amki et al⁵, in order to exclude capillary obstruction effects in post-ischemic no-reflow formation (Figure S12a). To evaluate no-reflow phenomenon, cerebral perfusion of naive or neutrophil-depleted SMACreER:ME-Linker mice and control littermates was monitored using LSCI along ischemic stroke over time.

Interestingly, through hematology analysis in peripheral blood, we do not find significant differences in neutrophil proportion and absolute number between SMACreER:ME-Linker mice and littermates physiologically, in contrast, ischemic stroke could promote neutrophil proliferation (5.21-fold in control mice) in 22 hours extremely (Figure S12b to S12g). Moreover, we found that half neutrophil depletion do not help in restoring no-reflow in both control mice and SMACreER:ME-Linker mice in MCAO ischemic model (Figure S12h and S12i), especially at the timepoint of 22 hours after occlusion. (Line 462-472)

Fig S12. Neutrophil depletion do not helps in restoring no-reflow in SMACreER:ME-Linker mice. a, Scheme of timepoints for MCAO surgery, CBF detection using LSCI, NSS detection, and histological assessment in neutrophil depletion assay. The antibody or rt-PA administration and hematology analysis were performed in the corresponding timepoint as appropriate. **b**, Scatter plots of the leukocyte classification in the peripheral whole blood using ProCyte Dx hematology analyzer between control (ME-Linker) and SMACreER:ME-Linker mice. **c**, Statistical analysis of the proportion of neutrophil (NEUT), lymphocyte (LYMPH), monocyte (MONO), eosinophil (EO), and basophil (BASO) in total number of leukocytes count, before Ly6G antibody and isotype antibody administration. **d**, The absolute number (K/ μ l) of all kinds of leukocyte in the peripheral whole blood in different groups, before Ly6G antibody and isotype

antibody administration. **e**, Scatter plots of the leukocyte classification in the peripheral whole blood using ProCyte Dx hematology analyzer at the time point of reperfusion 20h after ischemic stroke, in anti-Ly6G antibody and isotype antibody administrated mice. **f**, Statistical analysis of the proportion of different leukocytes in total number of leukocytes count, after Ly6G antibody and isotype antibody administration. **g**, The absolute number (K/ μ l) of all kinds of leukocyte in the peripheral whole blood in different groups, after Ly6G antibody and isotype antibody administration. **h**, LSCI images of the mouse whole brain indicate the time course changes in the cerebral CBF before and after MCAO surgery in different groups. **i**, Statistical analysis of the relative cerebral CBF changes at different time points before and after MCAO surgery in different groups (N = 7 or 8 mice for each groups). All measurements were normalized to the basal cerebral CBF before surgery. Data are expressed as the mean \pm SEM. Statistics for **g**, one-way ANOVA, followed by Tukey's post hoc analysis were used. Statistics for **d** and **i**, by using unpaired t test, p values (*) were evaluated between control (ME-Linker) and SMACreER:ME-Linker mice at the timepoint of Occ.2hRep.22h.

To summary, we hold the opinion that multiple factors orchestrated together in the formation of post-ischemic no-reflow. Most researchers would agree that the complete picture of ischemic pathological changes can not be exhibited in one particular kind of pathological process and disease model. We propose here that the vascular protection strategy in the ischemic stroke treatment can be as important as neuron protection, and the maintaining of arteriolar myogenic vasomotion are of critical importance in preventing post-ischemic no-reflow.

2. It is unclear how reestablishing vasomotion, which travels in opposite direction to the flow in arterioles, is able to restore tissue perfusion. Data on red blood cell flow in isolated vessels does not provide insight into the perfusion of the entire ischemic territory.

Response:

We apologize for the confusion about the observed object in the red blood flow detection assay, rather than the isolated vessels, the red blood flow in the capillary network (under pia around 50-250 μ m) was detected in live mice under 2PLSM, through the cranial window within the MCA perfusion region. To increase readability, we have added the scheme of timeline and experimental design of the assay (and also other assays in this paper).

The direction of arteriolar myogenic vasomotion is perpendicular to the arteriolar wall, which facilitating the periodic sinusoidal contraction and relaxation of the vascular walls. When the dampened myogenic vasomotion is reestablished through overexpression ME-Linker in SMC, the post-ischemic cerebral circulation impairments can be rescued largely in vivo as shown in Figure 6 and 7. These results are consistent with recently published theoretical biology papers, where they explained the efficient mechanism of vasomotion for the spatial regulation of microcirculation⁶, and the turbulence suppression function of pulsatile flows⁷.

3. Part of the problem is that cerebral blood flow was not measured, but only cerebral blood volume, which reflects mostly the venous compartment. Measurement of cerebral blood flow in the ischemic territory would be needed to assess the impact of vasomotion in the re-perfusion of the post-ischemic territory.

Response:

Thank you for this important feedback. The authors apologize for the confusion about the oversimplified statement of LSCI in monitoring cerebral blood perfusion, and the misleading about the description of the cerebral circulation/blood flow detection by cerebral blood volume (CBV). LSCI provides a rapid characterization of cerebral blood flow dynamics for functional monitoring microcirculation, which encode time-varying speckle patterns when the blood cells move in vessels of tissues⁸⁻¹⁰. LSCI perfusion unit is related to the underlying blood flow (nonlinear with the speckle contrast values) in the sample volume. However, the speckle technique does not provide any depth specificity⁸. In this way, the cerebral circulation data collected by LSCI is revised to cerebral blood flow (CBF) in this revision. What's more, we have optimized the LSCI ROIs in analyzing CBF data, to make sure the regions cover only MAC territory but not venous compartment. All the LSCI data in this study have been recalculated based on the above criterion, and the representative analyzed LSCI ROIs have been marked in the LSCI images in this revision.

Fig. 1. n, Statistical analysis of the relative CBF changes (LDF method) at different time points before and after MCAO or sham surgery (N = 4 mice for each group). o, TTC staining for mouse brain after 2 days of MCAO or sham surgery (N = 5 or 7 mice). p-q, Statistical analysis of the total infarct volume and the infarct ratio (% contralateral hemisphere) using TTC staining.

In addition, we have established laser Doppler flowmetry (LDF) system this time and added LDF measurement results in this revision, in order to expand on the characterisation of CBF changes during ischemia. As shown in Figure 1n, by using LDF, we have detected a dramatic decrease of the ipsilateral CBF and no-reflow phenomenon at the time point of Occ.2hRep.22h, and the CBF in sham-surgery group is recovered to before surgery levels. The CBF changes pattern during ischemia is similar between LSCI and LDF measurements, indicating the validity of both methods in detecting CBF changes.

Fig. 7. a, Scheme of timepoints for MCAO surgery, 2PLSM and experimental design. **b**, Statistical analysis of the relative CBF changes (LDF method) at different time points before and after MCAO or sham surgery in control (ME-Linker) and SMACreER:ME-Linker mice (N = 4 mice for each group). All measurements were normalized to the basal CBF before surgery. Data are expressed as the mean \pm SEM. **c**, TTC staining for control (ME-Linker) and SMACreER:ME-Linker mice after 2 days of MCAO or sham surgery (N = 4~7 mice). **d** and **e**, Statistical analysis of the total infarct volume and the infarct ration (% contralateral hemisphere) using TTC staining.

LDF system is also occupied in the short-term CBF changes (before and Occ.2hRep.22h) detection in the SMACreER:ME-Linker mice along ischemia. As shown in Figure 7a and 7b, the no-reflow phenomenon is largely rescued, followed by the significant reduction of a neurological deficits by TTC staining (Figure 7c, 7d and 7e).

4. In this regard, the studies in the transgenic mice expressing the ME-linker are problematic in that quantitative cerebral blood flow data before and after ischemia is not provided. Therefore, it cannot be ascertained if the reduced brain injury is the results of a less severe ischemic challenge.

Response:

Thank you for this feedback. As we had discussed above, the cerebral circulation data collected by LSCI is revised to cerebral blood flow (CBF) in this revision, and there is no statistical difference of the CBF reduction during occlusion (including time points of occ.6min and occ.2h, Figure 7g and 7h) between SMACreER:ME-Linker mice and their wild-type counterparts, indicating the unbiasedness and stability of modeling. Besides, the LDF assay had shown similar results, that is there is no significant difference of the relative CBF changes at the timepoint of Occ.30min between SMACreER:ME-Linker mice and littermates (Figure 7b). In addition, CBF by LDF could identify the difference of the relative CBF changes at the timepoint of Occ.30min between MCAO and Sham-surgery group significantly (Figure 7b), indicating the sensitivity of LDF is sufficient to explore the relative CBF changes during ischemic stroke.

5. Similarly, it remains to be determined if the ME-linker overexpressing mice have the same threshold to ischemic injury as their wild-type counterpart. Studies in organotypic brain slices exposed to oxygen-glucose deprivation may provide insight into this issue.

Response:

The authors thank for this suggestion. Since the establishment of the organotypic brain slices system remains challenging in our lab, we have simplified the method by using acute brain slices of SMACreER:ME-Linker mice and their wild-type counterparts, in checking their threshold to OGD insults. The results are shown in Figure S11.

Fig S11. ME-Linker overexpression in SMCs does not affect neuronal survival threshold to ischemic injury

a, Experimental timeline of the in vitro oxygen-glucose deprivation (OGD) assay of acute brain slices. The acute brain slices from control (ME-Linker) and SMACreER:ME-Linker mice were prepared quickly in half an hour. Afterwards, 1h of control or OGD condition was exposed to the corresponding slices, and the threshold to ischemic injury was detected through PI staining after PFA fixing. **b**, Representative images of PI (red) and Hoechst33342 (turquoise) staining results in control and OGD brain slices of control (ME-Linker) and SMACreER:ME-Linker mice. **c**, Statistical analysis of the brain cell death percentage (proportion of PI positive nucleus in Hoechst33342 positive nucleus) in control and OGD brain slices of control (ME-Linker) and SMACreER:ME-Linker mice (N = 4 slices). **d**, Representative images of PI (red) and NeuroTrace Fluorescent Nissl (turquoise) staining results in control and OGD brain slices of control (ME-Linker) and SMACreER:ME-Linker mice. **e**, Statistical analysis of the neuron death percentage (proportion of PI positive nucleus in Nissl positive nucleus) in control and OGD brain slices of control (ME-Linker) and SMACreER:ME-Linker mice (N = 4 slices). **f**, Representative images of Hoechst33342 (green) and NeuroTrace Fluorescent Nissl (magenta) staining results in control and OGD brain slices of control (ME-Linker) and SMACreER:ME-Linker mice. **g**, Statistical analysis of the neuron percentage (proportion of Nissl positive nucleus in Hoechst33342 positive nucleus) in control and OGD brain slices of control (ME-Linker) and SMACreER:ME-Linker mice (N = 4 slices). Data are expressed as the mean \pm SEM. Statistics were analyzed using unpaired t-tests.

A total 1 hour of oxygen and glucose deprivation challenges was adapted in the carefully prepared acute brain slices, and a 21% oxygen atmosphere and glucose containing aCSF medium was performed in no OGD group slices parallelly. The membrane impermeant dye propidium iodide (PI) was used to exclude from viable cells, the membrane-permeable dye hoechst33342 was used to label cells (nuclei) indiscriminately, and the NeuroTrace Deep-Red Fluorescent Nissl dye was used to label neurons in this assay. We found that not only in the all brain cells (hoechst33342 positive), but also in the neurons (Nissl positive), 1 hour OGD insult could induce comparable cell death increment in SMACreER:ME-Linker mice and littermates (Figure S11b to S11e). The neuron percentage in cerebral cortex was also not changes between two genotypes (Figure S11f and S11g). (Line 478-486)

6. The ischemic damage was assessed by fluoro-jade stain, an indirect marker of neuronal injury. A more robust assessment with histology should be performed at different times after ischemia to determine the volume of injury and if the protection is maintained over time.

Response:

The authors thank for this suggestion. To increase the robustness of the histological assessment, we performed two additional assays to identify the short-term (Figure 7a-e) and the long-term (Figure S13) volume of ischemic injury respectively. TTC staining was occupied in the short-term histological assessment. In the long-term histological assessment, we checked the expression of neuronal marker Map2 and NeuN, and also the fluorescent Nissl morphology in the serial brain slices after 2 hours occlusion and 2 weeks reperfusion. The results are discussed as below.

Fig. 7. a, Scheme of timepoints for MCAO surgery, 2PLSM and experimental design. **b**, Statistical analysis of the relative CBF changes (LDF method) at different time points before and after MCAO or sham surgery in control (ME-Linker) and SMACreER:ME-Linker mice (N = 4 mice for each group). All measurements were normalized to the basal CBF before surgery. Data are expressed as the mean \pm SEM. **c**, TTC staining for control (ME-Linker) and SMACreER:ME-Linker mice after 2 days of MCAO or sham surgery (N = 4~7 mice). **d** and **e**, Statistical analysis of the total infarct volume and the infarct ration (% contralateral hemisphere) using TTC staining.

Fig S13. ME-Linker overexpression in SMCs attenuate behavioral and histological injuries after ischemic stroke. **a**, Scheme of timepoints for MCAO surgery, NSS (neurological severity score) detection, and histological assessment in the long-term (2 weeks) of ischemic-induced neuronal injury evaluation assay. **b**, Statistical analysis of the NSS along two weeks after ischemic stroke in control (ME-Linker) and SMACreER:ME-Linker mice (N = 3 mice for each group). **c**, Heat map of inspection items included in the NSS examination along days after ischemic stroke, as shown with control (ME-Linker) mice in blue, and SMACreER:ME-Linker mice in red. **d**, Representative images of the mouse brains after ischemic stroke in control (ME-Linker) and SMACreER:ME-Linker mice in the timepoint of two weeks

after ischemic stroke. **e**, Statistical analysis of the cortical hemisphere atrophy rate after two weeks ischemic stroke in control (ME-Linker) and SMACreER:ME-Linker mice (N = 3 mice). **f**, (Left) Representative images of Map2 staining results in two weeks post-ischemic brain slices, the boundary of the ischemic area (magenta dotted line) was distinguished by the positive signal of Map2 labeled neuronal processes. (Right) Representative serial section of Map2 staining slices in control (ME-Linker) and SMACreER:ME-Linker mice after two weeks ischemic stroke. **g**, Statistical analysis of the neuronal injury (Map2 loss) volume after two weeks ischemic stroke in control (ME-Linker) and SMACreER:ME-Linker mice (N = 3 mice). **h**, (Left) Representative images of NeuN staining results in two weeks post-ischemic brain slices, the boundary of the ischemic area (magenta dotted line) was distinguished by the positive signal of NeuN labeled neuronal somas. (Right) Representative serial section of NeuN staining slices in control (ME-Linker) and SMACreER:ME-Linker mice after two weeks ischemic stroke. **i**, Statistical analysis of the neuronal injury (NeuN loss) volume after two weeks ischemic stroke in control (ME-Linker) and SMACreER:ME-Linker mice (N = 3 mice). **j**, (Left) Representative images of NeuroTrace Fluorescent Nissl staining results in two weeks post-ischemic brain slices, the boundary of the ischemic area (magenta dotted line) was distinguished by the breakage signal of Nissl substance around neuronal somas. (Right) Representative serial section of NeuroTrace Fluorescent Nissl staining slices in control (ME-Linker) and SMACreER:ME-Linker mice after two weeks ischemic stroke. **k**, Statistical analysis of the neuronal injury (Nissl breakage) volume after two weeks ischemic stroke in control (ME-Linker) and SMACreER:ME-Linker mice (N = 3 mice). Data are expressed as the mean \pm SEM. Statistics were analyzed using unpaired t-tests.

Strikingly, we found the no-reflow was fairly rescued (Figure 7b), followed by the significant decrease of the total infarct volume and infarct ratio by TTC at the time point of two days after occlusion, in SMACreER:ME-Linker mice comparing to control mice (Figure 7c to 7e). (Line 486-492)

In the long-term (two weeks) ischemic injury examination of the ipsilateral atrophied brains (Figure S13d and S13e), consistent with the FJC staining results, we found that the neuronal injury (including map2/NeuN loss and Nissl breakage) volume was declined significantly in SMACreER:ME-Linker mice (Figure S13f to S13k), indicates a positive role in post-ischemic neuroprotection. (Line 508-512)

7. The brain sections showing hemispheric atrophy do not show evidence of focal brain damage (fig. 7C). Therefore, the injury leading to subsequent atrophy cannot be appreciated.

Response:

We apologize for the confusion. Actually, the brain samples used in the hemispheric atrophy analysis (Figure 7i) are collected from the LSCI assay (Figure 7g) after 15 days reperfusion of 2 hours occlusion, which is also applied in the neuronal injury examination of the following FJC staining (Figure 7k). In other words, the positive FJC staining in the ipsilateral hemisphere is sufficient to show evidence of focal brain damage. To increase readability, we have added the scheme of timeline and experimental design (Figure 7f) of the assay (and also other assays in this paper).

8. The language needs editing e.g. line 145, 249, 250, 274, 278, etc.

Response:

All the textual unreadability has been corrected as follows.

We have substituted the words “fair and capable” with “rigorous and unbiased”. (Page8, Line160)

We have eliminated “that underlie myogenic spontaneous vasomotion” to increase conciseness. (Page13, Line279)

We revised the sentence as “to decode the mystery of spontaneous vasomotion recession post-ischemia.....”. (Page13, Line280)

We have substituted the words “clocker” with “oscillation” in this sentence. (Page14, Line318)

We revised the sentence as “.....could induce mitochondria-ER dissociation and Ca²⁺ oscillation recession.....”. (Page15, Line320)

References

- 1 Ames, A., 3rd, Wright, R. L., Kowada, M., Thurston, J. M. & Majno, G. Cerebral ischemia. II. The no-reflow phenomenon. *Am J Pathol* **52**, 437-453 (1968).
- 2 Hill, R. A. *et al.* Regional Blood Flow in the Normal and Ischemic Brain Is Controlled by Arteriolar Smooth Muscle Cell Contractility and Not by Capillary Pericytes. *Neuron* **87**, 95-110 (2015). <https://doi.org:10.1016/j.neuron.2015.06.001>
- 3 Cipolla, M. J., McCall, A. L., Lessov, N. & Porter, J. M. Reperfusion decreases myogenic reactivity and alters middle cerebral artery function after focal cerebral ischemia in rats. *Stroke* **28**, 176-180 (1997). <https://doi.org:10.1161/01.str.28.1.176>
- 4 Shrouder, J. *et al.* Continued dysfunction of capillary pericytes promotes no-reflow after experimental stroke in vivo. *bioRxiv*, 531258 (2023). <https://doi.org:10.1101/2023.03.06.531258>
- 5 El Amki, M. *et al.* Neutrophils Obstructing Brain Capillaries Are a Major Cause of No-Reflow in Ischemic Stroke. *Cell Rep* **33**, 108260 (2020). <https://doi.org:10.1016/j.celrep.2020.108260>
- 6 Farina, A., Fasano, A. & Rosso, F. Modeling of vasomotion in arterioles. *J Theor Biol* **544**, 111124 (2022). <https://doi.org:10.1016/j.jtbi.2022.111124>
- 7 Scarselli, D., Lopez, J. M., Varshney, A. & Hof, B. Turbulence suppression by cardiac-cycle-inspired driving of pipe flow. *Nature* **621**, 71-74 (2023). <https://doi.org:10.1038/s41586-023-06399-5>
- 8 Dunn, A. K., Bolay, H., Moskowitz, M. A. & Boas, D. A. Dynamic imaging of cerebral blood flow using laser speckle. *J Cereb Blood Flow Metab* **21**, 195-201 (2001). <https://doi.org:10.1097/00004647-200103000-00002>
- 9 Ayata, C. *et al.* Laser speckle flowmetry for the study of cerebrovascular physiology in normal and ischemic mouse cortex. *J Cereb Blood Flow Metab* **24**, 744-755 (2004). <https://doi.org:10.1097/01.Wcb.0000122745.72175.D5>
- 10 Kazmi, S. M., Richards, L. M., Schrandt, C. J., Davis, M. A. & Dunn, A. K. Expanding applications, accuracy, and interpretation of laser speckle contrast imaging of cerebral blood flow. *J Cereb Blood Flow Metab* **35**, 1076-1084 (2015). <https://doi.org:10.1038/jcbfm.2015.84>

Reviewer #2 (Remarks to the Author):

This study examines the rhythmicity of spontaneous vasomotion in the cerebral vasculature in vivo, and how it is affected during ischemic stroke. The authors correlated the spontaneous vasomotion activity with calcium signaling and stable ER-mitochondria contacts in vascular smooth muscle cells. The data suggest that both calcium oscillations and ER-mitochondria contact sites are altered after ischemic stroke. Data also suggest that re-establishing/strengthening the ER-mitochondria contacts prevents the loss of calcium oscillations, restores flow to the capillary region faster and more effectively, and ameliorate brain atrophy and cell death compared to control conditions. The reviewer has several comments for the authors' consideration.

Response:

The authors thank for the careful review of our manuscript.

1) A key concern is about physiological relevance. First, the spontaneous oscillations in control vascular smooth muscle are relatively small. Thus, it needs to be clarified how much they influence vascular function. Moreover, while the spontaneous oscillations go away after ischemic stroke and that correlates with no-reflow, the loss of spontaneous oscillations may result from the insult rather than a cause/effect.

Response:

The authors thank for this feedback. While the vascular spontaneous oscillation occurring at the frequencies below the rate of respiration had been observed experimentally as early as in 1853 in bat wings¹, the direct physiological relevance of vasomotion remains vague. Indeed, as it is shown in our data from anesthetized mice, the oscillatory amplitude of the cerebral arteriolar radius is 3.263 ± 0.251 % (Figure 1h), which is relative slight but robust, and could only be captured under high-resolution microscopy in vivo. However, there are increasing evidences to show that vasomotion and its corresponding flowmotion could promote functional microcirculation^{2,3}, protect tissue oxygenation⁴⁻⁷, and drive paravascular clearance of solutes^{8,9} in the central nervous system (Line 66-69). Based on these previous foundations, in order to validate the efficiency of arteriolar spontaneous oscillation in alleviating post-ischemic no-reflow, we created SMACreER:ME-Linker mice and found mitochondria and ER tethers in SMCs could restore cerebral optimal circulation through renovating intracellular calcium oscillation. We advanced the research area by demonstrating that the vasomotion restoration could promote cerebral circulation after recanalization, which links the vasomotion dysfunction to the pathophysiological relevance in ischemic stroke. In this way, the loss of spontaneous oscillations is proved to be one of the causal factors in no-reflow formation.

2) The nature of the calcium signals needs to be clarified. First, the authors give

the impression that they could be IP3R-mediated signals but also be mitochondria. The authors should establish the identity of these calcium oscillations. Also, what is a calcium clocker?

Response:

A full discussion around this topic has been added in this revision as below.

The authors apologize for the lack of clarification with the discussion of the origins of the spontaneous calcium oscillations in cerebral arteriolar SMCs. The ionic mechanisms of arteriolar vasomotion and been fully reviewed by Willam C. Cole et al¹⁰. In this review, based on massive published data, the authors summarize the current view that arterial vasomotion is dependent on a cytosolic oscillator involving the periodic release of internal calcium stores via IP3R from ER, coupled to rhythmic oscillations in membrane depolarization and eventually relied on the temporal oscillatory control over calcium-dependent of cross-bridge cycling¹⁰. Besides, in our current study, the spontaneous calcium oscillations can be also intruded by the decoupling of mitochondria and ER under the mitochondrial uncoupler challenges (CCCP) in vitro (Figure 4d and 4k) or ischemic challenges in vivo (Figure 3b and 5b), suggesting the necessity of mitochondrial calcium buffering function in maintaining spontaneous calcium oscillations in SMCs. Our data is consistent with the previous studies, where they found the Ca²⁺ shuttling between the mitochondria and ER has alluded to a pacemaker role in the generation of Ca²⁺ oscillation¹¹, and also mitochondria could provide a wide dynamic buffering range in the control of calcium signals in SMCs¹².

Fig S8. IP3R inhibition with 2-APB suppress calcium oscillation in primary SMCs

a, Experimental timeline of live imaging before and after 2-APB (10 μ M) in Fluo-4 loaded primary SMCs. **b**, Still-frame images and kymographs of primary-culture SMCs (red, Tdt+) before and after 2-APB (10 μ M) treatment. The calcium signal was indicated by Fluo-4. The magenta dots in kymographs indicate the calcium oscillation peaks in the ROI area (yellow line). **c**, Representative time-lapse fluorescent Fluo-4 signal changes trace in the cytoplasm of primary-culture SMCs before and after 2-APB (10 μ M) treatment. **d**, Statistical analysis of the calcium oscillation frequency and amplitude before and after 2-APB (10 μ M) treatment in primary-culture SMCs (N = 3 assays, n = 40 cells). Data are expressed as the mean \pm SEM. Statistics were analyzed using paired t-tests.

Regarding to validate the crucial role of IP3R in maintaining spontaneous calcium oscillation in our experimental setup, in this revision, we performed the live imaging assays in the cultured primary SMCs loaded with calcium probe Fluo-4. As shown in Figure S8, 2-APB (IP3R inhibitor) administration could block the spontaneous calcium oscillation in SMCs significantly, suggesting IP3R-induced calcium release should be the main source of the calcium oscillations intracellularly. (Line 324-327) In this way, we proposed here that both the calcium releasing through IP3R from ER, and the calcium transduction between mitochondria and ER had played critical roles in maintaining the spontaneous calcium oscillation in SMCs. Yet, we haven't checked how extracellular calcium may participate in this loop due to space limitations. Despite, in facing to the ischemia-induced mitochondria dysfunction, mitochondria and ER tethering should and have proved to be an efficient strategy to restore calcium oscillation in SMCs. The calcium clock (sorry for the spelling errors with clocker) is the intracellular oscillatory calcium releases, which is coupled with the surface membrane voltage clocks, to control the timekeeping mechanism of the sinoatrial nodal pacemaker cells¹³. In the first draft of this study, calcium clock refers to cytosolic calcium oscillation. To avoid misunderstanding, we have substituted calcium clock with calcium oscillation in this revision.

3) Data seem to suggest that maintaining ER-mitochondria contact sites in artery/arteriole vascular smooth muscle cells prevents/ameliorates any capillary blood flow dysfunction. This result contrasts with recent studies indicating that capillary blood flow restoration during pathology is mainly mediated by recovery of capillary function triggering retrograde signaling that dilates upstream arterioles and arteries. The question is, then, how does restoring altered mechanisms in vascular smooth muscle correct capillary dysfunction?

Response:

The authors thank for raising this important topic related to the work of Dr. Mark T. Nelson group¹⁴⁻¹⁶, where they found that capillary networks could initiate a retrograde, propagating, hyperpolarizing electrical signal that dilates upstream arterioles and increases CBF. In their studies, they proposed that arteriolar SMCs may not be ideal sensors endogenous vasoactive reagents, as the density of SMC-containing arterioles is relatively low within the cortex¹⁶. On the one hand, we agree the hypothesis that SMC-containing arterioles are less likely to interact with endogenous vasoactive reagents in comparison to the high-density cerebral capillary networks. On the other hand, we believe that cerebral arteries may play its active role in regulating CBF, as the ability of contractile arteriolar SMC to control regional blood flow has been proved in live mice¹⁷. Especially in our study, we are focusing on the dynamic feature of the cerebral arteries, to explore the relationship between arteriolar myogenic vasomotion and CBF regulation under ischemic challenges. Accordingly, we found that the reestablishment of myogenic spontaneous vasomotion along the arteriolar wall could alleviate post-ischemic no-reflow and neuronal injury, indicating that the upstream arterioles are capable to promote downstream circulation through evoking corresponding flowmotion. We also proposed here that the manipulation

of vascular diameter is not the only mechanism to regulate cerebral circulation in vivo. Our results are consistent with recently published theoretical biology papers, where they explained the efficient mechanism of vasomotion for the spatial regulation of microcirculation¹⁸, and the turbulence suppression function of pulsatile flows¹⁹. Certainly, all these theoretical discoveries should be and will be validated by biological assays in the future of our studies.

4) There are several concerns with the fluorescent imaging data and interpretation that require further validation. For example, is the expression of the GCaMP6 calcium sensor similar in control and ischemic mice?

Response:

The authors thank for this important feedback. Statements about this topic have been added in this revision as below.

In regarding to this concern, we are of the opinion that rather than the expression of the sensor, the effectiveness of this tool should be validated using functional assays. The concentrated KCl solution is widely used to mimic cortical spreading depolarization through producing breakdown of ion gradients²⁰. In this model, spreading depolarization shift is normally followed by a drastic vasoconstriction²⁰. To verify whether the GCaMP6s calcium sensor could indicate calcium levels in SMCs after ischemic stroke, we have performed additional calcium imaging experiments in the post-ischemic arterioles under KCl-triggered cortical spreading depolarization conditions.

Fig 3. j, Scheme of the KCl (300mM) administration strategy on the brain surface during post-ischemic period and experimental timeline of 2PLSM. **k**, Kymographs of MCA before and after (Occ.2hRep.22h) ischemic stroke under 2PLSM. Especially, high dose of KCl (300mM) was administrated at the timepoint of Occ.2hRep.22h, and a drastic KCl-induced spreading depolarization and vasoconstriction can be observed in these vasomotion/calcium oscillation depressed arterioles (under normal saline treatment conditions). The red signals are the intravenous injected RhoB dye and represents arteriolar lumen. The green signals in the vascular wall represents Ca^{2+} signals indicated by GCaMP6s. **l**, Time-lapse fluorescent GcaMP6s signal changes trace in the MCA along with ischemic stroke, and high dose of KCl administration (N = 3 mice).

Interestingly, we found that 300mM KCl treatment through the administration hole could evoke a remarkable growth of Ca²⁺ signals in arteriolar SMCs, where the calcium oscillation was once suppressed by ischemic insults (Figure 3k and 3l). In this way, the suppression of calcium oscillation in post-ischemic SMCs are excluded from the possibility of the GCaMP6s calcium sensor dysfunction. (Line 300-312)

The results are included in Figure 3j-k in this revision. In this way, the suppression of calcium oscillation in post-ischemic SMCs are excluded from the possibility of the GCaMP6s calcium sensor dysfunction.

The calcium imaging data in Figure 4J is not convincing. There are no apparent changes in calcium oscillations from the images provided. Also, the loss of basal signal in CCCP suggests changes in calcium sensor conditions that may confound data interpretation.

Response:

In the in vitro calcium live imaging assay, the interval of the time series frame-imaging was set as 1 second (to observe spontaneous calcium oscillation with ultra-slow frequency), while the resolution in spatial can be as high as 0.1615 $\mu\text{m}/\text{px}$. In this way, in spite of the robust spontaneous calcium oscillation can be observed easily in the movie data (see movie 4), the oscillation peaks in the representative kymographic images could only be recognized carefully. In addition, the relative small of the spontaneous calcium oscillation amplitude in SMCs also raised the difficulty in recognizing these events. We had labeled the magenta dots to indicate the calcium oscillation peaks in kymographs. The spontaneous calcium oscillation analysis was performed by running customized code on MATLAB, where the events were defined when the amplitude of the calcium oscillation exceeded the double SD of the rolling baseline (see methods).

The calcium sensor (NES-YTnC2) was occupied in this assay in indicating spontaneous calcium oscillation in SMCs, with the properties of relatively smaller molecular size and reduced calcium ion buffering^{21,22}. In the control virus group cells, CCCP treatment could induce dramatic basal calcium and calcium oscillation loss, but not in these ME-Linker virus group cells (Figure 4k). These results suggest that the calcium sensor is functional and is capable to indicate cytosol calcium changes during hours of live imaging, before and after CCCP treatment.

Figure 5E-F: basal calcium measurements are challenging to assess in isolated systems much less in an intact native system. How did the authors perform the measurements to reach the conclusion that basal cytoplasm is higher in the ER:ME-linker mice? Is SR calcium content different in ER:ME-linker mice compared to control mice? Is mitochondria calcium different in ER:ME-linker mice compared to control mice?

Response:

The authors apologize for the confusion. We mistook the the labeling of the analyzed parameter in Y-axis, and it is now revised to the right one ($F_{\text{occ.2hrep.22h}} - F_{\text{before}} / F_{\text{before}} \%$) in this revision (Figure 5g). This indicator illustrates the basal cytoplasmic calcium levels changes rate, in other words, we caculated the changes rate of the cytoplasmic calcium levels before and after ischemic stroke in SMACreER: Ai96 and SMACreER: ME-linker mice, respectively. The comparasion was operated between two genotypes and found significantly differences. As shown in the data, the minus values of the cytoplasmic calcium levels changes rate in SMACreER: Ai96 indicate there is decreasing of the cytoplasmic calcium levels after ischemic stroke in SMC in vivo, while this phenomenon is not appaered in SMACreER: ME-linker mice. Our data suggest that the mitochondria-ER tethering in restoring Ca^{2+} oscillation works sufficiently in vivo. (Line 399-407)

We agree that we can't reach the conclusion that basal cytoplasmic calcium levels is higher in SMACreER: ME-linker mice in our present experimental setup. The SR and mitochondria calcium detection in vivo is not availabve in our lab at present since we need the corresponding report mouse lines first, despite, we appreciate for these suggestions and would like to check in the future of our studies.

Experiments using the TMRM compound to examine mitochondria membrane potential changes need to be more convincing. The loss of fluorescence could result from different loading compared to control mice. Could the authors use another marker that does not change with ischemic stroke to convincingly establish that the loading of any indicator/sensor is not affected by the ischemic stroke?

Response:

The authors thank for this suggestion.

Fig. 4. a, Experimental design of the in vivo SMC $\Delta\Psi_m$ detection under 2PLSM. Both the probe of TMRM and Mitotracker Green were loaded through the cranial window before imaging. **b**, Representative images of MCA labeling with TMRM (red) and Mitotracker Green (green) through the carinal window before and after ischemic stroke under 2PLSM in vivo. Green arrowheads represent blood flow direction. **c**, Statistical analysis of the relative TMRM and Mitotracker Green levels in SMCs of the MCA in the time points before, 10 min and 90 min after occlusion (N = 3 mice, n = 6 vessels).

In this revision, we used the mitochondrial probe mitotracker Green as the reference, in combination with TMRM to examine MMP changes during ischemic stroke in vivo. In addition, we had improved the labeling efficiency in methodology, by ripping the pial mater carefully and slightly during probe loading (see revised methods). We found mitochondrial membrane potential ($\Delta\Psi_m$) loss gradually after ischemia onset but not for the reference dye (Figure 4b and 4c).(Line 329-335)

5) Do the ER:ME-linker mice show higher cerebral artery myogenic tone, which is expected if there is more “cooperation” between smooth muscle cells? Examples showing smooth muscle cell synchronization will go a long way to support the authors’ claims.

Response:

The authors thank for this feedback.

Fig S10. v-x, Vasomotion index analysis between control littermates (N = 6 mice, n = 38 vessels) and *SMACreER:ME-Linker* mice (N = 5 mice, n = 33 vessels), including frequency of rhythmic cycles (**v**), standard deviation (SD) of the peak intervals (**w**), and amplitude of changes in the vascular radius (**x**).

Regarding to the myogenic vasomotion analysis under physiological condition in *SMACreER:ME-Linker* mice and littermates, we found ME-Linker overexpression in SMC in vivo could improve myogenic vasomotion through promoting vasomotion amplitude (Figure S10v). Interestingly, there is no significant differences in vasomotion frequency and peak interval SD between *SMACreER:ME-Linker* mice and littermates (Figure S10w and S10x), suggest that the myogenic vasomotion is relatively conserved in the frequency domain. The statement about the myogenic vasomotion analysis under physiological condition in two genotypes was included in this revision. (Line 382-387)

We are in favor of the idea that the elevated myogenic vasomotion in *SMACreER:ME-Linker* mice should be one of the mechanisms to facilitate cooperation index on pial

arterioles, and as it is. While we do not know the intercellular mechanism of ME-Linker to manipulate cooperation index yet. Interestingly, the cooperation activity between SMCs at hundred micrometers scale may launch differently with the well-documented synchronization activity occurred on a scale of millimeter and observed in the arterioles of isolated peripheral tissue ex vivo²³⁻²⁵, as they implicated in different scales. This is the reason we named this phenomenon as cooperation activity, to distinguish from the previous reported synchronization along arterioles.

Minor comments:

1) Page 7 – line 145: consider changing the words “fair and capable” to “rigorous and unbiased”.

Response:

The authors thank for this suggestion, we have substituted the words “fair and capable” with “rigorous and unbiased”. (Line 160)

2) Page 9 – Lines 177-178: What is “chaotic vasomotion”? Change wording.

Response:

Sorry for the confusion, our intention by using the word “chaotic” is trying to describe the “abnormal” changes of the post-ischemic vasomotion. We have substituted the words “chaotic” with “irregular”. (Line 192)

3) Page 14 – Lines 303-304: The conclusion in this sentence is not supported by the data presented. Authors should change the wording.

Response:

The authors agree with the comment, we have toned down the interpretation for the moment as how $\Delta\Psi_m$ may affect Ca^{2+} oscillation can not be concluded through the CLEM assay. We have changed the sentence “.....is involved in Ca^{2+} oscillation attenuation induced.....” to “.....can be affected”. (Line 352-353)

4) Page 21 – Lines 460-462: Statements in this section are overstated. The authors never tested calcium antagonists and how they may alter vasomotion. This should be eliminated.

Response:

The authors agree and accept this suggestion. The whole sentence related to calcium antagonists and its discussion have been eliminated.

5) It is unclear whether the statistical comparisons performed in Fig 2J are appropriate. How was the data normalized?

Response:

This data summarize the change rate of among parameters, including the arteriolar diameter and blood velocity, the capillary blood velocity, and the CBF with LSCI/LDF, between the timepoints of before and after (occ.2hrep.22h) ischemic stroke (Figure 2k). All the parameters were normalized to before conditions, the values less than 100%

indicate there is decreasing of the arteriolar/capillary blood velocity and the cortical CBF. While the change rate of the arteriolar diameter is 100.5 ± 1.097 % before and after (occ.2hrep.22h) ischemic stroke, suggesting arteriolar diameter changes is not correlated with ischemic challenges.

References

- 1 Jones, T. W. Discovery That the Veins of the Bat's Wing (Which Are Furnished with Valves) Are Endowed with Rhythmical Contractility, and That the Onward Flow of Blood Is Accelerated by Such Contraction. *Edinb Med Surg J* **79**, 367-373 (1853).
- 2 Intaglietta, M. Vasomotion and flowmotion: physiological mechanisms and clinical evidence. *Vascular Medicine Review* **vmr-1**, 101-112 (1990).
<https://doi.org/10.1177/1358836X9000100202>
- 3 Pittman, R. N. Oxygen transport in the microcirculation and its regulation. *Microcirculation* **20**, 117-137 (2013). <https://doi.org/10.1111/micc.12017>
- 4 Rickards, C. A., Ryan, K. L., Cooke, W. H. & Convertino, V. A. Tolerance to central hypovolemia: the influence of oscillations in arterial pressure and cerebral blood velocity. *J Appl Physiol* (1985) **111**, 1048-1058 (2011).
<https://doi.org/10.1152/jappphysiol.00231.2011>
- 5 Lucas, S. J., Lewis, N. C., Sikken, E. L., Thomas, K. N. & Ainslie, P. N. Slow breathing as a means to improve orthostatic tolerance: a randomized sham-controlled trial. *J Appl Physiol* (1985) **115**, 202-211 (2013). <https://doi.org/10.1152/jappphysiol.00128.2013>
- 6 Anderson, G. K., Sprick, J. D., Park, F. S., Rosenberg, A. J. & Rickards, C. A. Responses of cerebral blood velocity and tissue oxygenation to low-frequency oscillations during simulated haemorrhagic stress in humans. *Exp Physiol* **104**, 1190-1201 (2019).
<https://doi.org/10.1113/ep087358>
- 7 Anderson, G. K. *et al.* Peaks and valleys: oscillatory cerebral blood flow at high altitude protects cerebral tissue oxygenation. *Physiol Meas* **42** (2021).
<https://doi.org/10.1088/1361-6579/ac0593>
- 8 Fultz, N. E. *et al.* Coupled electrophysiological, hemodynamic, and cerebrospinal fluid oscillations in human sleep. *Science* **366**, 628-631 (2019).
<https://doi.org/10.1126/science.aax5440>
- 9 van Veluw, S. J. *et al.* Vasomotion as a Driving Force for Paravascular Clearance in the Awake Mouse Brain. *Neuron* **105**, 549-561.e545 (2020).
<https://doi.org/10.1016/j.neuron.2019.10.033>
- 10 Cole, W. C., Gordon, G. R. & Braun, A. P. Cellular and Ionic Mechanisms of Arterial Vasomotion. *Adv Exp Med Biol* **1124**, 297-312 (2019). https://doi.org/10.1007/978-981-13-5895-1_12
- 11 Ishii, K., Hirose, K. & Iino, M. Ca²⁺ shuttling between endoplasmic reticulum and mitochondria underlying Ca²⁺ oscillations. *EMBO Rep* **7**, 390-396 (2006).
<https://doi.org/10.1038/sj.embor.7400620>
- 12 McCarron, J. G., Olson, M. L. & Chalmers, S. Mitochondrial regulation of cytosolic Ca²⁺ signals in smooth muscle. *Pflugers Arch* **464**, 51-62 (2012).
<https://doi.org/10.1007/s00424-012-1108-9>

- 13 Lakatta, E. G., Maltsev, V. A. & Vinogradova, T. M. A coupled SYSTEM of intracellular Ca²⁺ clocks and surface membrane voltage clocks controls the timekeeping mechanism of the heart's pacemaker. *Circ Res* **106**, 659-673 (2010).
<https://doi.org:10.1161/circresaha.109.206078>
- 14 Longden, T. A. *et al.* Capillary K(+)-sensing initiates retrograde hyperpolarization to increase local cerebral blood flow. *Nat Neurosci* **20**, 717-726 (2017).
<https://doi.org:10.1038/nn.4533>
- 15 Gonzales, A. L. *et al.* Contractile pericytes determine the direction of blood flow at capillary junctions. *Proc Natl Acad Sci U S A* **117**, 27022-27033 (2020).
<https://doi.org:10.1073/pnas.1922755117>
- 16 Sancho, M. *et al.* Adenosine signaling activates ATP-sensitive K(+) channels in endothelial cells and pericytes in CNS capillaries. *Sci Signal* **15**, eab15405 (2022).
<https://doi.org:10.1126/scisignal.abl5405>
- 17 Hill, R. A. *et al.* Regional Blood Flow in the Normal and Ischemic Brain Is Controlled by Arteriolar Smooth Muscle Cell Contractility and Not by Capillary Pericytes. *Neuron* **87**, 95-110 (2015). <https://doi.org:10.1016/j.neuron.2015.06.001>
- 18 Farina, A., Fasano, A. & Rosso, F. Modeling of vasomotion in arterioles. *J Theor Biol* **544**, 111124 (2022). <https://doi.org:10.1016/j.jtbi.2022.111124>
- 19 Scarselli, D., Lopez, J. M., Varshney, A. & Hof, B. Turbulence suppression by cardiac-cycle-inspired driving of pipe flow. *Nature* **621**, 71-74 (2023).
<https://doi.org:10.1038/s41586-023-06399-5>
- 20 Ayata, C. & Lauritzen, M. Spreading Depression, Spreading Depolarizations, and the Cerebral Vasculature. *Physiol Rev* **95**, 953-993 (2015).
<https://doi.org:10.1152/physrev.00027.2014>
- 21 Barykina, N. V. *et al.* NTnC-like genetically encoded calcium indicator with a positive and enhanced response and fast kinetics. *Sci Rep* **8**, 15233 (2018).
<https://doi.org:10.1038/s41598-018-33613-6>
- 22 Subach, O. M. *et al.* YTnC2, an improved genetically encoded green calcium indicator based on toadfish troponin C. *FEBS Open Bio* **13**, 2047-2060 (2023).
<https://doi.org:10.1002/2211-5463.13702>
- 23 Segal, S. S. & Duling, B. R. Conduction of vasomotor responses in arterioles: a role for cell-to-cell coupling? *Am J Physiol* **256**, H838-845 (1989).
<https://doi.org:10.1152/ajpheart.1989.256.3.H838>
- 24 Peng, H., Matchkov, V., Ivarsen, A., Aalkjaer, C. & Nilsson, H. Hypothesis for the initiation of vasomotion. *Circ Res* **88**, 810-815 (2001). <https://doi.org:10.1161/hh0801.089603>
- 25 Hashitani, H., Mitsui, R., Masaki, S. & Van Helden, D. F. Pacemaker role of pericytes in generating synchronized spontaneous Ca²⁺ transients in the myenteric microvasculature of the guinea-pig gastric antrum. *Cell Calcium* **58**, 442-456 (2015).
<https://doi.org:10.1016/j.ceca.2015.06.012>

Reviewer #3 (Remarks to the Author):

The present study investigated how cerebral ischemia reperfusion injury affects physiological cerebral vasomotion, as well as mechanisms underlying observed post-reperfusion deficits. The authors observed that Ca²⁺ oscillation in SMC underlie physiological arteriolar vasomotion in mice, likely due to communication between SR and mitochondria, all these processes are disrupted by ischemia / reperfusion. Interestingly, expression of a linker peptide increases areas of proximity between SR and mitochondria, partially restoring Ca²⁺ oscillations and vasomotion. Overall, this is a well-designed and executed study, and the findings are relevant to the field and add to the knowledge of cellular alterations caused by ischemia-reperfusion that affect cerebral microvascular control.

Response:

The authors thank for the careful review of our manuscript.

Some minor issues were identified, described in detail below:

1. Unclear why the authors chose to use CCCP in cell culture experiments to investigate mitochondria-SR linkages rather than performing an oxygen-glucose deprivation model, which fits better with the MCAO data in the remainder of the manuscript. Given that the authors generated a mouse strain expressing the ME-linker in SMC, the cell culture experiments become unnecessary, as they do not add much value to the final conclusion of the study. This reviewer's recommendation is to either remove them from the manuscript or repeat the experiments in an OGD paradigm;

Response:

Thanks for this feedback. CCCP was used in SMC in vitro to mimic the urgent MMP loss after the on-set of ischemic stroke in vivo. OGD was not occupied in the current study as we found SMC constricted severely during OGD, which interrupted calcium live imaging extremely. We insist to retain these results in the manuscript, as these results had suggested the close relationship between mito-ER contact and spontaneous calcium oscillation in SMCs. In this way, logically, lead us to establish the ME-linker-floxed mouse in the following study.

2. The mitochondria-SR link in control of SMC Ca²⁺ oscillations is an interesting hypothesis, and the data presented in the manuscript are compelling. It would be interesting to investigate the mechanisms underlying this communication, and where the Ca²⁺ is coming from and what organelle is buffering Ca²⁺ release. Are the Ca²⁺ oscillations coming from opening of IP3R or RyR2 in the SR, then buffered by mitochondria? Or is the opposite happening, where mitochondria are responsible for Ca²⁺ release and SERCA buffers it? This should be explored further, if not experimentally at least in the Discussion;

Response:

Thank you for your attention to this important topic, a fully discussion around this topic has been added in this revision as below.

The authors apologize for the lack of clarification with the discussion of the origins of the spontaneous calcium oscillations in cerebral arteriolar SMCs. The ionic mechanisms of arteriolar vasomotion and been fully reviewed by Willam C. Cole et al¹. In this review, based on massive published data, the authors summarize the current view that arterial vasomotion is depndent on a cytosolic oscillator involving the periodic release of internal calcium stores via IP3R from ER, coupled to rhythmic oscillations in membrane depolorization and eventually relied on the temporal oscillatory control over calcium-dependent of cross-bridge cycling¹. Besides, in our current study, the spontaneous calcium oscillations can be also intruped by the decoupling of mitochondria and ER under the mitochondrial uncoupler challenges (CCCP) in vitro (Figure 4d and 4k) or ischemic challenges in vivo (Figure 3b and 5b), suggesting the necessity of mitochondrial calcium buffering function in maintaining spontaneous calcium oscillations in SMCs. Our data is consitent with the previous studies, where they found the Ca²⁺ shuttling between the mitochondria and ER has alluded to a pacemaker role in the generation of Ca²⁺ oscillation², and also mitochondria could provide a wide dynamic buffering range in the control of calcium signals in SMCs³.

Fig S8. IP3R inhibition with 2-APB suppress calcium oscillation in primary SMCs

a, Experimental timeline of live imaging before and after 2-APB (10 μ M) in Fluo-4 loaded primary SMCs. **b**, Still-frame images and kymographs of primary-culture SMCs (red, Tdt+) before and after 2-APB (10 μ M) treatment. The calcium signal was indicated by Fluo-4. The magenta dots in kymographs indicate the calcium oscillation peaks in the ROI area (yellow line). **c**, Representative time-lapse fluorescence trace in the cytoplasm of primary-culture SMCs before and after 2-APB (10 μ M) treatment. **d**, Statistical analysis of the calcium oscillation frequency and amplitude before and after 2-APB (10 μ M) treatment in primary-culture SMCs (N = 3 assays, n = 40 cells). Data are expressed as the mean \pm SEM. Statistics were analyzed using paired t-tests.

Regarding to validate the crucial role of IP3R in maintaining spontaneous calcium oscillation in our experimental setup, in this revision, we performed the live imaging assays in the cultured primary SMCs loaded with calcium probe Fluo-4. As shown in Figure S8, 2-APB (IP3R inhibitor) administration could block the spontaneous calcium oscillation in SMCs significantly, suggesting IP3R-induced calcium release should be the main source of the calcium oscillations intracellularly. (Line 324-327) In this way, we proposed here that both the calcium releasing through IP3R from ER, and the calcium transduction between mitochondria and ER had played critical roles in maintaining the spontaneous calcium oscillation in SMCs. Yet, we haven't checked how extracellular calcium may participate in this loop due to space limitations. Despite, in facing to the ischemia-induced mitochondria dysfunction, mitochondria and ER tethering should and have proved to be an efficient strategy to restore calcium oscillation in SMCs.

3. Measurement of basal Ca²⁺ levels in SMC using genetically-encoded indicators are unreliable, as differences in basal fluorescence can be explained by distinct pattern in GCaMP6 expression post-ischemia / genetic manipulation rather than real alterations in cytosolic Ca²⁺. These measurements should be removed from the manuscript, or corroborated using a different technique (for example, ratiometric dyes such as Fura2-AM);

Response:

The authors thank for this important feedback. Statements about this topic have been added in this revision as below.

Fig 3. j, Scheme of the KCl (300mM) administration strategy on the brain surface during post-ischemic period and experimental timeline of 2PLSM. **k**, Kymographs of MCA before and after (Occ.2hRep.22h) ischemic stroke under 2PLSM. Especially, high dose of KCl (300mM) was administrated at the timepoint of Occ.2hRep.22h, and a drastic KCl-induced spreading depolarization and vasoconstriction can be observed in these vasomotion/calcium oscillation depressed arterioles (under normal saline treatment conditions). The red signals are the intravenous injected RhoB dye

and represents arteriolar lumen. The green signals in the vascular wall represents Ca^{2+} signals indicated by GCaMP6s. I, Time-lapse fluorescent GcaMP6s signal changes trace in the MCA along with ischemic stroke, and high dose of KCl administration (N = 3 mice).

In regarding to this concern, we are of the opinion that rather than the expression of the sensor, the effectiveness of this tool should be validated using functional assays. The concentrated KCl solution is widely used to mimic cortical spreading depolarization through producing breakdown of ion gradients⁴. In this model, spreading depolarization shift is normally followed by a drastic vasoconstriction⁴. To verify whether the GCaMP6s calcium sensor could indicate calcium levels in SMCs after ischemic stroke, we have performed additional calcium imaging experiments in the post-ischemic arterioles under KCl-triggered cortical spreading depolarization conditions.

Interestingly, we found that 300mM KCl treatment through the administration hole could evoke a remarkable growth of Ca^{2+} signals in arteriolar SMCs, where the calcium oscillation was once suppressed by ischemic insults (Figure 3k and 3l). In this way, the suppression of calcium oscillation in post-ischemic SMCs are excluded from the possibility of the GCaMP6s calcium sensor dysfunction. (Line 300-312)

The results are included in Figure 3j-k in this revision. In this way, the suppression of calcium oscillation in post-ischemic SMCs are excluded from the possibility of the GCaMP6s calcium sensor dysfunction.

4. In Supplemental Figure S8 the authors included diastolic blood pressure and heart rate, were measured using tail-cuff pletismography. Both measurements are unreliable using this technique, thus they should not be included in the manuscript, even though the data show that there is no difference between groups;

Response:

The authors agree and accept this suggestion, we have eliminated the diastolic blood and heart rate data collected by the tail-cuff plethysmography.

Fig S10. ME-Linker overexpression in SMCs does not affect cardiac functions, blood pressure, and body temperature. **a**, Representative echocardiographic images in B-Mode and M-Mode of the left ventricular (LV) in control littermates and *SMACreER:ME-Linker* mice. **b-m**, Statistical analysis of heart rate (**b**), LV mass (**c**), LV ejection fraction (**d**), LV fractional shortening (**e**), cardiac output (**f**), stroke volume (**g**), and ventricular end-systolic (;s) or end-diastolic (;d) left ventricular interior diameter (LVID) (**h** and **i**), left ventricular posterior wall (LVPW) (**j** and **k**), and left ventricular anterior wall (LVAV) (**l** and **m**) in control littermates and *SMACreER:ME-Linker* mice. **n**, Representative echocardiographic images in PW Doppler-Mode and PW tissue-Mode in detecting mitral valve (MV) blood velocity and MV velocity. **o**, Statistical analysis of MV function by calculating E/E' values in control littermates and *SMACreER:ME-Linker* mice. **p**, Representative echocardiographic images in PW Doppler-Mode of aortic arch (Ao Arch), aortic valve (AV) and pulmonary valve (PV). **q-s**, Statistical analysis of Ao Arch (**q**), AV(**r**) and PV(**s**) peak velocity in control littermates and *SMACreER:ME-Linker* mice. **t**, Analysis of systolic blood pressure in control littermates and *SMACreER:ME-Linker* mice before and after 1 week of tamoxifen treatment (N = 3 or 5 mice). **u**, Analysis of body temperature in control littermates and *SMACreER:ME-Linker* mice before and after 1 week of tamoxifen treatment (N = 5 or 7 mice). Data are expressed as the mean \pm SEM. All data were analyzed using unpaired t-tests.

Moreover, we included the heart function detection using transthoracic echocardiography in this revision (Figure S10). We found there is no structural or functional differences along comprehensive heart examination processes between SMACreER:ME-Linker mice and littermates (Figure S10a-s). (Line 376-378)

5. In the methodology it is stated that the study included both male and female mice, but there is no mention if the data were segregated and analyzed for potential sex differences in responses. These should be performed and included in the manuscript. Ideally, each datapoint in graphs should be segregated by biological sex;

Response:

Thanks for this feedback.

Fig. S2. Arteriolar vasomotion index analysis between male and female mice.

a-c, Vasomotion index analysis between male (N = 6 mice, n = 38 vessels) and female (N = 5 mice, n = 33 vessels), including frequency of rhythmic cycles (a), standard deviation (SD) of the peak intervals (b), and amplitude of changes in the vascular radius (c). Data are expressed as the mean \pm SEM. Unpaired t-tests were used for statistics.

We had recalculated by separating biological sex, and found there is no significant different in arteriolar vasomotion index between male and female mice, indicating sex-related factors had very limited impact on the characteristics of arteriolar myogenic vasomotion physiologically. (Line 155-157) All the results are included in the manuscript (Figure S2a-c).

6. There are some minor grammar inconsistencies and spelling errors throughout the manuscript. Suggest a review of English grammar and spelling.

Response:

The authors have checked and revised all the textual errors throughout the manuscript.

References

- 1 Cole, W. C., Gordon, G. R. & Braun, A. P. Cellular and Ionic Mechanisms of Arterial Vasomotion. *Adv Exp Med Biol* **1124**, 297-312 (2019). https://doi.org:10.1007/978-981-13-5895-1_12
- 2 Ishii, K., Hirose, K. & Iino, M. Ca²⁺ shuttling between endoplasmic reticulum and mitochondria underlying Ca²⁺ oscillations. *EMBO Rep* **7**, 390-396 (2006). <https://doi.org:10.1038/sj.embor.7400620>
- 3 McCarron, J. G., Olson, M. L. & Chalmers, S. Mitochondrial regulation of cytosolic Ca²⁺ signals in smooth muscle. *Pflugers Arch* **464**, 51-62 (2012). <https://doi.org:10.1007/s00424-012-1108-9>
- 4 Ayata, C. & Lauritzen, M. Spreading Depression, Spreading Depolarizations, and the Cerebral Vasculature. *Physiol Rev* **95**, 953-993 (2015). <https://doi.org:10.1152/physrev.00027.2014>

Reviewers' comments:

Reviewer #1 (Remarks to the Author):

The authors have done an excellent job at revising the paper, including a large amount of new data to address the comments of the referee.

A minor question concerns whether the methods used to measure dynamic changes in CBF (LSCI, LDF) can also be used to measure resting CBF in absolute terms. Knowledge of absolute CBF (ml/100g/min) would be needed to state that CBF before induction of ischemia is the same in transgenic and WT littermates.

Reviewer #3 (Remarks to the Author):

The authors have performed a thorough and careful revision of the original manuscript, and included new data, as well as more detailed discussions of the findings. As a result, the manuscript is greatly improved.

Reviewer #4 (Remarks to the Author):

The revision has solidified the manuscript quite significantly. I highly appreciate the additional figures and experiments included in the manuscript and rebuttal letter to clarify some confusing points. I have a few extra comments.

In continuity of discussion in question 4): CCCP treatment in ME-linker virus group cells (Figure 4k) induced basal calcium loss. This can be observed in the background of the linescan traces in the lower right of Figure 4k. We still can not exclude the possibility of a change of calcium sensor conditions after applying CCCP.

Further, the location of both ROIs placed in the post CCCP conditions (Figure 4k upper right, lower right) cannot match with the linescan patterns. To be more detailed, based on the location of ROI in CCCP and control virus, the 2D linescan is supposed to show a black vertical stripe on the left, but not in the middle. Based on the location of ROI in CCCP and ME-linker, the background of 2D linescan should be very even, not with vertical black stripes.

In continuity of discussion in question 5): The authors' response does not answer the original questions. SMCs in ER:ME-linker mice may potentially have a higher basal calcium level than control littermates and, therefore higher cerebral artery myogenic tone, further constricting the arteries / arterioles in resting state. To clarify this, a comparison of calcium level and resting-state vessel diameters between ER:ME-linker and control mice will be appreciated. Secondly, the vasomotion amplitude, frequency and peak interval can not fully reflect the smooth muscle cell synchronization. The measurement should be performed across neighboring SMCs.

COMMSBIO-23-1417B, manuscript entitled "Ca²⁺ Oscillation in Vascular Smooth Muscle Cells Control Myogenic Spontaneous Vasomotion and Counteract Post-ischemic No-reflow"

Below are the point-by-point replies to the referees' concerns, with the comments in bold (black) our responses in regular (blue) text. The revised text in the manuscript were indicated by line number.

Reviewers' comments:

Reviewer #1 (Remarks to the Author):

The authors have done an excellent job at revising the paper, including a large amount of new data to address the comments of the referee.

Response:

We appreciate this support and thank for your valuable comments on our work.

A minor question concerns whether the methods used to measure dynamic changes in CBF (LSCI, LDF) can also be used to measure resting CBF in absolute terms. Knowledge of absolute CBF (ml/100g/min) would be needed to state that CBF before induction of ischemia is the same in transgenic and WT littermates.

Response:

Both LSCI and LDF provide continuous measurements of blood flow in the tissue sample volume, however, it is not appropriate to use absolute flow units for these techniques such as ml/100g/minute of tissue, due to the varying cerebral vasculature patterns in different animals. To justify the detect of CBF in absolute terms, it is necessary to calibrate for the particular site and tissue type before each measurement, which is impractical in reality.

Fig S11. ME-Linker overexpression in SMCs elevates basal cytoplasmic calcium levels and vasomotion amplitude, but does not intrups resting-state CBF and MCA diameter. a, LSCI images of the mouse whole brain in control littermates and SMACreER:ME-Linker mice. b, Statistical analysis of the LSCI perfusion unit in control and transgenic littermates (N = 10 or 8 mice, n = 20 or 16 ROIs).

It has been reported that, by using the noninvasive arterial spin labeling (ASL) MRI method, the absolute cortical CBF values in anesthetized C57BL/6J mice are measured by ranging from around 100 to 180 ml/100g/minute of tissue¹. These parameters can be used to state the resting absolute CBF values in the control mice with C57BL/6J background. Regarding to figure out whether there is any resting cortical CBF difference in transgenic and control littermates, we have analyzed the bilateral MCA blood flow before ischemia induction, by using the original LSCI perfusion unit in this revision (Supplementary Figure 11a and 11b). We found there is no significant difference between transgenic and control littermates in resting LSCI perfusion unit, indicating that ME-Linker overexpression in SMCs dose not interrupt the resting absolute cortical CBF values. (Line 381-384)

References

- 1 Munting, L. P. *et al.* Influence of different isoflurane anesthesia protocols on murine cerebral hemodynamics measured with pseudo-continuous arterial spin labeling. *NMR Biomed* **32**, e4105 (2019). <https://doi.org:10.1002/nbm.4105>

Reviewer #3 (Remarks to the Author):

The authors have performed a thorough and careful revision of the original manuscript, and included new data, as well as more detailed discussions of the findings. As a result, the manuscript is greatly improved.

Response:

We appreciate this support on our manuscript and thank for the help in improving our work.

Reviewer #4 (Remarks to the Author):

The revision has solidified the manuscript quite significantly. I highly appreciate the additional figures and experiments included in the manuscript and rebuttal letter to clarify some confusing points. I have a few extra comments.

Response:

We appreciate this support on our clarification and thank for the contribution to make our manuscript more precise on many fronts.

In continuity of discussion in question 4): CCCP treatment in ME-linker virus group cells (Figure 4k) induced basal calcium loss. This can be observed in the

background of the linescan traces in the lower right of Figure 4k. We still can not exclude the possibility of a change of calcium sensor conditions after applying CCCP.

Response:

We agree that we do not have experimental evidences to support the integrity of The calcium sensor (NES-YTnC2) after CCCP administration. The key point here we are going to propose is that ME-Linker is helpful in sustaining the calcium oscillation in SMCs after $\Delta\Psi_m$ loss. Under the identical imaging and drug treatment protocol, the calcium oscillation signals can be observed in these ME-Linker positive SMCs, meanwhile, the signals lost fastly after CCCP applying in control SMCs. Let's assume that NES-YTnC2 had been partially degraded after CCCP administration, these residual sensors were sufficient to indicate the the calcium oscillation signals (attenuated but sustained) in the ME-Linker positive SMCs, but not in these control SMCs. Not like the gradually SMC $\Delta\Psi_m$ loss after ischemic stroke in vivo (Figure 4a-c), this in vitro model in mimicking SMC $\Delta\Psi_m$ loss can be more directly and strongly, in this way, the calcium oscillation in SMC can be interrupted more fastly and robustly.

Further, the location of both ROIs placed in the post CCCP conditions (Figure 4k upper right, lower right) cannot match with the linescan patterns. To be more detailed, based on the location of ROI in CCCP and control virus, the 2D linescan is supposed to show a black vertical stripe on the left, but not in the middle. Based on the location of ROI in CCCP and ME-linker, the background of 2D linescan should be very even, not with vertical black stripes.

Response:

We thank for this careful review and apologize for the confusion about the ROI locations in Figure 4k. Indeed, we double checked the source data and found these ROIs were minorly mismatched due to manual labeling. In this revision, we have corrected and redrewed the ROIs with dotted line to indicate reslice position, in the representative frame-scan images and kymographs. To be more detailed, in upper right of Figure 4k, the inclined left black vertical stripe in the kymograph represents the small black hole in the SMC cytoplasm after CCCP treatment. In ME-Linker SMCs, CCCP treatment could attenuate the evenness of the cytoplasmic calcium signals (shown as in the upper left area of this SMC), which could be the result of the unevenness pattern of the mitochondria-ER contacts. In this way, in the lower right of Figure 4k, the vertical black stripes can be observed in the kymograph.

In continuity of discussion in question 5): The authors' response does not answer the original questions. SMCs in ER:ME-linker mice may potentially have a higher basal calcium level than control littermates and, therefore higher cerebral artery myogenic tone, further constricting the arteries / arterioles in resting state. To clarify this, a comparison of calcium level and resting-state vessel diameters between ER:ME-linker and control mice will be appreciated. Secondly, the vasomotion amplitude, frequency and peak interval can not fully reflect the smooth muscle cell synchronization. The measurement should be performed

across neighboring SMCs.

Response:

The authors thank for this feedback and the suggestion. In this revision, we integrate a comprehensive comparison of resting-state arteriolar-related parameters between SMACreER:ME-Linker and control mice in supplementary figure 11 (shown as below).

Fig S11. ME-Linker overexpression in SMCs elevates basal cytoplasmic calcium levels and vasomotion amplitude, but does not intrups resting-state CBF and MCA diameter. **a**, LSCI images of the mouse whole brain in control littermates and SMACreER:ME-Linker mice. **b**, Statistical analysis of the LSCI perfusion unit in control and transgenic littermates (N = 10 or 8 mice, n = 20 or 16 ROIs). **c**, Representative images of ME-Linker and GcaMP6s (Ai96) signals in SMACreER:ME-Linker and SMACreER:ME-Linker;Ai96 mice. **d**, Statistical analysis of basal cytoplasmic calcium levels between two transgenic mice (N = 2 mice, n = 50 SMCs). **e-g**, Vasomotion index analysis between control littermates (N = 6 mice, n = 38 vessels) and SMACreER:ME-Linker mice (N = 5 mice, n = 33 vessels), including frequency of rhythmic cycles (**e**), standard deviation (SD) of the peak intervals (**f**), and amplitude of changes in the vascular radius (**g**). **h**, (Upper) Representative images of cerebral vasculature and the shceme of the cranial

window/2PLSM imaging location. (Lower) Representative images of imaged MCA under 2PLSM in SMACreER: Ai47 and SMACreER:ME-Linker mice, the double-headed arrows indicate the measured main MCA. **d**, Statistical analysis of the resting-state MCA (around 6 mm distal from MCA root under 2PLSM) diameter (N = 4 mice, n = 8 vessels). Data are expressed as the mean \pm SEM. All data were analyzed using unpaired t-tests.

Firstly, we found there is no significant difference between transgenic and control littermates in resting LSCI perfusion unit (Figure S11a and 11b), indicating that ME-Linker overexpression in SMCs does not interrupt the resting absolute cortical CBF values. (Line 381-384) Furthermore, to compare the calcium level in SMC in vivo, we analyze our pre-prepared data of the SMACreER:ME-Linker: Ai96 mice, and the control SMACreER: Ai96 mice, which the basal SMC calcium signals were reported by the equivalent sensor and imaged by the identical laser intensity under 2PLSM. Importantly, we found ME-Linker overexpression in SMC in vivo could improve basal cytoplasmic calcium levels (Figure S11c and S11d) and myogenic vasomotion through promoting vasomotion amplitude (Figure S11e), but can not improve vasomotion frequency and peak interval SD (Figure S11f and S11g), suggest that the myogenic vasomotion is relatively conserved in the frequency domain. (Line 384-390) In addition, ME-Linker overexpression in SMC is not sufficient to change the resting-state main MCA diameter (around 6 mm distal from MCA root under 2PLSM) (Figure S11h and S11i), indicating there is no abnormal arteriolar constriction in SMACreER:ME-Linker mice. (Line 390-393) Above all, we conclude here that ME-Linker are capable to promote resting-state vasomotion amplitude through elevating basal calcium levels. Moreover, the calcium oscillation promotion by ME-Linker are limited within a homeostatic resting-state (Line 393-394), rather than an excessive calcium level to constrict arterioles and interrupt CBF.

We use the amplitude, frequency and peak interval SD to describe spontaneous vasomotion, rather than the SMC synchronization. As you suggested, we indeed developed a 'cooperation index' to characterize the micro view of the synchronization activity between neighboring SMCs (Line 439-440, Figure 6j). We use the term 'cooperation index' to investigate synchronization activity on pial arterioles at hundred micrometers scale, but not the 'synchronization index', as it has been well-documented synchronization activity occurred on a scale of millimeter (Line 434-435). We found ME-Linker could improve post-ischemic cooperation index. As you discussed before, in combination of the data in supplementary figure 11 in this revision, the elevation of the cooperation index in SMACreER:ME-Linker could be the result of the enhancement of the Ca^{2+} oscillation and vasomotion amplitude in SMCs.

REVIEWERS' COMMENTS:

Reviewer #1 (Remarks to the Author):

No further comments

Reviewer #4 (Remarks to the Author):

After careful revision and discussion, the manuscript is now in a good quality to be published.